# Radical chemistry in the Pearl River Delta: observations and modeling of OH and HO₂ radicals in Shenzhen 2018

Xinping Yang[1,2], Keding Lu[1,2,*], Xuefei Ma[1,2], Yue Gao[1,2], Zhaofeng Tan[3], Haichao Wang[4], Xiaorui Chen[1,2], Xin Li[1,2], Xiaofeng Huang[5], Lingyan He[5], Mengxue Tang[5], Bo Zhu[5], Shiyi Chen[1,2], Huabin Dong[1,2], Limin Zeng[1,2], Yuanhang Zhang[1,2,*]

[1]State Key Joint Laboratory of Environmental Simulation and Pollution Control, College of Environmental Sciences and Engineering, Peking University, Beijing, China

[2]State Environmental Protection Key Laboratory of Atmospheric Ozone Pollution Control, Peking University, Beijing, China

[3]Institute of Energy and Climate Research, IEK-8: Troposphere, Forschungszentrum Juelich GmbH, Juelich, Germany

[4]School of Atmospheric Sciences, Sun Yat-Sen University, Zhuhai, China

[5]Laboratory of Atmospheric Observation Supersite, School of Environment and Energy, Peking University Shenzhen Graduate School, Shenzhen, China

Correspondence to: Keding Lu (k.lu@pku.edu.cn), Yuanhang Zhang (yhzhang@pku.edu.cn)

**Abstract.** The ambient radical concentrations were measured continuously by laser-induced fluorescence during the STORM (STudy of the Ozone foRmation Mechanism) campaign at the Shenzhen site, located in the Pearl River Delta in China, in the autumn of 2018. The diurnal maxima were $4.5 \times 10^6$ cm⁻³ for OH radicals and $4.2 \times 10^8$ cm⁻³ for HO₂ radicals (including an estimated interference of 23%-28% from RO₂ radicals during the daytime), respectively. The state-of-the-art chemical mechanism underestimated the observed OH concentration, similar to the other warm-season campaigns in China. The OH underestimation was attributable to the missing OH sources, which can be explained by the X mechanism. Good agreement between the observed and modeled OH concentrations was achieved when an additional numerical X equivalent to 0.1 ppb NO concentrations was added into the base model. The isomerization mechanism of RO₂ derived from isoprene contributed approximately 7% to the missing OH production rate and the oxidation of isoprene oxidation products (MACR and MVK) had no significant impact on the missing OH sources, demonstrating further exploration of unknown OH sources is necessary. A significant HO₂ heterogeneous uptake was found in this study, with an effective uptake coefficient of 0.3. The model with the HO₂ heterogeneous uptake can simultaneously reproduce the OH and HO₂ concentrations when the amount of X changed from 0.1 to 0.25 ppb. The ROx primary production rate was dominated by photolysis reactions, in which the HONO, O₃, HCHO, and carbonyls photolysis accounted for 29%, 16%, 16%, and 11% during the daytime, respectively. The ROx termination rate was dominated by the reaction of OH + NO₂ in the morning, and thereafter the radical self-combination gradually became the major sink of ROx in the afternoon. As the sum of the respective oxidation rates of the pollutants via reactions with oxidants, the atmospheric oxidation capacity was evaluated, with a peak of 11.8 ppb h⁻¹ around noontime. The ratio of $P(O_3)_{net}$ to $AOC_{VOCs}$, which indicates the yield of net ozone production from VOCs oxidation, trended to increase and then decrease as the NO concentration increased. The median ratios ranged within 1.0-4.5, with the maximum existing when the NO

concentration was approximately 1 ppb. The nonlinear relationship between the yield of net ozone production from VOCs oxidation and NO concentrations demonstrated that optimizing the NOx and VOCs control strategies is critical to controlling ozone pollution effectively in the future.

## 1 Introduction

Severe ambient ozone ($O_3$) pollution is one of China's most significant environmental challenges (Shu et al., 2020;Li et al., 2019;Wang et al., 2020;Ma et al., 2019b;Wang et al., 2017a). Despite the reduction in emissions of $O_3$ precursors, $O_3$ concentration is increasing, especially in urban cities. The $O_3$ average trends for the focus megacity clusters are 3.1 ppb $a^{-1}$, 2.3 ppb $a^{-1}$, 0.56 ppb $a^{-1}$, and 1.6 ppb $a^{-1}$ for North China Plain (NCP), Yangtze River Delta (YRD), Pearl River Delta (PRD), and Szechwan Basin (SCB), respectively (Li et al., 2019). The nonlinearity between $O_3$ and precursors illustrates that exploring the cause of $O_3$ production is necessary. The tropospheric $O_3$ is only generated in the photolysis of nitrogen dioxide ($NO_2$) which is produced as the by-product within the radical cycling. Thus, the investigation of radical chemistry is critical to controlling secondary pollution.

Hydroxyl radicals (OH), the dominant oxidant, control the atmospheric oxidation capacity (AOC) in the troposphere. The OH radicals convert primary pollutants to secondary pollutants and are simultaneously transformed into peroxy radicals ($HO_2$ and $RO_2$). Within the interconvert of ROx (= OH, $HO_2$, and $RO_2$), secondary pollutants are generated, and thus the further exploration of radical chemistry is significant. The radical closure experiment, an effective indicator for testing our understanding of radical chemistry, has been conducted since the central role of OH radicals was recognized in the 1970s (Levy, 1971;Hofzumahaus et al., 2009). The underestimation of OH radicals in environments characterized by low nitrogen oxides (NO) and high volatile organic compounds (VOCs) has been identified (Lu et al., 2013;Lu et al., 2012;Tan et al., 2017;Tan et al., 2019;Yang et al., 2021;Hofzumahaus et al., 2009;Lelieveld et al., 2008;Whalley et al., 2011). New radical mechanisms involving unclassical OH regeneration have been proposed, mainly including Leuven Isoprene Mechanism (LIM) and X mechanism (Peeters and Muller, 2010;Peeters et al., 2014;Peeters et al., 2009;Hofzumahaus et al., 2009). The LIM which has been integrated into the current radical mechanism is still insufficient to explain the OH missing sources. The X mechanism was identified several times, but the amount of the numerical species, X, varied in different environments, and the nature of X is still unknown (Hofzumahaus et al., 2009;Lu et al., 2013;Lu et al., 2012;Tan et al., 2017;Tan et al., 2019;Yang et al., 2021;Ma et al., 2022a). Therefore, further exploration of radical regeneration sources is necessary.

Due to the strong photochemistry influenced by high temperatures and strong radiation, severe $O_3$ pollution appeared to occur in YRD and PRD, especially in PRD (Ma et al., 2019b;Wang et al., 2017a). Radicals, the dominant oxidant in the troposphere, have been measured during warm seasons in NCP (Yufa 2006, Wangdu 2014, and Beijing 2016), YRD (Taizhou 2018), SCB (Chengdu 2019), and PRD (Backgarden 2006, and Heshan 2014) in China (Lu et al., 2013;Lu et al., 2012;Tan et

al., 2017;Tan et al., 2019;Yang et al., 2021;Tan et al., 2021;Ma et al., 2022a). The radical observations in PRD, where the cities
are suffering from severe $O_3$ pollution, have not been conducted since 2014, and thus the oxidation capacity here has not been
clear in recent years. Therefore, we carried out a continuous comprehensive field campaign (STudy of the Ozone foRmation
Mechanism - STORM) involving radical observations in Shenzhen, one of the megacities in PRD, in the autumn of 2018.
Overall, the following will be reported in this study.
(1) The observed radical concentrations, and the comparison between the radical observations and simulations.
(2) The exploration of the unclassical OH regeneration sources based on the experimental budget.
(3) The sources and sinks of ROx radicals.
(4) The evaluation of the atmospheric oxidation capacity.
**2 Methodology**
**2.1 Measurement site and instrumentation**
The STORM campaign was conducted from September to October 2018 in Peking University Shenzhen Graduate School
(22.60 deg N, 113.97 deg E), in the west of Shenzhen, Guangdong province. As shown in Fig. 1, this site, which belongs to
the urban site, is located in the university town, and is surrounded by residential and commercial areas. The northwest of the
site is close to the Shenzhen Wildlife Park, and the northeast is close to the Xili Golf Club (Yu et al., 2020). The Tanglang
Mountain Park with active biogenic emissions is located about 1 km southeast of the site. Overall, this site has no significant
local pollution sources nearby, but can represent the urban pollution characteristics (Huang et al., 2012a;Huang et al.,
2012b;Gao et al., 2018).

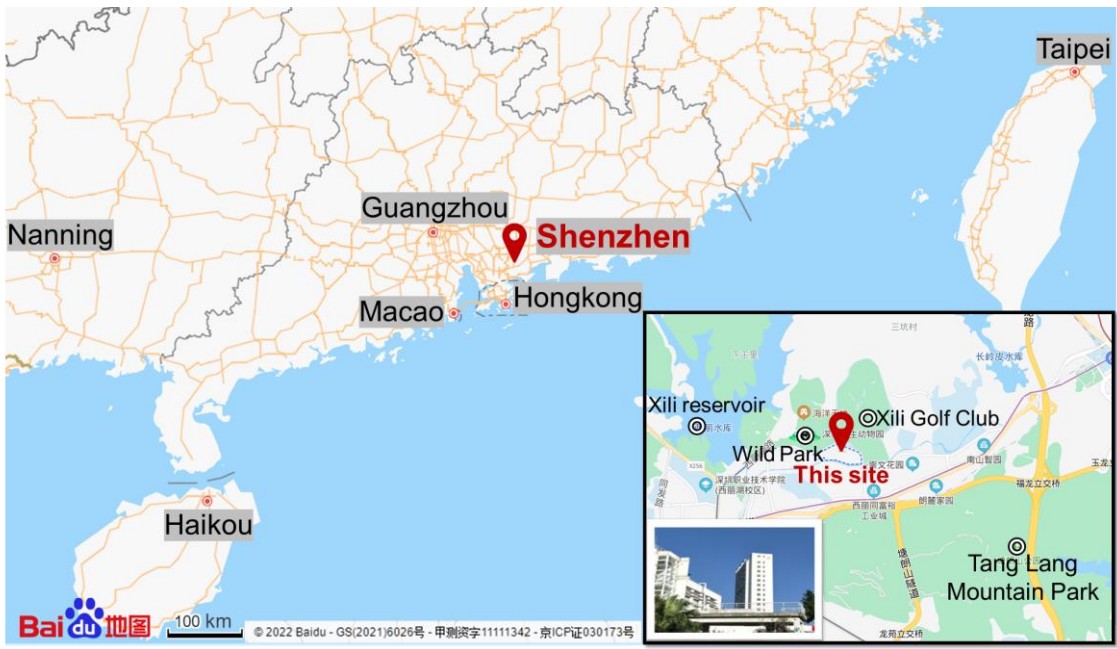

**Figure 1: Geographical location and surrounding environmental conditions of the measurement site in the STORM campaign (The**

 

Most instruments were set up on the top of a four-story academic building (about 20 m). Besides HOx radicals measured by
the Peking University-Laser Induced Fluorescence system (PKU-LIF) (see the details in Sect. 2.2), a comprehensive set of
trace gases was conducted to support the exploration of radical chemistry, including meteorological parameters (temperature,
pressure, relative humidity, *etc.*), photolysis frequency, OH reactivity ($k_{OH}$) and the trace gases (NO, NO$_2$, O$_3$, VOCs, *etc.*). $k_{OH}$
was measured by the Laser flash Photolysis-Laser Induced Fluorescence system (LP-LIF). Most of the inorganic trace gases
(O$_3$, CO, NO, NO$_2$, and SO$_2$) were simultaneously measured by two sets of instruments, and good agreement was achieved
within the uncertainty. VOCs species (alkanes, alkenes, aromatics, isoprene, and oxygenated VOCs (OVOCs)) were measured
using a gas chromatograph following a mass spectrometer (GC-MS). In addition, HONO and HCHO were measured as well.
Table S1 in the Supplementary Information presents the experimental details of the meteorological and chemical parameters
during this campaign.
**2.2 The OH and HO$_2$ measurements**
The OH and HO$_2$ radicals were measured by PKU-LIF based on the fluorescence assay by gas expansion (FAGE) technique.
The principle has been reported in previous studies, and only a brief description of the instrument is presented here. Further
detailed information on the instrument can be found in previous studies (Heard and Pilling, 2003;Fuchs et al., 2008;Holland
et al., 1995;Hofzumahaus et al., 1996;Fuchs et al., 2011).
In principle, OH resonance fluorescence is released in the OH excitation by a 308 nm pulsed laser, and then OH radicals are
detected directly. HO$_2$ radicals are converted into OH via NO, and then they are detected. The system contains a laser module
and a detection module. Ambient air was drawn into two independent, parallel, low-pressure (3.5 mBar) cells through two
parallel nozzles with 0.4 mm diameter pinhole. The OH radicals are excited into resonance fluorescence in the OH detection
cell and detected by micro-channel plate detectors (MCP). In the HO$_2$ detection cell, NO is injected and converts HO$_2$ to OH
radicals, and then OH radicals are excited by the laser and release resonance fluorescence. Besides, an OH reference cell in
which a large OH concentration is generated by pyrolysis of water vapor on a hot filament is applied to automatically correct
the laser wavelength.
Owing to the failure of the reference cell in this campaign, the NO mixing ratios injected into the HO$_2$ detection cell were
set to be higher than those in other campaigns in China because the HO$_2$ cell needed to be used as a reference cell to correct
laser wavelength. In this campaign, NO mixing ratios were switched between 25 ppm (low NO mode) and 50 ppm (high NO
mode). We calculated the HO$_2$-to-OH conversion efficiencies under the two different NO concentrations by calibrating the
PKU-LIF system. HO$_2$-to-OH conversion efficiencies in low NO mode ranged within 80%-95%, while those in high NO mode
reached 100%, demonstrating that the high NO concentration is sufficient to achieve the complete HO$_2$-to-OH conversion and
thus the HO$_2$ measurement was affected by RO$_2$ radicals. Prior studies have reported the relative detection sensitivities ($\alpha_{RO_2}$)
for the major $RO_2$ species, mainly from alkenes, isoprene and aromatics. Fuchs et al. (2011) reported that the relative $RO_2$
detection sensitivities are approximately constant when the NO concentration is so high that $HO_2$ conversion in the detection
is nearly complete. Thus, when the $HO_2$-to-OH conversion efficiencies reach 100%, the relative $RO_2$ detection sensitivities
reported by Fuchs et al. (2011) and Lu et al. (2012) can be used for the correction of $HO_2$ concentrations (Fuchs et al., 2011;Lu
et al., 2012;Lu et al., 2013). Herein, only the $HO_2$ observations in high NO mode were chosen and they were denoted as $[HO_2^*]$,
which was the sum of the true $HO_2$ concentration and a systematic bias from the mixture of $RO_2$ species $i$ which were detected
with different relative sensitivities $\alpha_{RO_2}^i$, as shown in Eq. (1) (Lu et al., 2012). The true $HO_2$ concentration was difficult to be
calculated because the observed concentrations of $RO_2$ and their speciation were not available. Herein, we simulated the $HO_2$
and $HO_2^*$ concentrations by the model, and the $RO_2$ interference yields which were used for correction were the modeled
values reported by Lu et al. (2012) in the PRIDE-PRD2006 campaign in which the $HO_2$-to-OH conversion efficiencies also
reached 100% due to the injection of pure NO in the $HO_2$ detection cell. The interference from $RO_2$ radicals was estimated to
be the difference between the modeled $HO_2$ and $HO_2^*$ concentrations. Overall, the measurement uncertainties of OH and $HO_2^*$
radicals were 11% and 15%, respectively, as shown in Table S1 in the Supplementary Information.
$$[HO_2^*] = [HO_2] + \sum(\alpha_{RO_2}^i \times [RO_2]_i) \tag{1}$$
Additionally, prior studies reported that OH measurement might be affected by the potential interference, when the sampled
air contains ozone, alkenes and BVOCs (Mao et al., 2012;Fuchs et al., 2016;Novelli et al., 2014), indicating the environmental
conditions are important to the production of interference. The pre-injector is usually used to test the potential OH interference,
and has been applied to our PKU-LIF system to quantify the possible interferences for several campaigns, including the
campaigns conducted at the Wangdu, Heshan, Huairou, Taizhou and Chengdu sites (Tan et al., 2017;Tan et al., 2019;Tan et al.,
2018;Yang et al., 2021;Ma et al., 2022b). No significant internal interference was found in the prior studies, demonstrating the
accuracy of the PKU-LIF system has been determined for several times. Moreover, to further explore the potential interference
in this campaign, we compared the major environmental conditions, especially $O_3$, alkenes and isoprene, between Shenzhen
and Wangdu sites, as shown in the Supplementary Information. The results indicated that the environmental condition in
Shenzhen was less conducive to generating interference than that in Wangdu, and the details were presented in the
Supplementary Information. Besides the environmental conditions, the prior studies reported that the product of the reaction
of $RO_2$ with OH, trioxides (ROOOH), might lead to an OH interference signal. The reactions of $RO_2$ radicals with OH radicals
might be competitive with other sinks for $RO_2$ radicals (Fittschen, 2019;Fittschen et al., 2019;Berndt et al., 2022). Fittschen et
al. (2019) reported that the OH interference signals might come from the ROOOH heterogeneous decomposition on the walls
of the FAGE cell or the entrance nozzle, but they also noted that the ROOOH interference is highly dependent on the design
and measurement conditions of different FAGE instruments. Therefore, we integrated the reactions of the ROOOH production
and destruction into the base model herein, with the ROOOH production rate constant of $1.5 \times 10^{-10}$ $cm^3$ $s^{-1}$ and the destruction
rate constant of $10^{-4}$ s$^{-1}$ (the details are presented in the Supplementary Information) (Fittschen et al., 2019). Figure. S1 (a)
presents the modeled ROOOH concentrations during this campaign, with a maximum of about $4.4\times10^9$ cm$^{-3}$. The correlation
of the modeled ROOOH concentrations and the ratios of OH observations to OH simulations, and the correlation of the
modeled ROOOH concentrations and the difference between OH observations and simulations both demonstrated that no
significant relevance between ROOOH and the underestimation of OH radicals, as shown in Fig. S1 (b-c). Additionally, the
ROOOH values modeled in our another campaign (Taizhou, 2018) were comparable to or even slightly higher than the
simulations in this study, and the chemical modulation tests in Taizhou confirmed the ROOOH is not a significant OH
interference in our PKU-LIF system (Ma et al., 2022b). Overall, the OH interference during this campaign was negligible
according to the analysis of the behavior of PKU-LIF system in previous campaigns, the comparison of environmental
conditions between this campaign and Wangdu campaign, and the exploration of the impact of ROOOH on the discrepancy of
OH observations and simulations. However, we should acknowledge that the unmeasured interference might have an effect on
radical measurement. More precise chemical modulation tests are needed in the future.

**2.3 Closure experiment**
As an effective tool to explore the atmospheric radical chemistry, the radical closure experiment can investigate the state-of-
the-art chemical mechanism because of the extremely short lifetime of radicals (Stone et al., 2012;Lu et al., 2019). A zero-
dimensional box model was used to conduct the radical closure experiment, and the overall framework was reported by Lu et
al. (2019). In this work, we conducted the radical closure experiment based on the Regional Atmospheric Chemical Mechanism
updated with the latest isoprene chemistry (RACM2-LIM1), as Tan et al. (2017) described in detail. The model was constrained
by the measured meteorological, photolysis frequency, and the critical chemical parameters (CO, NO, NO$_2$, VOCs, *etc.*). The
H$_2$ and CH$_4$ mixing ratios were set to 550 ppb and 1900 ppb, respectively. The model was operated in time-dependent mode
with a 5-min time resolution, and a 2-d spin-up time was to make the unconstrained species approach the steady state relative
to the constrained species.
As Lu et al. (2012) described, there are two types of radical closure experiment. One is the comparison of observed and
modeled radical concentrations, and the other is the comparison of radical production and destruction rates. The most
significant difference between the above is that the latter is conducted with the observed radical concentrations and $k_{OH}$
constrained. The comparison of radical production and destruction rates, which is also called radical experimental budget, can
test the accuracy of the state-of-the-art chemistry mechanisms based on the equivalent relationship between the radical
production and destruction rates. The production rates of OH, HO$_2$, and RO$_2$ radicals are quantified from all the known sources.
The destruction rates of HO$_2$ and RO$_2$ radicals are the sum of the known sinks, while the OH destruction rate can be directly
calculated as the product of the observed OH concentrations and the observed $k_{OH}$ (Tan et al., 2019;Yang et al., 2021). The OH
destruction rate is the total sinks of OH radicals because of the direct $k_{OH}$ observation, and thus the discrepancy between the
OH destruction and production rates denotes the missing OH sources. The detailed reactions and the reaction rate constants
related to OH, $HO_2$, and $RO_2$ radicals can be found in Tan et al. (2019) and Yang et al. (2021).
**2.4 AOC evaluation**
The life time of the trace gases is controlled not only by the oxidant concentration but also by its second-order rate constant,
so the atmospheric oxidation capacity (AOC) proposed by Geyer et al. (2001) is most suitable to evaluate the relative
importance of each oxidant (Elshorbany et al., 2009). AOC is the core driving force of complex air pollution, and determines
the removal rate of trace gases and the production rates of secondary pollutants (Liu et al., 2021). As an effective indicator for
atmospheric oxidation intensity, the evaluation of AOC can provide crucial information on the atmospheric composition of
harmful and climate forcing species (Elshorbany et al., 2009). AOC is defined as the sum of the respective oxidation rates of
the pollutants via reactions with oxidants (Elshorbany et al., 2009;Geyer et al., 2001;Zhu et al., 2020). According to the
definition of AOC, it can be calculated by the Eq. (2).
$$AOC = \sum_i k_{Y_i}[Y_i][X] \tag{2}$$
where $Y_i$ are the pollutants (CO, $CH_4$, and VOCs), $X$ are the main atmospheric oxidants (OH, $O_3$, $NO_3$), and $k_{Y_i}$ is the bi-
molecular rate constant for the reaction of $Y_i$ with $X$. AOC includes all combination of pollutants $Y$ and oxidants $X$. The
higher AOC, the higher removal rate of the atmospheric pollutants, and thus the higher production rate of secondary pollutants
(Yang et al., 2020b). Simultaneous measurements of OH and the key trace gases are available in the study. $NO_3$ concentration
could be simulated by the box model with the observed parameters constrained.
**3. Results**
**3.1 Meteorological and chemical conditions**
Figure 2 gives an overview of the meteorological and chemical parameters from 05 October to 28 October 2018, when OH
and $HO_2$ radicals were measured. The diurnal variations of the temperature (T), relative humidity (RH), $j(O^1D)$, and $j(NO_2)$
followed a regular pattern from day to day. The overall meteorological conditions were characterized by high temperature
(about 20~30 °C), high relative humidity (60~80%), and intensive radiation with $j(O^1D)$ up to $2.0 \times 10^{-5}$ $s^{-1}$ and $j(NO_2)$ up to
$6.0 \times 10^{-3}$ $s^{-1}$. The relative humidity and photolysis-frequency in this autumn campaign were similar to those in the summer
campaign conducted in Chengdu (Yang et al., 2021). The temperature in this campaign was lower than that in Chengdu, but
similar to that in the autumn campaign in Heshan located in PRD as well (Tan et al., 2019;Yang et al., 2021).

205        The concentration of CO showed a weak diurnal variation, indicating there was the non-obvious accumulation of

anthropogenic emissions on a regional scale. NO concentration peaked at 12 ppb during morning rush hour when the traffic
emission was severe, and thereafter, $O_3$ concentration started to increase with the decreasing of NO concentration. The maxima
of $O_3$ hourly concentration were high up to 120 ppb. According to the updated National Ambient Air Quality Standard of China
(GB3095-2012), $O_3$ concentration exceeded the Class-II limit values (hourly averaged limit 93 ppb) on several days (6, 7, 8,
and 26 October) when the environmental condition was characterized by high temperature and low relative humidity. $NO_2$
concentration was high at night because of the titration effect of $O_3$ with NO.
Along with the high $O_3$ concentration on 6, 7, 8, and 26 October, high HCHO concentration was also recorded during the
corresponding periods, indicating HCHO was mainly produced as secondary pollutants because of the active photochemistry
in this campaign. Isoprene, which is mostly derived from biogenic emissions and mainly affected by temperature, peaked
around noontime. Tan et al. (2019) reported the median concentration of HCHO and isoprene concentrations were 6.8 ppb and
0.6 ppb during 12:00-18:00 at Heshan site. Similarly, the median concentration of HCHO and isoprene concentrations in this
study were 4.9 ppb and 0.4 ppb during the corresponding periods, respectively. As a proxy for traffic intensity, the toluene to
benzene ratio (T/B), which is below 2, means the traffic emissions are the major sources of VOCs (Brocco et al., 1997). In this
campaign, the T/B gradually dropped from 07:00 until it reached the minimum value at 09:00, indicating traffic emission
contributed more to VOCs during morning rush hour than during other periods. However, the T/B values, which varied within
a range of 7-12, were above 2, and thus VOCs emission during this campaign was mainly from other sectors such as those
involving solvent evaporation.

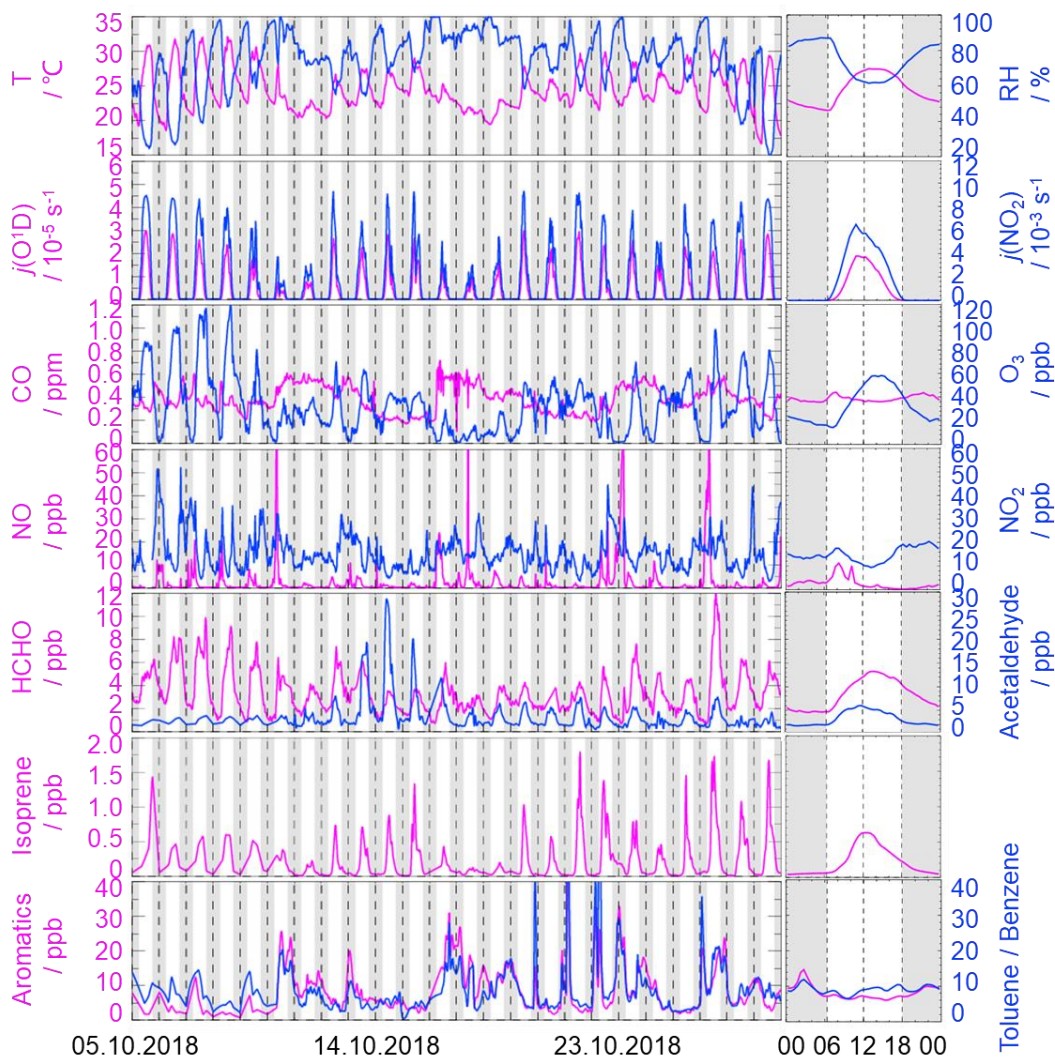


**Figure 2: Timeseries and diurnal profiles of the observed meteorological and chemical parameters in the STORM campaign. The grey areas denote nighttime.**

Moreover, we compared the environmental conditions between the Backgarden (rural site), Heshan (suburban site), and Shenzhen (urban site) campaigns conducted in PRD in Table S3 in the Supplementary Information. No significant discrepancy in temperature was found in the Shenzhen and Heshan campaigns, which were both conducted in autumn. The temperature in the Backgarden campaign conducted in summer was higher than those in Shenzhen and Heshan. The relative humidity in Shenzhen and Backgarden was higher than that in Heshan. Compared to the chemical conditions in the Heshan campaign conducted in autumn as well, the concentrations of CO, NO, $NO_2$, HONO, alkenes, aromatics, and HCHO in Shenzhen were lower, which might be because there were no significant local pollution sources nearby at the Shenzhen site although it was an urban site. However, the concentration of $O_3$ which is the typical secondary pollutant in Shenzhen was higher than that in Heshan. Compared to the environmental conditions in Heshan, the higher $O_3$ concentration in Shenzhen might benefit from the weather condition which was characterized by the stronger solar radiation and slightly higher temperatures.

## 3.2 Observed and modeled OH and HO₂ radicals

The OH and $HO_2$ radicals were measured during 05-28 October 2018. The timeseries of the observed and modeled HOx concentrations are displayed in Fig. S2 (a-b) in the Supplementary Information. Data gaps were caused by the rain, calibration, and maintenance. The daily maxima of the observed OH and $HO_2^*$ concentrations varied in the range of $(2-9) \times 10^6$ cm$^{-3}$ and $(2-14) \times 10^8$ cm$^{-3}$, respectively. As in previous campaigns, the largest OH concentrations appeared around noontime and showed a high correlation with $j(O^1D)$, a proxy for the solar UV radiation driving much of the primary radical production (Tan et al., 2019).

Figure 3 (a-b) shows the diurnal profiles of the observed and modeled HOx concentrations. The HOx radicals showed similar diurnal behavior to those reported in other campaigns (Ma et al., 2019a;Tan et al., 2017;Tan et al., 2019;Tan et al., 2018;Yang et al., 2021). The observed OH and $HO_2^*$ concentrations reached a maximum around 12:00 and 13:30, respectively. The diurnal maxima of the observed and modeled OH concentrations were $4.5 \times 10^6$ cm$^{-3}$ and $3.5 \times 10^6$ cm$^{-3}$. Compared to the other campaigns conducted in PRD (Backgarden and Heshan), the diurnal maximum of the observed OH concentration in Shenzhen was equal to that observed in Heshan, but much lower than that observed in Backgarden where the observed OH concentration was nearly $15 \times 10^6$ cm$^{-3}$ (Hofzumahaus et al., 2009;Tan et al., 2019). The higher OH concentration at Backgarden site was closely correlated to the stronger solar radiation, as shown in Table S3 in the Supplementary Information. The diurnal observed and modeled OH concentrations agreed within their 1-σ uncertainties of measurement and simulation (11% and 40%). However, when the NO mixing ratio (Fig. 2) dropped from 10:00 gradually, a systematic difference existed, with the observed OH concentration being about $1 \times 10^6$ cm$^{-3}$ higher than the modeled OH concentration. The OH concentrations observed in the environments with low NO levels were underestimated by the state-of-the-art models at Backgarden (summer) and Heshan (autumn) sites in PRD as well, and the OH underestimation was identified to be universal at low NO conditions in China (Lu et al., 2013;Lu et al., 2012;Ma et al., 2019a;Tan et al., 2017;Yang et al., 2021;Ma et al., 2022b). The reason on OH underestimation was further discussed in Section 4.1.

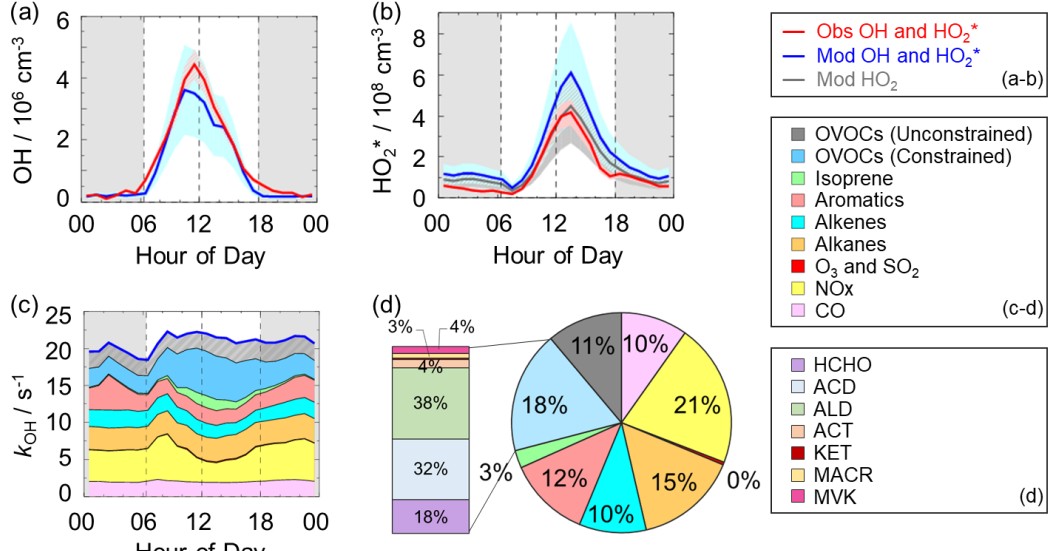


**Figure 3: (a-b) The diurnal profiles of the observed and modeled OH, $HO_2^*$ and $HO_2$ concentrations. (c) The diurnal profiles of the**
**modeled $k_{OH}$. (d) The composition of the modeled $k_{OH}$. The red areas in (a-b) denote 1-σ uncertainties of the observed OH and $HO_2^*$**
**concentrations. The blue areas in (a-b) denote 1-σ uncertainties of the modeled OH and $HO_2^*$ concentrations, and the dark grey**
**area in (b) denotes 1-σ uncertainties of the modeled $HO_2$ concentrations. The grey areas in (a-c) denote nighttime. ACD denotes**
**acetaldehydes. ALD denotes the C3 and higher aldehydes. ACT and KET denote acetone and ketones. MACR and MVK denote**
**methacrolein and methyl vinyl ketone.**
The diurnal maximum of the observed $HO_2^*$, the modeled $HO_2^*$ and the modeled $HO_2$ concentrations were $4.2 \times 10^8$ cm$^{-3}$,
$6.1 \times 10^8$ cm$^{-3}$, and $4.4 \times 10^8$ cm$^{-3}$, respectively. The difference between the modeled $HO_2^*$ and $HO_2$ concentrations can be
considered a modeled $HO_2$ interference from $RO_2$ (Lu et al., 2012). The $RO_2$ interference was small in the morning, while it
became larger in the afternoon. It ranged within 23%-28% during the daytime (08:00-17:00), which was comparable with
those at the Backgarden and Yufa sites in China, Borneo rainforest in Malaysia (OP3 campaign, aircraft), and UK (RONOCO
campaign, aircraft) (Lu et al., 2012;Lu et al., 2013;Jones et al., 2011;Stone et al., 2014). The observed $HO_2^*$ was overestimated
by the model, indicating the $HO_2$ heterogeneous uptake might have a significant impact during this campaign. The diurnal
maximum of $HO_2^*$ concentration observed in Shenzhen was much lower than those observed at the Yufa and Backgarden sites
(Hofzumahaus et al., 2009;Lu et al., 2012;Lu et al., 2013). The high modeled $HO_2$/OH ratio around noontime (11:00-15:00),
which was about 138, was found in this campaign, which was higher than those at the Backgarden and Chengdu sites (Yang
et al., 2021;Hofzumahaus et al., 2009). High $HO_2$/OH ratio is normally found only in clean air at low NOx (= NO + $NO_2$)
concentrations (Hofzumahaus et al., 2009;Stevens et al., 1997). As an indicator that can reflect the interconversion reaction
between $HO_2$ and OH, the conversion efficiency in this campaign was slightly slower than those at the Backgarden and
Chengdu sites.
**3.3 OH reactivity**
$k_{OH}$ is the pseudo-first-order loss rate coefficient of OH radicals, and it is equivalent to the reciprocal OH lifetime (Fuchs et

al., 2017;Lou et al., 2010;Yang et al., 2019). In this campaign, $k_{OH}$ was measured only for several days (05-19 October 2018) by the LIP-LIF system, which has been reported in the previous study (Liu et al., 2019). The timeseries of the observed and modeled $k_{OH}$ during 05-19 October 2018 are presented in Fig. S3 in the Supplementary Information. A good agreement between the observed $k_{OH}$ and modeled $k_{OH}$ within the uncertainties was achieved, and thus the model can be believed to reproduce the observed $k_{OH}$ values within the whole campaign. Moreover, to reflect the $k_{OH}$ in the whole campaign, the modeled values were shown in the $k_{OH}$ diurnal profiles (Fig. 3 (c)) and $k_{OH}$ timeseries (Fig. S2 (c)) during 05-28 October 2018. The modeled $k_{OH}$ showed a weak diurnal variation and varied from 18 s$^{-1}$ to 22 s$^{-1}$. Compared to the $k_{OH}$ variation in Shenzhen, the $k_{OH}$ observed at the Backgarden and Heshan sites in PRD showed a stronger diurnal variation, with a minimum value at around noontime and a maximum value at daybreak. Additionally, the $k_{OH}$ values in this campaign were lower than those at Backgarden (20-50 s$^{-1}$) and Heshan (22-32 s$^{-1}$) sites (Lou et al., 2010;Tan et al., 2019). Similar with the good agreement between the observed and modeled $k_{OH}$ during the several days in Shenzhen, the observed $k_{OH}$ in Backgarden was matched well with the modeled $k_{OH}$ which has included the OVOCs reactivity. In terms of the $k_{OH}$ in Heshan, Tan et al. (2019) reported that only half of the observed $k_{OH}$ was explained by the calculated $k_{OH}$ which was calculated from the measured trace gas concentrations. The missing $k_{OH}$ in Heshan was likely caused by unmeasured VOCs, demonstrating the necessary to measure more abundant VOCs species, especially OVOCs species.

As shown in Fig. 3 (d), we presented the composition of modeled $k_{OH}$. The inorganic compounds contributed approximately 31% to $k_{OH}$, in which the CO and NOx reactivity accounted for 10% and 21%, respectively. The NOx reactivity was displayed versus time, with a maximum during the morning peak. The peak concentration during the morning peak was associated with traffic emissions.

Compared with the inorganic reactivity, the larger fraction of $k_{OH}$ came from the VOCs group, with a contribution of 69% to $k_{OH}$. The contribution of alkanes, alkenes, and aromatics were 15%, 10%, and 12%, respectively. The isoprene reactivity related to temperature was mainly concentrated during the daytime, whereas the aromatics reactivity at night was higher. As for the OVOCs species, we measured several OVOCs species, including HCHO, acetaldehydes (ACD) and higher aldehydes (ALD), acetone (ACT), ketones (KET) and isoprene oxidation products (methacrolein (MACR) and methyl vinyl ketone (MVK)), and thus we constrained these species in the model. The constrained OVOCs species accounted for 18% in the total $k_{OH}$, where HCHO, ACD, and ALD were the major contributors, with contributions of 18%, 32%, and 38% to the constrained OVOCs, respectively. The contribution of aldehydes in this study (16%) was larger than that in Beijing (Whalley et al., 2021) and smaller with that in Wangdu (Fuchs et al., 2017). The remaining reactivity was attributed to the unconstrained OVOCs reactivity, which came from the model-generated intermediate species (glyoxal, methylglyoxal, methyl ethyl ketone, methanol, etc.), with a contribution of 11% to the total $k_{OH}$. Large fraction of OVOCs reactivities in $k_{OH}$ was also found in some previous studies (Lou et al., 2010;Lu et al., 2013;Fuchs et al., 2017;Whalley et al., 2021). About 50% of $k_{OH}$ was explained by OVOCs at Backgarden site, and HCHO, ACD and ALD, and oxygenated isoprene products were the most important OH reactants in

OVOCs, with a contribution of 30-40%, and other 10-20% came from other oxygenated compounds (ketones, dicarbonyl compounds, alcohols, hydroperoxides, nitrates etc.) (Lou et al., 2010). HCHO, ACD, MVK, MVCR and glyoxal accounted for one-third of the total $k_{OH}$ at Wangdu site (Fuchs et al., 2017). The large unconstrained OVOCs reactivity indicated it is necessary to measure more VOCs species in the future.

**4. Discussion**

**4.1 Radical closure experiment**

In this study, we conducted OH radical closure experiment which is called OH experimental budget as well. As discussed in Section 3.3, it is believed that the model can reproduce the observed $k_{OH}$. Herein, to conduct the OH experiment budget in the whole campaign, we used the modeled $k_{OH}$ to calculate the OH destruction rate because the $k_{OH}$ was only measured on several days. The diurnal profiles of OH production and destruction rates, and the compositions of OH production rate were displayed in Fig. 4, with maxima of 14 ppb h$^{-1}$ and 17 ppb h$^{-1}$ around noontime, respectively. The OH production rate from known sources is quantified from the primary sources (photolysis of HONO, photolysis of $O_3$, ozonolysis of alkenes) and secondary sources (dominated by $HO_2$ + NO, and $HO_2$ + $O_3$). The primary and secondary sources accounted for 78% and 22% of the total calculated production rate, respectively. Similar with the prior studies, the largest fraction of OH production rate came from $HO_2$ + NO, with a contribution up to 76% of the known OH production rate. As the major primary OH sources, the HONO and $O_3$ photolysis contributed 13% and 7% to the total calculated OH production rate, respectively.

The OH production rate matched well with the destruction rate only in the early morning to about 10:00. Thereafter, the OH destruction rate was larger than the production rate, which could explain the underestimation of OH concentration by the model. As shown in Fig. 4 (b), the discrepancy between the OH production and destruction rates at around 11:00-15:00, which was approximately of (3.1~4.6) ppb h$^{-1}$, cannot be explained by the combined experimental uncertainties. The discrepancy was attributed to the missing OH sources because $k_{OH}$ was constrained in this study. The biggest additional OH source was approximately 4.6 ppb h$^{-1}$, which occurred at about 12:00, when the OH production and destruction rates were 11.9 ppb h$^{-1}$ and 16.5 ppb h$^{-1}$, respectively. The unknown OH source accounted for about one third of the total OH production rate, indicating the exploration of missing OH source was significant to study the radical chemistry. It is noted that the OH production rate was overestimated because we used $HO_2^*$ concentrations instead of $HO_2$ concentrations here. Thus, the missing OH source was the lower limit here, demonstrating more unknown OH sources need to be further explored. Details on unknown OH sources are given below (Sect. 4.2).

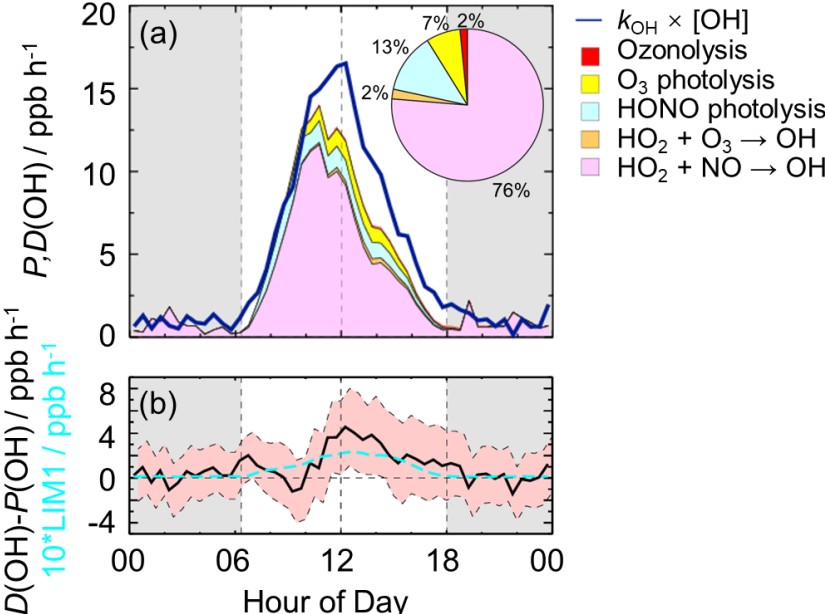

**Figure 4: (a) The diurnal profiles of OH production and destruction rates and the proportions of different known sources in the calculated production rate during the daytime. The blue line denotes the OH destruction rate, and the colored areas denote the calculated OH production rates from the known sources. (b) The missing OH source which was the discrepancy between the OH destruction and production rates, and the OH production rate which was ten times the production rate derived from LIM1 mechanism. The red shaded areas denote the combined uncertainty from the experimental errors of the measured quantities (Table S1) and the reaction rate coefficients. The grey areas denote nighttime.**

## 4.2 Radical chemistry in low NO regime

### 4.2.1 Influencing factors of OH underestimation

As analyzed in Sect. 4.1, the underestimation of OH concentration was attributable to the missing OH source. It is necessary to explore the influencing factor for gaining further insight into the missing source. Scientists reported that more significant OH underestimation would appear with the decreasing NO concentration and increasing isoprene concentration (Lu et al., 2012;Ren et al., 2008;Hofzumahaus et al., 2009;Lelieveld et al., 2008;Whalley et al., 2011;Tan et al., 2017;Yang et al., 2021). Herein, we further explored the effect of NO concentration on missing OH source. The NO dependence on observed and modeled HOx concentrations and the NO dependence on HOx observed-to-modeled ratios were illustrated in Fig. 5 and Fig. S4. The OH concentrations were normalized by the averaged $j(O^1D)$ to eliminate the influence of radiation on radicals. The OH concentration showed an increasing trend with the increase of NO concentrations in low NO regime (below 1 ppb) due to the increased OH radicals from propagation via peroxy reactions with NO, and then decreased with the increase of NO concentrations in high NO regime (above 1 ppb) due to the OH loss by the reactions via $NO_2$ (Ehhalt, 1999). The base model can reproduce the observed OH concentration in high NO regime, while underestimate OH concentration in low NO regime. As for $HO_2^*$ radicals, the observed and modeled $HO_2^*$ concentrations decreased with the increase of NO concentrations. The

364  model overestimated the observations, indicating that the heterogeneous uptake might make a significant role in $HO_2$ sinks in

365  this campaign. Overall, NOx plays a crucial role in radical chemistry due to their impact on radical propagation and termination

366  reactions.

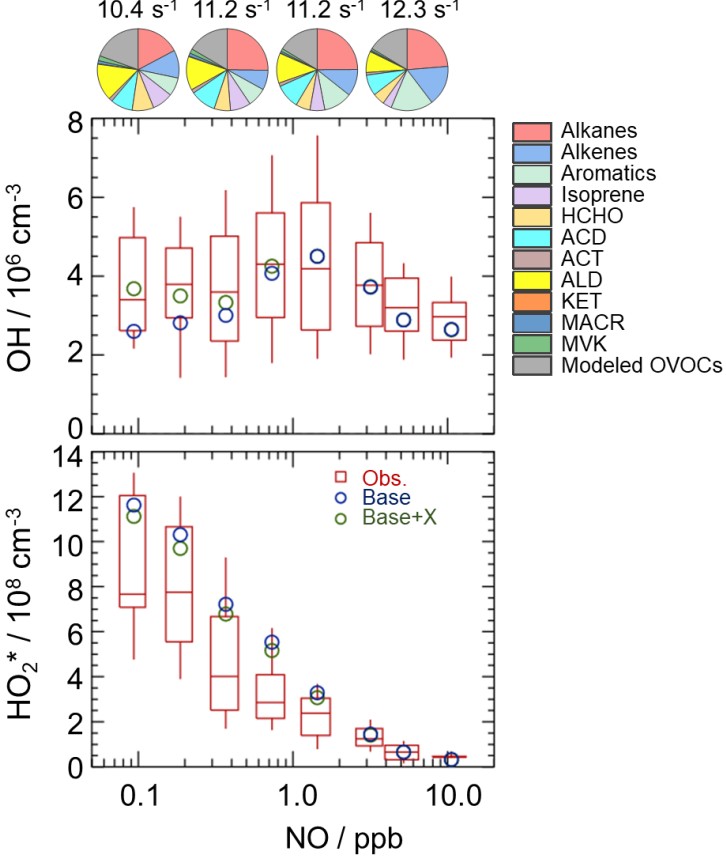

367

**Figure 5: NO dependence on OH and $HO_2^*$ radicals. The red box-whisker plots give the 10%, 25%, median, 75%, and 90% of the HOx observations. The blue circles show the median values of the HOx simulations by the base model, and the green circles show the HOx simulations by the model with X mechanism. Total VOCs reactivity and their organic speciation are presented by pie charts at the different NO intervals at the top. Only daytime values and NO concentration above the detection limit of the instrument were chosen. ACD and ACT denote acetaldehyde and acetone, respectively. ALD denotes the C3 and higher aldehydes. KET denotes ketones. MACR and MVK, which are both the isoprene oxidation products, denote methacrolein and methyl vinyl ketone, respectively.**

To further explore the influencing factors of OH underestimation, we presented the speciation VOCs reactivity under the
different NO intervals, as shown in Fig. 5 and Table S4 in the Supplementary Information. The isoprene reactivity and total
OVOCs reactivity (the sum of HCHO, ACD, ACT, ALD, KET, MACR, MVK and the modeled OVOCs) increased with the
decrease of NO concentrations, while the anthropogenic VOCs reactivity (alkanes, alkenes and aromatics) was higher in high
NO regime. Additionally, the $O_3$ concentration in low NO regime was significantly higher than those in high NO regime, and
the temperature was slightly higher in low NO regime, demonstrating the photochemistry was more active in low NO regime
in this campaign. Overall, the photochemistry and composition of VOCs reactivity, especially the isoprene and OVOCs species
(mainly HCHO, ACD, ALD and the modeled OVOCs), might closely impact the missing OH sources.

**4.2.2 Quantification of missing OH sources**

Hofzumahaus et al. (2009) proposed an existence of a pathway for the regeneration of OH independent of NO, including the conversions of $RO_2 \rightarrow HO_2$ and $HO_2 \rightarrow OH$ by a numerical species called X. With a retrospective analysis, the unclassical OH recycling pathway was identified to be universal at low NO conditions in China. The amount of X varies with environmental conditions, and the X concentrations were 0.85 ppb, 0.4 ppb, 0.1 ppb, 0.4 ppb, 0.1 and 0.25 ppb at Backgarden, Yufa, Wangdu, Heshan, Taizhou, and Chengdu sites (Hofzumahaus et al., 2009;Lu et al., 2012;Lu et al., 2013;Tan et al., 2017;Yang et al., 2021;Ma et al., 2022b).

In this study, we tested this unclassical X mechanism. Good agreement between observations and simulations of OH radicals was achieved when a constant mixing ratio of 0.1 ppb of X was added into the base model. As shown in Fig. 5, the model with X mechanism agreed with the observed OH concentrations even at low NO conditions. Unclassical OH recycling was identified again in this study. Nevertheless, X is an artificial species that behaves like NO, and thus the nature of X is still unknown to us. Compared to the Shenzhen site, the required X concentration at the Backgarden and Heshan sites in PRD was higher, which might be affected by the different air masses in the three sites. The $k_{OH}$ at Shenzhen site was much lower than those at the Backgarden and Heshan sites (Lu et al., 2013), and a weaker diurnal variation of $k_{OH}$ in Shenzhen was observed. Under the influence of the East Asian monsoon, the prevailing wind for PRD area is mostly southerly during the summer months and mostly northerly during the winter months (Fan et al., 2005;Zhang et al., 2008). The Backgarden site is located in Guangzhou, and the Heshan site is located in Jiangmen. The two cities are along the north-south axis, and thus the air masses of the Backgarden and Heshan sites are intimately linked with each other, while the air mass in Shenzhen is more similar to Hongkong (Zhang et al., 2008). Compared to the VOCs reactivity in the air mass at Backgarden and Yufa sites reported by Lu et al. (2013), lower isoprene reactivity and OVOCs reactivity were observed in Shenzhen site. As discussed in Section 4.2.1, the OH underestimation might be closely related to the composition of VOCs reactivity. Therefore, further exploration of this unclassical OH recycling is needed to improve our understanding of radical chemistry, especially the mechanisms related to isoprene and OVOCs.

As for the potential influence of isoprene and OVOCs on the missing OH source, $RO_2$ isomerization reactions have also been shown to be of importance for the atmospheric fate of $RO_2$ from isoprene (Peeters et al., 2009;Peeters et al., 2014). The latest isoprene isomerization mechanism, which is called LIM1, has been coupled into our current base model. However, LIM1 mechanism was not included in the OH experimental budget which was conducted with the observations constrained, as shown in Section 4.1. Herein, we evaluated the contribution of LIM1 mechanism to the missing OH sources, as shown in Fig. 4 (b). LIM1 mechanism can explain approximately 7% of the missing OH sources during 10:00-16:00, when the missing OH production rate and the OH production rate derived from LIM1 mechanism were 2.47 ppb h$^{-1}$ and 0.17 ppb h$^{-1}$, respectively.

Additionally, prior studies also reported that OH regeneration might be achieved from the oxidation of MACR and MVK,

which are the major first-generated products of isoprene (Fuchs et al., 2018;Fuchs et al., 2014). As a potential explanation for
the high OH concentration, the impacts of MACR and MVK oxidation were evaluated here. The modification of MACR
oxidation scheme added the H-migration reactions of MACR oxidation products (Fuchs et al., 2014). The modification of
MVK oxidation scheme added the reactions of MVK oxidation products with $HO_2$ radicals and the H-migration reactions of
MVK oxidation products (Fuchs et al., 2018). As presented in Fig. S5 in the Supplementary Information, no significant of the
MACR and MVK oxidation schemes was found in this campaign.
Overall, a large part of missing OH sources was not explained by the isoprene chemistry. In the future, the impact of OVOCs
species which was another potential OH source on missing OH sources need to be further evaluated.

**4.3 $HO_2$ heterogeneous uptake**

The $HO_2^*$ overestimation was identified by comparing the observed and modeled $HO_2^*$ concentrations in Sect. 3.2 and Sect.
4.2.1. The $HO_2$ heterogeneous uptake has been proposed to be a potential sink of $HO_2$ radicals, and thus could influence the
radical chemistry and the formation of secondary pollution, especially in high-aerosol environments (Song et al., 2021;Song
et al., 2022;Tan et al., 2020;Kanaya et al., 2000;Kanaya et al., 2007;Li et al., 2019). The impact of $HO_2$ uptake chemistry on
radical concentration is different under different environmental conditions (Whalley et al., 2015;Mao et al., 2010;Li et al.,
2019). To evaluate the contribution of $HO_2$ uptake chemistry to radical concentrations in this study, we coupled $HO_2$
heterogeneous uptake into the base model (RACM2-LIM1) and conducted three sensitivity experiments, as shown in R1 and
Eq. (3).
$HO_2$ + aerosol $\rightarrow$ products                                                             R1
$$k_{HO_2+\text{aerosol}} = \frac{\gamma * \text{ASA} * v_{HO_2}}{4} \tag{3}$$
where ASA [$\mu m^2$ $cm^{-3}$], which represents the aerosol surface area concentration, can be estimated by multiplying the mass
concentration of $PM_{2.5}$ [$\mu g$ $m^{-3}$] by 20 here because there were no direct ASA observations in this campaign (Chen et al.,
2019;Wang et al., 2017b). $v_{HO_2}$, which can be calculated by Eq. (4), refers to the mean molecular velocity of $HO_2$ with a unit
of cm $s^{-1}$.
$$v_{HO_2} = \sqrt{\frac{8 * R * T}{0.033 * \Pi}} \tag{4}$$
where T [K] and R [J $mol^{-1}$ $K^{-1}$] denote the ambient temperature and gas constant. γ, the $HO_2$ effective uptake coefficient,
parameterizes the influence of some processes (Tan et al., 2020). γ varies in the highly uncertain range of 0-1 (Song et al.,
2022), and is the most critical parameter to impact $HO_2$ uptake chemistry. Only several observations of γ have been reported
(Taketani et al., 2012;Zhou et al., 2021;Zhou et al., 2020). The measured γ at the Mt. Tai site and Mt. Mang site were 0.13-
0.34 and 0.09-0.40, respectively (Taketani et al., 2012). The average value of the measured γ was 0.24 in Kyoto, Japan in the
summer of 2018 (Zhou et al., 2020). Zhou et al. (2021) reported the lower-limit values for median and average values of the
measured $\gamma$ were 0.19 and 0.23±0.21 in Yokohama, Japan in the summer of 2019. Additionally, Li et al. (2018) set 0.2 as the
value of $\gamma$ in the model, and Tan et al. (2020) calculated the $\gamma$ of 0.08±0.13 by the analysis of the measured radical budget in
Wangdu.
Here, we applied the two $\gamma$ (0.2 and 0.08), which have been used in the model, to evaluate the impact of $HO_2$ uptake on
radical concentrations, as shown in Fig. 6. The modeled $HO_2^*$ cannot match well with the observations when $\gamma$ of 0.08 and 0.2
was set in the model. As the $\gamma$ increased to approximately 0.3, good agreement between the modeled and observed $HO_2^*$
concentration was achieved, demonstrating that a significant heterogeneous uptake might exist in this campaign. It should be
noted that the $HO_2$ heterogeneous uptake ($\gamma$ = 0.3) reduced the modeled OH concentrations by around 20% compared to the
OH simulations in the base model during the daytime (08:00-18:00). Sensitivity tests illustrated that good agreements of OH
observations-simulations and $HO_2^*$ observations-simulations were both achieved when the amount of X changed from 0.1 ppb
to 0.25 ppb and the $HO_2$ effective uptake coefficient was 0.3, as shown in Fig. S6 in the Supplementary Information. Compared
to the Backgarden and Heshan sites, the amount of X in Shenzhen was still lower despite a significant $HO_2$ heterogeneous
uptake, which might be closely related to the environmental conditions as discussed in Sect. 4.2.

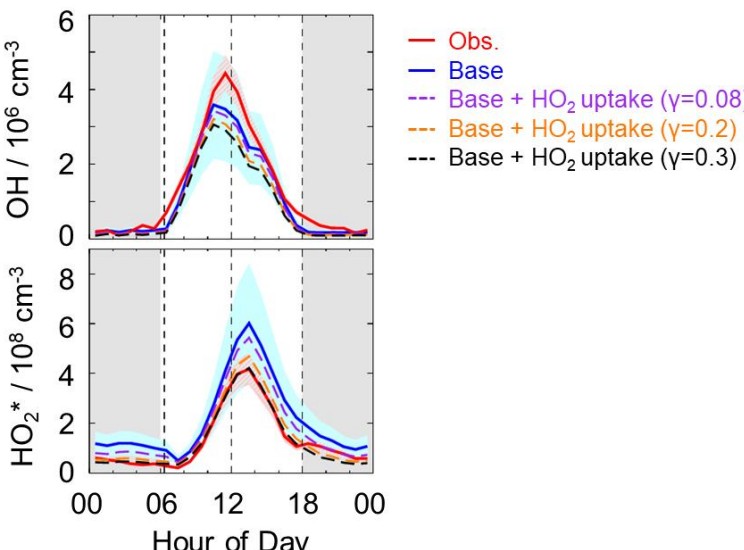


**Figure 6: The diurnal profiles of the observed and modeled radical concentrations. The red and blue areas denote 1-σ uncertainties of measured and simulated radical concentrations by the base model, respectively. The orange, purple and black lines denote the simulations by the model which added the $HO_2$ heterogeneous uptake with different uptake coefficient. The grey areas denote nighttime.**

It is noted that the estimated strong influence is speculative because of the uncertainties of measurements and simulations.
Overall, the $\gamma$ evaluated in this study was comparable with those observed at the Mt. Tai and Mt. Mang in China, and Kyoto
and Yokohama in Japan.

**4.4 Sources and sinks of ROx**

The detailed analysis of radical sources and sinks was crucial to exploring radical chemistry. The experimental budget for HO$_2$
and RO$_2$ radicals could not be conducted because RO$_2$ was not measured during this campaign. Herein, we showed the
simulated results by the base model. Figure 7 illustrates the diurnal profiles of ROx primary production rate ($P$(ROx)) and
termination rate ($L$(ROx)), and the contributions of different channels during the daytime.

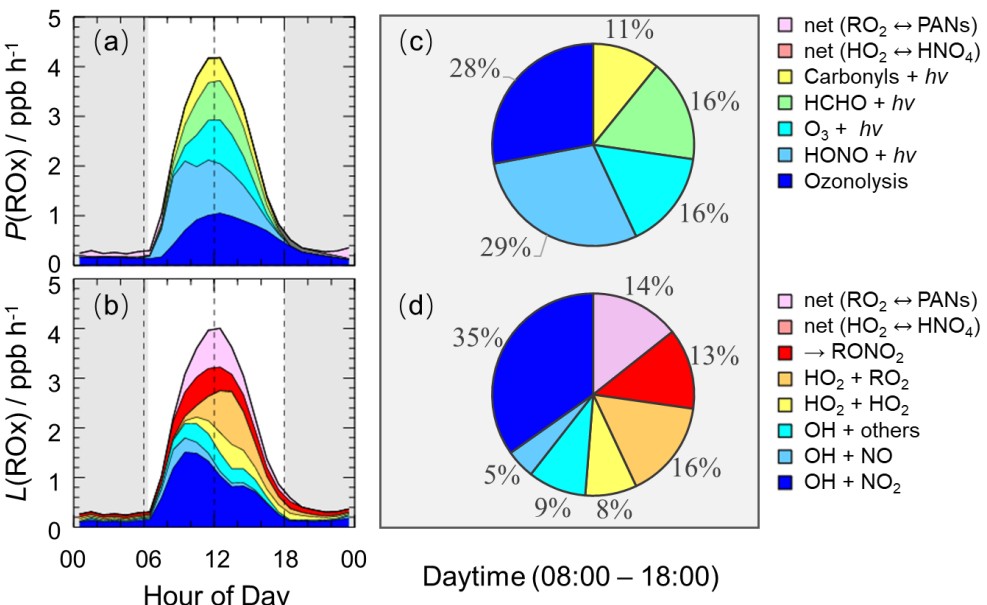


**Figure 7: The diurnal profiles of ROx primary production rate (a) and termination rate (b) simulated by the base model, and the**
**contributions of different channels to ROx primary production rate (c) and termination rate (d) during the daytime (08:00-18:00).**
**The grey areas denote nighttime.**
The ROx primary production and termination rates were basically in balance for the entire day, with maxima of 4 ppb h$^{-1}$
around noontime. The ROx primary production rate was similar to those at Heshan (4 ppb h$^{-1}$) and Wangdu (5 ppb h$^{-1}$) sites,
but lower than those at Backgarden (11 ppb h$^{-1}$), Yufa (7 ppb h$^{-1}$), Taizhou (7 ppb h$^{-1}$) and Chengdu (7 ppb h$^{-1}$) sites (Lu et al.,
2013;Lu et al., 2012;Tan et al., 2017;Tan et al., 2019;Yang et al., 2021). During the daytime, the $P$(ROx) mainly came from
the OH and HO$_2$ primary production. HONO and O$_3$ photolysis dominated the OH primary production, and HCHO photolysis
dominated the HO$_2$ primary production. Thus, $P$(ROx) was dominated by the photolysis reactions, in which the photolysis of
HONO, O$_3$, HCHO, and carbonyls accounted for 29%, 16%, 16%, and 11% during the daytime, respectively. In the early
morning, HONO photolysis was the most important primary source of ROx, and the contribution of O$_3$ photolysis became
progressively larger and was largest at noontime. A large discrepancy between the ratio of HONO photolysis rate to O$_3$
photolysis rate in summer/autumn and that in winter occurs generally. The vast majority of OH photolysis source is attributed
to HONO photolysis in winter because of the higher HONO concentration and lower O$_3$ concentration. About half of $L$(ROx)
came from OH termination, which occurred mainly in the morning, and thereafter, radical self-combination gradually became
the major sink of ROx in the afternoon. OH + NO$_2$, OH + NO, and OH + others contributed 35%, 5%, and 9% to $L$(ROx),
respectively. $HO_2 + HO_2$ and $HO_2 + RO_2$ accounted for 8% and 16% in $L(ROx)$.
**4.5 AOC evaluation**
AOC controls the abundance of precursors and the production of secondary pollutants (Yang et al., 2020a;Elshorbany et al.,
2009), and thus it is necessary to quantify AOC for understanding photochemical pollution. The AOC has been evaluated in
previous studies, as shown in Table 1. Overall, the AOC values in summer are higher than those in autumn and winter, and the
values at lower latitudes are higher than those at higher latitudes for the same season. The vast majority of AOC in previous
studies are evaluated based on the non-observed radical concentrations.
**Table 1: Summary of OH concentrations and AOC values reported in previous field campaigns.**

| Location | Season, year | Site | Observed or non-observed of OH radicals | AOC / $10^8$ molecules $cm^{-3}$ $s^{-1}$ | References |
|---|---|---|---|---|---|
| Beijing, China | summer, 2018 | urban | non-observed values | 0.89[a] | (Liu et al., 2021) |
| Beijing, China | summer, 2018 | suburban | non-observed values | 0.85[a] | (Liu et al., 2021) |
| Beijing, China | winter, 2018 | urban | non-observed values | 0.21[a] | (Liu et al., 2021) |
| Beijing, China | winter, 2018 | suburban | non-observed values | 0.16[a] | (Liu et al., 2021) |
| Hongkong, China | summer, 2011 | suburban | non-observed values | 2.04[a,b] | (Xue et al., 2016) |
| Santiago, Chile | summer, 2005 | urban | non-observed values | 3.4[a] | (Elshorbany et al., 2009) |
| Hong Kong, China | late summer, 2012 | coastal | non-observed values | 1.4[c] | (Li et al., 2018) |
| Hong Kong, China | autumn, 2012 | coastal | non-observed values | 0.62[c] | (Li et al., 2018) |
| Hong Kong, China | winter, 2012 | coastal | non-observed values | 0.41[c] | (Li et al., 2018) |
| Shanghai, China | summer, 2018 | urban | non-observed values | 1.0[c] | (Zhu et al., 2020) |
| Berlin, Germany | summer, 1998 | suburban | non-observed values | 0.14[d] | (Geyer et al., 2001) |
| Xianghe, China | autumn, 2019 | suburban | non-observed values | 0.49[c] | (Yang et al., 2020a) |
| Beijing, China | summer, 2014 | urban | non-observed values | 1.7[a] | (Feng et al., 2021) |

Note that:
[a] Peak values in the diurnal profiles; [b] Values on 25 August 2021; [c] Maximum over a period of time; [d] Maximum on some day.
Herein, we explored the AOC in Shenzhen based on the observed radical concentrations for the first time. As illustrated in
Fig. 8 (a), the diurnal profile of AOC exhibited a unimodal pattern, which was the same as the diurnal profile of OH
concentration and $j(NO_2)$, with a peak around noontime. The diurnal peak of AOC was $0.75 \times 10^8$ molecules $cm^{-3}$ $s^{-1}$ (11.8 ppb
$h^{-1}$). Comparatively, AOC in this study was comparable to those evaluated in Beijing (summer, 2018) and Hong Kong (autumn,
2012) (Li et al., 2018;Liu et al., 2021), but much lower than those evaluated in Hong Kong (summer, 2011) and Santiago

(summer, 2005) (Xue et al., 2016;Elshorbany et al., 2009).

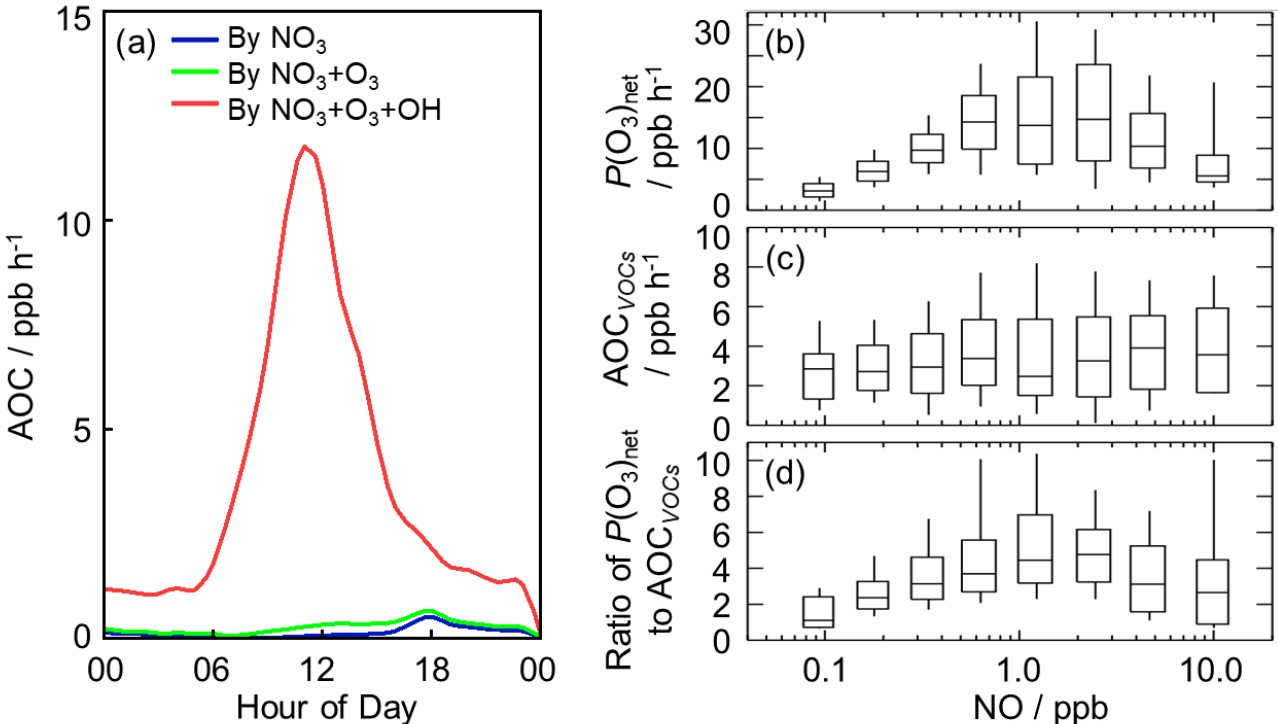


**Figure 8: (a) The diurnal profiles of AOC in this campaign. (b) NO dependence on $P(O_3)_{net}$ during the daytime. (c) NO dependence**
**on AOC$_{VOCs}$ during the daytime, and AOC$_{VOCs}$ denotes the atmospheric oxidation capacity only from the VOCs oxidation. (d) NO**
**dependence on the ratio of $P(O_3)_{net}$ to AOC$_{VOCs}$ during the daytime. The box-whisker plots in (b-d) give the 10%, 25%, median, 75%,**
**and 90% $P(O_3)_{net}$, AOC$_{VOCs}$ and the ratio of $P(O_3)_{net}$ to AOC$_{VOCs}$, respectively.**
As expected, the dominant contributor to the AOC during this campaign was OH, followed by $O_3$ and $NO_3$. Figure S7 shows
the fractional composition of the total AOC. The OH radical contributed about 95.7% of AOC during the daytime (08:00-
18:00). $O_3$, as the second important oxidant, accounted for only 2.9% of AOC during the daytime. The contribution of $NO_3$ to
AOC during the daytime can be ignored, with a contribution of 1.4%. At night, the contributions of $O_3$ and $NO_3$ to AOC were
higher. OH, $O_3$ and $NO_3$ accounted for 75.6%, 6.4%, and 18% in the first half of night (18:00-24:00), and they accounted for
87.7%, 5%, and 7.3% in the second half of night (00:00-08:00).
As the indictor for secondary pollution, net $O_3$ production rate, $P(O_3)_{net}$, can be calculated from the $O_3$ formation rate ($F(O_3)$)
and the loss rate ($L(O_3)$), as shown in Eq. (5-7) (Tan et al., 2017). The diurnal profiles of the speciation $F(O_3)$ and $L(O_3)$ were
shown in Fig. S8 in the Supplementary Information. The diurnal maxima of the modeled $F(O_3)$ and $L(O_3)$ were 18.9 ppb h$^{-1}$
and 2.8 ppb h$^{-1}$, with the maximum $P(O_3)_{net}$ of 16.1 ppb h$^{-1}$ at around 11:00. The modeled $P(O_3)_{net}$ in this study was comparable
to the net $O_3$ production rate in Wangdu in summer (Tan et al., 2017), while the net ozone production rate in Shenzhen was
much higher than the gross $O_3$ produciton rate in Beijing in winter (Tan et al., 2018).
$$F(O_3) = k_{HO_2+NO}[HO_2][NO] + \sum_i k_{RO_2i+NO}[RO_2]_i[NO] \tag{5}$$
$$L(O_3) = \theta j(O^1D)[O_3] + k_{O_3+OH}[O_3][OH] + k_{O_3+HO_2}[O_3][HO_2] + (\sum(k^i_{alkenes+O_3}[alkenes^i]))[O_3] \tag{6}$$
$$P(O_3)_\text{net} = F(O_3) - L(O_3) - k_{NO_2+OH}[NO_2][OH] \qquad (7)$$
where $\theta$ is the fraction of $O^1D$ from ozone photolysis that reacts with water vapor.
Herein, we presented the NO dependence on $P(O_3)_\text{net}$, $AOC_{VOCs}$, and the ratio of $P(O_3)_\text{net}$ to $AOC_{VOCs}$ in Fig. 8 (b-d), in which
$AOC_{VOCs}$ denotes the atmospheric oxidation capacity only from the VOCs oxidation, which includes the channels of primary
VOCs (excluding OVOCs, and mainly alkanes, alkenes, aromatics and isoprene) with OH radicals. An upward trend of $P(O_3)_\text{net}$
was presented with the increase of NO concentration when NO concentration was below 1 ppb, while $P(O_3)_\text{net}$ decreased with
the increase of NO concentration because $NO_2$ became the sink of OH radicals gradually when NO concentration was above
1 ppb. In terms of the NO dependence on $AOC_{VOCs}$, no significant variation was found, indicating VOCs oxidation was weakly
impacted by NO concentrations in this campaign. Since $AOC_{VOCs}$ can represent the VOCs oxidant rate, and thus the ratio of
$P(O_3)_\text{net}$ to $AOC_{VOCs}$ can reflect the yield of net ozone production from VOCs oxidation. Similar to $P(O_3)_\text{net}$, the ratios increased
with the increase of NO concentration when NO concentration was below 1 ppb, while the ratios decreased with the increase
of NO concentration when NO concentration was above 1 ppb, indicating the yield of net $O_3$ production from VOCs oxidation
would be lower within the low NO regime (< 1 ppb) and high NO regime (> 1 ppb). The median ratios ranged from 1.0 to 4.5,
and the maximum of the median ratios existed when NO concentration was approximately 1 ppb, with a value of approximately
4.5. The nonlinear response of the yield of net ozone production to NO indicated that it is necessary to optimize the NOx and
VOC control strategies for the reduction of $O_3$ pollution effectively.

**5 Conclusions**
The STORM field campaign was carried out at Shenzhen site in the autumn of 2018, providing the continuous OH and $HO_2^*$
observations in PRD since the Heshan campaign in 2014. The maximum diurnal OH and $HO_2^*$ concentrations, which were
measured by the PKU-LIF system, were $4.5 \times 10^6$ cm$^{-3}$ and $4.2 \times 10^8$ cm$^{-3}$, respectively. The observed OH concentration was
equal to that measured at the Heshan site (autumn campaign) but was lower than those measured in the summer campaigns in
China (Backgarden, Yufa, Wangdu, Taizhou and Chengdu sites). The observed $HO_2^*$ concentrations included the true $HO_2$
concentrations and an estimated interference from $RO_2$ radicals, and was much lower than those measured at the Backgarden
and Yufa sites in China.
The base model (RACM2-LIM1) could reproduce the observed OH concentration before 10:00, and thereafter, OH was
underestimated by the model when NO concentration dropped to low levels. The results of the radical experimental budget
indicated that OH underestimation was likely attributable to an unknown missing OH source at low NO regime. We diagnosed
the missing OH source by sensitivity runs, and unclassical OH recycling was identified again in this study. Good agreement
between the modeled and observed OH concentrations was achieved when a constant mixing ratio of the numerical species X,
equivalent to 0.1 ppb NO, was added into the base model. Additionally, we found isoprene and OVOCs might closely influence
the missing OH sources by comparing the composition of VOCs reactivity under the different NO intervals. Isoprene
isomerization mechanism (LIM1) can explain approximately 7% of the missing OH production rate, and no significant
contribution of MACR and MVK oxidation was found. As another potential OH source, OVOCs species should be further
explored to explain the remaining missing OH sources. As for $HO_2$ radicals, the overestimation of $HO_2^*$ concentration was
found, indicating that $HO_2$ heterogeneous uptake with the effective uptake coefficient of 0.3 might make a significant role in
$HO_2$ sinks. Good agreements of OH observations-simulations and $HO_2^*$ observations-simulations were both achieved when
the amount of X changed from 0.1 ppb to 0.25 ppb and the $HO_2$ effective uptake coefficient was 0.3.
The quantification of production and destruction channels of ROx radicals is essential to explore the chemical processes of
radicals. The ROx primary production and termination rates were balanced for the entire day, with maxima of 4 ppb $h^{-1}$, similar
to those at the Heshan and Wangdu sites. Photolysis channels dominated the ROx primary production rate, and the HONO, $O_3$,
HCHO, and carbonyls photolysis accounted for 29%, 16%, 16%, and 11% during the daytime, respectively. The most fraction
of ROx termination rate came from the reaction of OH + $NO_2$ in the morning. The radical self-combination gradually became
the major sink of ROx in the afternoon with the decreasing of NO concentrations. The reaction of OH + $NO_2$ and radical self-
combination accounted for 35% and 24% during the daytime, respectively.
In this campaign, AOC exhibited well-defined diurnal patterns, with a peak of 11.8 ppb $h^{-1}$. As expected, OH radicals, which
were the dominant oxidant, accounted for 95.7% of the total AOC during the daytime. $O_3$ and $NO_3$ contributed 2.9% and 1.4%
to total AOC during the daytime, respectively. The ratio of $P(O_3)_{net}$ to $AOC_{VOCs}$, which denotes the yield of net ozone production
from VOCs oxidation, trended to increase and then decrease as NO concentration increased, with a range of 1.0-4.5. Optimizing
the NOx and VOCs control strategies might be significant to realize the reduction of ozone concentrations based on the
nonlinear relationship between the yield of net ozone production from VOCs oxidation and NO concentrations.

*Data availability.* The data used in this study are available from the corresponding author upon request (k.lu@pku.edu.cn).

*Author contributions.* YH Zhang and KD Lu conceived the study. XP Yang analyzed the data and wrote the manuscript with
inputs from KD Lu. XP Yang, XF Ma, Y Gao contributed to the measurements of the HOx concentrations. All authors
contributed to the discussed results and commented on the manuscript.

*Competing interests.* The authors declare that they have no conflict of interest.

*Acknowledgment.* The authors thank the science teams of the STORM-2018 campaign. This work was supported by the
Beijing Municipal Natural Science Foundation for Distinguished Young Scholars (JQ19031),the National Research Program
for Key Issue in Air Pollution Control (2019YFC0214800) , and the National Natural Science Foundation of China (Grants
No. 91544225, 21522701, 91844301).
**Appendix A. Supplementary data**

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
