# Peer review of "Radical chemistry in the Pearl River Delta: observations and modeling of OH and HO₂ radicals in Shenzhen 2018"

_Atmospheric Chemistry and Physics, 2022_

## Author Comment (AC1)

**Response to Reviewer comment #1:**

**General comments:**

The paper by Yang et al. presents results from a measurement campaign in September / October 2018 at Shenzhen in the Pearl River Delta. Unfortunately, the collected data-set is not state-of-the art and the subsequent interpretation suffers from this lack of data and thus the manuscript does not bring any useful new insight into atmospheric chemistry. The shortcomings in the data-set compared with what is current state-of-the art are:

**Reply**

Thanks for your helpful comments which would help us to improve the manuscript. The comments and suggestions are valuable and very helpful for us. We have taken all these suggestions into account and have made corrections in this revised manuscript. Below are our responses to the specific comments, highlighted in blue, with changes to the manuscript highlighted in green.

**Specific comments:**

1. OH reactivity has not been measured: the analysis of the radicals budget is based on calculated OH reactivity, which is unsatisfactorily, especially given that $k_{OH}$ measurements are now widely available and add much confidence to the data set. Missing OH reactivity is widely observed under various conditions, and field campaigns quantifying OH and $HO_2$ should also measure OH reactivity to unravel possible missing OH reactivity, rather than using the calculated OH reactivity as a lower limit to evaluate the experimental OH and $HO_2$ data.

**Reply**

In this campaign, we measured $k_{OH}$ only during 05-19 October by laser the flash photolysis-laser induced fluorescence (LP-LIF) system (Liu et al., 2019), despite the absence of $k_{OH}$ continuous measurement during the period of radical observations (05-28 October 2018). The information on the LP-LIF system is added in Table S1 in the Supplementary Information. The timeseries of the observed and modeled $k_{OH}$ are presented in Fig. S2, in which data gaps were caused by the maintenance of the LP-LIF system. Timeseries of the observed and modeled $k_{OH}$ indicated that the simulations matched well with the observations within the uncertainties during 05-19 October 2018. Therefore, the model can be believed to reproduce the observed $k_{OH}$ values within the whole campaign.

We have added Figure S2 into the Supplementary Information, and revised the description of $k_{OH}$ in Sections 2.1, 3.3, and 4.1, and Table S1 in the Supplementary Information.

[Figure]

Figure S2: Timeseries of the observed and modeled $k_{OH}$ during 05-19 October 2018. The red and blue areas denote 1-σ uncertainties of the observations and simulations by the model, respectively. The grey areas denote nighttime.

**Revision**

**(1) Section 2.1:**

$k_{OH}$ was measured by the laser flash photolysis-laser induced fluorescence (LP-LIF) system.

**(2) Section 3.3:**

In this campaign, $k_{OH}$ was measured only for several days by the LIP-LIF system, which has been reported in the previous study (Liu et al., 2019). The timeseries of the observed and modeled $k_{OH}$ during 05-19 October 2018 are presented in Fig. S2 in the Supplementary Information. A good agreement between the observed and modeled $k_{OH}$ within the uncertainties was achieved, and thus the model can be believed to reproduce the observed $k_{OH}$ values within the whole campaign.

Moreover, to reflect the $k_{OH}$ in the whole campaign, the modeled values were shown in the $k_{OH}$ diurnal profiles (Fig. 3c) during 05-28 October 2018.

**(3) Section 4.1:**

As discussed in Section 3.3, it is believed that the model can reproduce the observed $k_{OH}$. Herein, to conduct the OH experiment budget in the whole campaign, we used the modeled $k_{OH}$ to calculate the OH destruction rate because the $k_{OH}$ was only measured in several days.

2. The reference cell for stabilizing the laser wavelength did not work during the campaign: rather high NO concentrations have therefore been used in the FAGE for $HO_2$ conversion, and it can be doubted that no interference from $RO_2$ measurements occurred under these conditions. This is even more strange, as the authors indicate line 99, that there is no obvious difference in $HO_2$ signal, when changing from 10 to 20 ppm. Does this mean the instrument works already under 100% $HO_2$ conversion? Then, an $RO_2$ interference seems very likely. However, 100% conversion efficiency is unlikely, as in a recent paper of the same group, describing a campaign carried out just a few months before in May / June 2018 (Ma et al., OH and $HO_2$ radicals chemistry at a suburban site during the EXPLORE-YRD

campaign in 2018, ACPD, doi.org/10.5194/acp-2021-1021), a conversion efficiency of 20% was obtained using 5 ppm. Or maybe the authors wanted to say that no obvious difference in $HO_2$ concentration was observed. Then, the $HO_2$ conversion rates under different NO concentrations need to be specified.

**Reply**

Thanks for your helpful suggestions. We have rechecked all the data and made corrections in the revised manuscript.

**Revision**

**(1) Section 2.2:**

In this campaign, NO mixing ratios were switched between 25 ppm (low NO mode) and 50 ppm (high NO mode). We calculated the $HO_2$ conversion rates under the two different NO concentrations by calibrating the PKU-LIF system. $HO_2$ conversion rates in low NO mode ranged within 80%-95%, while those in high NO mode were over 100%, demonstrating that the $HO_2$ measurement was affected by $RO_2$ radicals. Prior studies have reported the relative detection sensitivities ($\alpha_{RO_2}$) for the major $RO_2$ species, mainly from alkenes, isoprene and aromatics, when the $HO_2$ conversion rate was over 100% (Fuchs et al., 2011; Lu et al., 2012; Lu et al., 2013). Therefore, only the $HO_2$ observations in high NO mode were chosen and they were denoted as [$HO_2^*$], which was the sum of the true $HO_2$ concentration and a systematic bias from the mixture of $RO_2$ species $i$ which were detected with different relative sensitivities $\alpha_{RO_2}^i$, as shown in Eq. (1) (Lu et al., 2012). The true $HO_2$ concentration was difficult to calculated due to the $RO_2$ concentration measurements and their speciation were not available. Herein, we simulated the $HO_2$ and $HO_2^*$ concentrations by the model. The interference from $RO_2$ was estimated to be the difference between the $HO_2$ and $HO_2^*$ concentrations.

$$[HO_2^*] = [HO_2] + \sum(\alpha_{RO_2}^i \times [RO_2]_i) \tag{1}$$

**(2) The figures and descriptions of HO₂ concentration were revised in Section 3.2:**

The diurnal maximum of the observed $HO_2^*$, the modeled $HO_2^*$ and the modeled $HO_2$ concentrations were $4.2 \times 10^8$ cm$^{-3}$, $6.1 \times 10^8$ cm$^{-3}$, and $4.4 \times 10^8$ cm$^{-3}$, respectively. The difference between the modeled $HO_2^*$ and $HO_2$ concentrations can be considered a modeled $HO_2$ interference from $RO_2$ (Lu et al., 2012). The $RO_2$ interference was small in the morning, while it became larger in the afternoon. It ranged within 23%-28% during the daytime (08:00-17:00), which was comparable with those in the Backgarden and Yufa sites in China, Borneo rainforest in Malaysia (OP3 campaign, aircraft), and UK (RONOCO campaign, aircraft) (Lu et al., 2012; Lu et al., 2013; Jones et al., 2011; Stone et al., 2014). The observed $HO_2^*$ was overestimated by the model, indicating the $HO_2$ heterogeneous uptake might have a significant impact during this campaign. The diurnal maximum of $HO_2^*$ concentration observed in Shenzhen was much lower than those observed in the Yufa and Backgarden sites (Lu et al., 2012; Lu et al., 2013; Hofzumahaus et al., 2009).

[Figure]

Figure 3: (a-b) The diurnal profiles of the observed and modeled OH, $HO_2^*$ and $HO_2$ concentrations. (c) The diurnal profiles of the modeled $k_{OH}$. (d) The composition of the modeled $k_{OH}$. The red areas in (a-b) denote 1-σ uncertainties of the observed OH and $HO_2^*$ concentrations. The blue areas in (a-b) denote 1-σ uncertainties of the modeled OH and $HO_2^*$ concentrations, and the grey area in (b) denotes 1-σ uncertainties of the modeled $HO_2$ concentrations. The grey areas in (a-c) denote nighttime. ACD denotes acetaldehydes. ALD denotes the C3 and higher aldehydes. ACT and KET denote acetone and ketones. MACR and MVK denote methacrolein and methyl vinyl ketone.

**(3) The timeseries of HO₂ concentrations were revised in Figure S1:**

[Figure]

Figure S1: Timeseries of the OH, $HO_2^*$, $HO_2$ concentrations and $k_{OH}$ in this study. The grey areas denote nighttime.

**(4) The observed HO₂ concentrations can influence the OH experimental budget, so the description in Section 4.1 was revised:**

It is noted that the OH production rate was overestimated because we used $HO_2^*$ concentrations instead of $HO_2$ concentrations here. Thus, the missing OH source was the lower limit here, demonstrating more unknown OH sources need to be further explored.

[Figure]

Figure 4: (a) The diurnal profiles of OH production and destruction rates and the proportions of different known sources in the calculated production rate during the daytime. The blue line denotes the OH destruction rate, and the colored areas denote the calculated OH production rates from the known sources. (b) The missing OH source which was the discrepancy between the OH destruction and production rates, and the OH production rate which was ten times the production rate derived from LIM1 mechanism. The grey areas denote nighttime.

**(5) The NO dependence of HOx radicals in Fig. 5 was revised:**

[Figure]

Figure 5: NO dependence of OH and $HO_2^*$ radicals. The red box-whisker plots give the 10%, 25%, median, 75%, and 90% of the HOx observations. The blue circles show the median values of the HOx simulations by the base model, and the green circles show the HOx simulations by the model with X mechanism. Total VOCs reactivity and their organic speciation are presented by pie charts at the different NO intervals at the top. Only daytime values and NO concentration above the detection limit of the instrument were chosen. ACD and ACT denote acetaldehyde and acetone, respectively. ALD denotes the C3 and higher aldehydes. KET denotes ketones. MACR and MVK, which are both the isoprene oxidation products, denote methacrolein and methyl vinyl ketone, respectively.

3. Your model underestimates OH concentration under low NO conditions. I am very surprised that you claim that this is due to an unknown chemical X species, without even loosing a word about possible interferences in the OH measurements. Such increasing OH interferences with decreasing NO concentrations have been identified unequivocally with different FAGE instruments, and an experimental technique has been developed to quantify such possible interferences, this needs to be discussed. And even though some FAGE systems might be more prone to this interference than others, the FAGE community seems to agree on, that occasional measurements with such a pre-injector system are indispensable during field campaigns, especially when low NO concentrations are expected during the campaign. Looking at your above-mentioned paper describing a field campaign a few months before this one, it seems that you had already developed such a pre-injector system at the time of the campaign, because you had already used it. So why did you not use it in this campaign? In my opinion it is idle to discuss OH measurements that are underestimated by the model at low NO conditions, as long as OH-interference to an unknown species has been excluded by experiments using a pre-injector.

**Reply**

Thanks for your suggestions, the pre-injector system did not be applied in this campaign, but it is believed that the interference in OH measurement in this campaign was negligible by analyzing the PKU-LIF system and the environmental conditions during the campaign.

PKU-LIF system has been applied to measure HOx concentrations for several campaigns. We used the pre-injector system to quantify the possible interferences since 2014, including the campaigns conducted in Wangdu site (Tan et al., 2017), Heshan site (Tan et al., 2019), Huairou site (Tan et al., 2018), Taizhou site (Ma et al., 2022, in review, ACPD), and Chengdu site (Yang et al., 2021). No significant internal interference was found in the prior studies, demonstrating that the accuracy of the PKU-LIF system has been determined for several times.

Moreover, the potential interference may exist when the sampled air contained alkenes, ozone, and BVOCs (Mao et al., 2012; Fuchs et al., 2016; Novelli et al., 2014), indicating the environmental conditions, especially $O_3$, alkenes and isoprene, are important to the OH interferences. To further explore the potential interference in this campaign, we take Wangdu campaign as an example to compare the major environmental conditions between the prior campaigns and Shenzhen campaign here. During the Wangdu campaign, the chemical modulation tests were conducted on 29 June, 30 June, 02 July, and 05 July 2014, respectively (Tan et al., 2017). The daily mean $O_3$, alkenes (ethene, butadiene and other anthropogenic dienes, internal alkenes and terminal alkenes) and isoprene concentrations during the daytime

on 29 June were 94.1, 3.8, 1.9 ppb, those on 30 June were 92.2, 2.7, and 1.9 ppb, those on 02 July were 52.9, 1.5, and 0.5 ppb, and those on 05 July were 68.5, 2.4, and 0.9 ppb. The $O_3$, alkenes and isoprene concentrations on 29 June were highest among those on 29 June, 30 June, 02 July and 05 July, and thus the potential interference on 29 June can be considered the highest among the four days. The results indicated that the potential interference during the daytime in Wangdu was negligible.

Here, we also showed the major parameters related to OH interference in Shenzhen in Table S2 in the Supplementary Information. The daily mean $O_3$ and isoprene concentrations during the daytime in Shenzhen were within 8.6-91.7 ppb and 0.1-1.0 ppb, which were both lower than those on 29 June in Wangdu. In terms of the alkenes, only 10,16-17 October 2018 in Shenzhen campaign were higher than that observed on 29 June in Wangdu, but the $O_3$ concentrations on the three days in Shenzhen were only 21.9, 13.9, and 8.6 ppb, and the isoprene concentrations on the three days in Shenzhen were only 0.3, 0.2, and 0.1 ppb, respectively. Overall, the environmental condition in Shenzhen was less conducive to generating potential OH interference than that in Wangdu. Therefore, it is not expected that OH measurement in this campaign was affected by the internal interference.

We have added the description of interval interference in Section 2.2 and the Supplementary Information.

Table S2: The daily mean $O_3$, alkenes (ethene, butadiene and other anthropogenic dienes, internal alkenes and terminal alkenes) and isoprene concentrations during the daytime (08:00-17:00) in the STORM campaign in this study.

| Date / Species | 10-05 | 10-06 | 10-07 | 10-08 | 10-09 | 10-10 | 10-11 | 10-12 | 10-13 | 10-14 | 10-15 | 10-16 |
|---|---|---|---|---|---|---|---|---|---|---|---|---|
| $O_3$ (ppb) | 81.4 | 83.8 | 91.7 | 86.7 | 48.1 | 21.9 | 30.2 | 42.6 | 46.8 | 38.7 | 40.2 | 13.9 |
| Alkenes (ppb) | 1.4 | 1.8 | 3.6 | 2.3 | 3.2 | 5.4 | 2.9 | 2.4 | 2.6 | 1.4 | 1.6 | 4.9 |
| Isoprene (ppb) | 0.4 | 0.4 | 0.4 | 0.5 | 0.4 | 0.3 | 0.1 | 0.3 | 0.4 | 0.4 | 0.6 | 0.2 |

| Date / Species | 10-17 | 10-18 | 10-19 | 10-20 | 10-21 | 10-22 | 10-23 | 10-24 | 10-25 | 10-26 | 10-27 | 10-28 |
|---|---|---|---|---|---|---|---|---|---|---|---|---|
| $O_3$ (ppb) | 8.6 | 16.2 | 39.4 | 45.8 | 47.2 | 25.2 | 40.9 | 36.5 | 55.2 | 56.5 | 60.9 | 60.8 |
| Alkenes (ppb) | 4.7 | 3.2 | 2.1 | 1.3 | 1.2 | 2.5 | 2.9 | 2.7 | 1.2 | 3.4 | 1.7 | 1.7 |
| Isoprene (ppb) | 0.1 | 0.1 | 0.5 | 0.3 | 0.8 | 0.8 | 0.4 | 0.3 | 0.5 | 1.0 | 0.6 | 0.8 |

**Revision**

**(1) Section 2.2:**

Additionally, prior studies reported that OH measurement might be affected by the potential interference, when the sampled air contained ozone, alkenes and BVOCs (Mao et al., 2012; Fuchs et al., 2016; Novelli et al., 2014), indicating the environmental conditions are important to the production of interference. The pre-injector is usually used to test the potential OH

interference, and has been applied to our PKU-LIF system to quantify the possible interferences for several campaigns, including the campaigns conducted in Wangdu, Heshan, Huairou, Taizhou and Chengdu sites (Tan et al., 2017; Tan et al., 2019; Tan et al., 2018; Yang et al., 2021). No significant internal interference was found in the prior studies, demonstrating the accuracy of the PKU-LIF system has been determined for several times. Moreover, to further explore the potential interference in this campaign, we compared the major environmental conditions, especially $O_3$, alkenes and isoprene, between Shenzhen and Wangdu sites, as shown in the Supplementary Information. The environmental condition in Shenzhen was less conducive to generating interference than that in Wangdu, and the details were presented in the Supplementary Information. Therefore, it is not expected that the OH measurements in this campaign were affected by the internal interference.

**(2) The detailed information on potential OH interference was added in the Supplementary Information:**

We compared the environmental conditions in Shenzhen and Wangdu sites. The chemical modulation tests, which was applied to test the potential OH interference, were conducted on 29 June, 30 June, 02 July and 05 July 2014 in Wangdu (Tan et al., 2017). During the campaign in Wangdu, the daily mean $O_3$, alkenes (ethene, butadiene and other anthropogenic dienes, internal alkenes and terminal alkenes) and isoprene concentrations during the daytime on 29 June were 94.1, 3.8, 1.9 ppb, those on 30 June were 92.2, 2.7, and 1.9 ppb, those on 02 July were 52.9, 1.5, and 0.5 ppb, and those on 05 July were 68.5, 2.4, and 0.9 ppb, respectively. The $O_3$, alkenes and isoprene concentrations on 29 June were the highest among those on 29 June, 30 June, 02 July and 05 July, and thus the potential interference on 29 June can be considered the highest among the four days. The chemical modulation results indicated that the potential interference during the daytime in Wangdu was negligible (Tan et al., 2017).

As shown in Table S2, the $O_3$, alkenes and isoprene concentrations in Shenzhen were within 8.6-91.7 ppb, 1.2-5.4 ppb, and 0.1-1.0 ppb, respectively. The $O_3$ concentrations in Shenzhen (8.6-91.7 ppb) were lower than those on 29 June (94.1 ppb) and 30 June (92.2 ppb) in Wangdu. Similarly, the isoprene concentrations in Shenzhen (0.1-1.0 ppb) were also lower than those on 29 June (1.9 ppb) and 30 June (1.9 ppb) in Wangdu. In terms of the alkenes, only the concentrations on 10, 16-17 October 2018 (4.7-5.4 ppb) in Shenzhen were higher than that observed on 29 June (3.8 ppb) in Wangdu, but the $O_3$ concentrations on the three days in Shenzhen were only 21.9, 13.9, and 8.6 ppb, and the isoprene concentrations on the three days in Shenzhen were only 0.3, 0.2, and 0.1 ppb, respectively.

Overall, the environmental condition in Shenzhen was less conducive to generating potential OH interference than that in Wangdu. Therefore, it is not expected that OH measurement in this campaign was affected by the internal interference.

**References**

[revised manuscript text omitted]

---

## Author Comment (AC2)

**Response to Reviewer comment #2:**

**General comments:**

This manuscript describes OH and $HO_2$ measurements in Shenzhen during Autumn 2018 and the skills of a photochemical box model to reproduce the observed radical levels when constrained with simultaneous observations of key reactants, to test the atmospheric photochemistry theory. The results showed that the model underestimated OH levels while it reproduced $HO_2$ levels well. Missing recycling from $HO_2$ to OH by species X was suggested, while the required levels of X were not very high (0.1 ppb). The atmospheric oxidation capacity was calculated on the observation basis and compared with the $O_3$ formation rate. The results, though not including OH reactivity measurements, are worth to be added to those from numbers of previous field studies, for future diagnostics of missing processes in the model. Though the number of field studies in China has been increasing, the chemical conditions are very different among the studies and more field evidence is necessary. However, for this purpose, clarification is necessary at several points. First, uncertainty analysis needs to be provided for both observations and model simulations, particularly when the authors claim introduction of X to explain the model's underestimation of the OH levels. Second, the authors cited values or results from previous studies for comparison but in-depth analysis/discussion across the studies were not provided in search for missing processes. The authors should specify the important characteristics of the conditions studied during this campaign and what is enabled with the observational results. Third, I am afraid that the major OH term of the atmospheric oxidation capacity is not very innovative; it is just the OH reactivity multiplied to the OH concentrations, i.e., OH loss rate, and thus it is very natural that it correlates with $F(O_3)$, i.e., the $HO_2/RO_2$ + NO rate, when the OH loss produces $RO_2/HO_2$ and the peroxy radicals undergo reactions with NO. Overall, I would suggest major revisions regarding the points above and the following specific comments.

**Reply**

   We thank you for all the valuable comments and suggestions which are helpful for the important guiding significance to us. We have taken all these comments and suggestions into account as follows:

(1)  First, we added the uncertainty analysis for the observed and modeled radical concentrations, as shown in the revised figure in the Reply to Question 11.

(2)  Second, the comparisons of radical concentrations and chemical conditions between this campaign and other campaigns were added, as shown in Reply to Question 3 and Question 9. Additionally, we explored the influencing factors of missing OH sources and further evaluated the contribution of isoprene chemistry to radical sources.

(3)  Third, as for the analysis of AOC, we added the NO dependence of the ratio of $P(O_3)$ to AOC to explore the ozone formation from VOCs oxidation, as shown in the Reply to Question 17.

   Below are our responses to the specific comments, highlighted in blue, with changes to the manuscript highlighted in green.

**Specific comments:**

1. Page 1, Line 26. Definition of the atmospheric oxidation capacity should be briefly mentioned in Abstract.

**Reply**

We have added the definition of atmospheric oxidation capacity in the Abstract, and we changed the unit of AOC according to Question 17.

**Revision**

As the sum of the respective oxidation rates of the pollutants via reactions with oxidants, the atmospheric oxidation capacity was evaluated, with a peak of 11.8 ppb h$^{-1}$ around noontime.

2. Page 1, Line 26. x -> times character.

**Reply**

As the Reply to Question 1, the '×' here has been deleted. Besides, we have revised the character of '×' in the whole manuscript.

3. Page 2, Line 56. What are the important chemical conditions for this STORM campaign, in terms of differences from previous studies done in PRD, for example, city center/rural, NOx/BVOC levels, seasons etc.? From the inset map of Figure 1, I was not able to see if the site was in the city or in a rural region.

**Reply**

We have added the description of the site in Section 2.1 and the comparison of environmental conditions between the three campaigns conducted in PRD (Backgarden, Heshan, and Shenzhen) in Section 3.1.

**Revision**

**(1) Section 2.1:**

As shown in Fig. 1, the Shenzhen site, which belongs to the urban site, is located in the university town, and is surrounded by residential and commercial areas.

Overall, this site has no significant local pollution sources nearby, but can represent the urban pollution characteristics (Huang et al., 2012b; Huang et al., 2012a; Gao et al., 2018).

**(2) Section 3.1:**

Moreover, we compared the environmental conditions between the Backgarden (rural site), Heshan (suburban site), and Shenzhen (urban site) campaigns conducted in PRD in Table S3 in the Supplementary Information. No significant discrepancy in temperature was found in the Shenzhen and Heshan campaigns, which were both conducted in autumn. The temperature in the Backgarden campaign conducted in summer was higher than those in Shenzhen and Heshan. The relative humidity in Shenzhen and Backgarden was higher than that in Heshan. Compared to the chemical conditions in the Heshan campaign conducted in autumn as well, the

concentrations of CO, NO, $NO_2$, HONO, alkenes, aromatics, and HCHO in Shenzhen were lower, which might be because there were no significant local pollution sources nearby in the Shenzhen site although it was an urban site. However, the concentration of $O_3$ which is the typical secondary pollutant in Shenzhen was higher than that in Heshan. Compared to the environmental conditions in Heshan, the higher $O_3$ concentration in Shenzhen might benefit from the weather condition which was characterized by the stronger solar radiation and slightly higher temperatures.

Table S3: The meteorological parameters, photolysis rate constant, and concentrations of trace gases in the Backgarden, Heshan, and Shenzhen campaigns.

| Parameters | Backgarden (2006, summer) | Heshan (2014, autumn) | Shenzhen (2018, autumn) |
|---|---|---|---|
| Temperature (K) | 303.9 | 295.1 | 297.5 |
| Pressure (hPa) | 1002.9 | 1010.1 | 1009.7 |
| Relative humidity (%) | 72.3 | 66 | 75.4 |
| $j(O^1D)$ / $10^{-5}$ $s^{-1}$ | 3.6 | 1.3 | 1.8 |
| $j(NO_2)$ / $10^{-3}$ $s^{-1}$ | 7.6 | 4.1 | 5.7 |
| CO / ppb | 948.6 | 642.7 | 386.6 |
| NO / ppb | 5.7 | 3.6 | 2.6 |
| $NO_2$ / ppb | 14.3 | 18.7 | 14.9 |
| $O_3$ / ppb | 32.3 | 26.5 | 32.2 |
| HONO / ppb | 1.0 | 1.4 | 0.5 |
| Alkanes / ppb | 13.9 | 16.7 | 20.2 |
| Alkenes / ppb | 2.1 | 6.0 | 2.8 |
| Aromatics / ppb | 11.2 | 8.6 | 8.2 |
| HCHO / ppb | -- | 5.9 | 3.3 |

Note that:

The $j(O^1D)$ and $j(NO_2)$ were the mean values during the noontime, and other parameters were the mean values during the whole day.

4. Page 3, line 68. The coordinate should be 22.60 deg N and 113.97 deg E? (decimal points)

**Reply**

We revised the coordinate as your suggestions.

5. Page 4, lines 89-96. Any literature to which the readers refer for further information of the specific FAGE instrument? Also, the uncertainty in OH and $HO_2$ measurements should be quantified.

**Reply**

We added more references on the FAGE instrument and added the HOx measurement uncertainties in the manuscript which have been represented in Table S1 in the Supplementary Information.

**Revision**

**Section 2.2:**

Further detailed information on the instrument can be found in previous studies (Heard and Pilling, 2003; Fuchs et al., 2008; Holland et al., 1995; Hofzumahaus et al., 1996; Fuchs et al., 2011).

Overall, the measurement uncertainties of OH and $HO_2^*$ radicals were 11% and 15%, respectively, as shown in Table S1 in the Supplementary Information.

6. Page 4, line 107. Did the author mean latest isoprene chemistry?

**Reply**

Yes, RACM2-LIM1 means the latest isoprene chemistry. We wrote the wrong word 'lasted', and revised it to 'latest'.

**Revision**

In this work, we conducted the radical closure experiment based on the Regional Atmospheric Chemical Mechanism updated with the latest isoprene chemistry (RACM2-LIM1).

7. Page 4, line 110. What are the "long-lived" species? Readers may think $CO_2$ or $CH_4$ as long-lived, which are not surely supposed in this context. I believe they are the modeled carbonyls/peroxides etc. But did they reach steady state within 2 days of integration? Did the authors assume a fast turnover time constant (dilution constant) for them? If so, any justification of the assumption?

**Reply**

The expression was not accurate, and thanks for your suggestion. Herein, the spin-up for the model is used to let the unconstrained species, which are mainly some intermediate species, approach the steady state relative to the constrained species. We have revised the description in Section 2.3.

**Revision**

The model was operated in time-dependent mode with a 5-min time resolution, and a 2-d spin-up time which was to make the unconstrained species approach the steady state relative to the constrained species.

8. Page 4, lines 118. It is confusing to mention the observed k_OH from other studies, as that measurement was not available for this particular study.

**Reply**

In this campaign, we measured $k_{OH}$ only during 05-19 October by the laser flash photolysis-laser induced fluorescence (LP-LIF) system (Liu et al., 2019), despite the absence of $k_{OH}$ continuous measurement during the period of radical observations (05-28 October 2018). The information on LP-LIF is shown in Table S1 in the Supplementary Information. The timeseries

of the observed and modeled $k_{OH}$ are presented in Fig. S2, in which data gaps were caused by the maintenance of the LP-LIF system. Timeseries of the observed and modeled $k_{OH}$ indicated that the simulations matched well with the observations within the uncertainties during 08-12 October 2018. Therefore, the model can be believed to reproduce the observed $k_{OH}$ values within the whole campaign.

We have added Figure S2 into the Supplementary Information, and revised the description of $k_{OH}$ in Sections 2.1, 3.3, and 4.1, and Table S1 in the Supplementary Information.

[Figure]

Figure S2: Timeseries of the observed and modeled $k_{OH}$ during 05-19 October 2018. The red and blue areas denote 1-σ uncertainties of the observations and simulations by the model, respectively. The grey areas denote nighttime.

**Revision**

**(1) Section 2.1:**

$k_{OH}$ was measured by the laser flash photolysis-laser induced fluorescence (LP-LIF) system.

**(2) Section 3.3:**

In this campaign, $k_{OH}$ was measured only for several days by the LIP-LIF system, which has been reported in the previous study (Liu et al., 2019). The timeseries of the observed and modeled $k_{OH}$ during 05-19 October 2018 are presented in Fig. S2 in the Supplementary Information. A good agreement between the observed and modeled $k_{OH}$ within the uncertainties was achieved, and thus the model can be believed to reproduce the observed $k_{OH}$ values within the whole campaign.

Moreover, to reflect the $k_{OH}$ in the whole campaign, the modeled values were shown in the $k_{OH}$ diurnal profiles (Fig. 3c) during 05-28 October 2018.

**(3) Section 4.1:**

As discussed in Section 3.3, it is believed that the model can reproduce the observed $k_{OH}$. Herein, to conduct the OH experiment budget in the whole campaign, we used the modeled $k_{OH}$ to calculate the OH destruction rate because the $k_{OH}$ was only measured in several days.

9.  Page 8, Line 181. More discussion is preferred; what are the similarities and what are the differences to/from the previous studies in PRD?

**Reply**

The comparison between the three studies in Shenzhen, Heshan and Backgarden sites in PRD did be necessary. Here, we added the comparisons of HOx concentrations, $k_{OH}$ and missing OH sources in Sections 3.2, 3.3 and 4.2.2, respectively.

**Revision**

**(1) Section 3.2:**

Compared to the other campaigns conducted in PRD (Backgarden and Heshan), the diurnal maximum of the observed OH concentration in Shenzhen was equal to that observed in Heshan, and much lower than that observed in Backgarden where the observed OH concentration was nearly $15 \times 10^6$ cm$^{-3}$ (Hofzumahaus et al., 2009; Tan et al., 2019). The higher OH concentration in Backgarden site was closely correlated to the stronger solar radiation, as shown in Table S3 in the Supplementary Information.

The diurnal maximum of $HO_2^*$ concentration observed in Shenzhen was slightly higher than that observed in Heshan ($3 \times 10^8$ cm$^{-3}$), but much small than that observed in Backgarden ($18 \times 10^8$ cm$^{-3}$) (Hofzumahaus et al., 2009; Tan et al., 2019; Lu et al., 2012).

**(2) Section 3.3:**

Compared to the $k_{OH}$ variation in Shenzhen, the $k_{OH}$ observed in Backgarden and Heshan sites in PRD showed a stronger diurnal variation, with a minimum value at around noontime and a maximum value at daybreak. The $k_{OH}$ ranges in Backgarden and Heshan were 20-50 s$^{-1}$ and 22-32 s$^{-1}$ (Lou et al., 2010; Tan et al., 2019). Similar with the good agreement between the observed and modeled $k_{OH}$ during the several days in Shenzhen, the observed $k_{OH}$ in Backgarden was matched well with the modeled $k_{OH}$ which has included the OVOCs reactivity. In terms of the $k_{OH}$ in Heshan, Tan et al. (2019) reported that only half of the observed $k_{OH}$ was explained by the calculated $k_{OH}$ which was calculated from the measured trace gas concentrations. The missing $k_{OH}$ in Heshan was likely caused by unmeasured VOCs, demonstrating the necessary to measure more abundant VOCs species, especially OVOCs species.

**(3) Section 4.2.2:**

Compared to Shenzhen site, the X concentration in the Backgarden and Heshan sites in PRD were higher, which might be affected by the different air masses in the three sites. The $k_{OH}$ in Shenzhen site was much lower than those in Backgarden and Heshan sites, and the weaker variation of $k_{OH}$ in Shenzhen was observed. Under the influence of the East Asian monsoon, the prevailing wind for PRD area is mostly southerly during the summer months and mostly northerly during the winter months (Fan et al., 2005; Zhang et al., 2008). The Backgarden site is located in Guangzhou, and the Heshan site is located in Jiangmen. The two cities are along the north-south axis, and thus the air masses of the Backgarden and Heshan sites are intimately linked with each other, while the air mass in Shenzhen is more similar to Hongkong (Zhang et al., 2008).

10. Page 8, Figure 3c and d. The fraction of the modeled OVOCs is fairly large. More explanation is needed what these species are and how their concentrations are justified.

**Reply:**

Large fraction of OVOCs reactivities in $k_{OH}$ was also found in some previous studies (Lou et al., 2010; Lu et al., 2013; Fuchs et al., 2017; Whalley et al., 2021). About 50% of $k_{OH}$ was explained by OVOCs in Backgarden site. HCHO, acetaldehydes and higher aldehydes, and oxygenated isoprene products were the most important OH reactants in OVOCs, with a contribution of 30-40%, and other 10-20% came from other oxygenated compounds (ketones, dicarbonyl compounds, alcohols, hydroperoxides, nitrates etc.) (Lou et al., 2010). HCHO, acetaldehydes, MVK, MVCR and glyoxal accounted for one-third of the total $k_{OH}$ in Wangdu site (Fuchs et al., 2017).

In this study, we measured several OVOCs species, including HCHO, acetaldehydes (ACD) and higher aldehydes (ALD), acetone (ACT), ketones (KET) and isoprene oxidation products (MACR and MVK), so we constrained these species in the model and revised the composition of $k_{OH}$ in Fig. 3(c-d). The constrained OVOCs species accounted for 18% in the total $k_{OH}$, where HCHO, ACD, and ALD were the major contributors, with contributions of 18%, 32%, and 38% to the constrained OVOCs, respectively. The contribution of aldehydes to the total $k_{OH}$ in this study (16%) was larger than that in Beijing (Whalley et al., 2021) and smaller with that in Wangdu (Fuchs et al., 2017). The unconstrained OVOCs reactivity, mainly from the model-generated intermediate species (glyoxal, methylglyoxal, methyl ethyl ketone, methanol, etc.), accounted for 11% in the total $k_{OH}$ in this campaign.

[Figure]

Figure 3: (a-b) The diurnal profiles of the observed and modeled OH, $HO_2^*$ and $HO_2$ concentrations. (c) The diurnal profiles of the modeled $k_{OH}$. (d) The composition of the modeled $k_{OH}$. The red areas in (a-b) denote 1-σ uncertainties of the observed OH and $HO_2^*$ concentrations. The blue areas in (a-b) denote 1-σ uncertainties of the modeled OH and $HO_2^*$ concentrations, and the grey area in (b) denotes 1-σ uncertainties of the modeled $HO_2$ concentrations. The grey areas in (a-c) denote nighttime. ACD denotes acetaldehydes. ALD denotes the C3 and higher aldehydes. ACT and KET denote acetone and ketones. MACR and MVK denote methacrolein and methyl vinyl ketone.

As for the unmeasured OVOCs species, it is difficult to justifying their concentrations, since the in-situ measuremnts of these species are missing, but the plausibility for them can be

checked. In this study, the mean $k_{OH}$ value was 20.8 s$^{-1}$, and thus the reactivitiy of the unconstrained OVOCs species was 2.3 s$^{-1}$. In the Backgarden site, the reactivity of oxygenated compounds (ketones, dicarbonyl compounds, alcohols, hydroperoxides, nitrates etc.), which accounted for 10%-20% of the total $k_{OH}$, was about 3.3-6.6 s$^{-1}$ (the mean $k_{OH}$ was about 32.9 s$^{-1}$) (Lou et al., 2010). The reactivity of these OVOCs species in our study were of similar magnitude as the reported by Lou et al. (2010), indicating the modeled unconstrained OVOCs reactivitys can be believed in this study.

**Revision:**

**Section 3.3**

Compared with inorganics reactivity, the larger fraction of $k_{OH}$ came from the VOCs group, with a contribution of 69% to $k_{OH}$. The contribution of alkanes, alkenes, and aromatics were 15%, 10%, and 12%, respectively. The isoprene reactivity related to temperature was mainly concentrated during the daytime, whereas the aromatics reactivity at night was higher. As for the OVOCs species, we measured several OVOCs species, including HCHO, acetaldehydes (ACD) and higher aldehydes (ALD), acetone (ACT), ketones (KET) and isoprene oxidation products (MACR and MVK), so we constrained these species in the model. The constrained OVOCs species accounted for 18% in the total $k_{OH}$, where HCHO, ACD, and ALD were the major contributors, with contributions of 18%, 32%, and 38% to the constrained OVOCs, respectively. The contribution of aldehydes in this study (16%) was larger than that in Beijing (Whalley et al., 2021) and smaller with that in Wangdu (Fuchs et al., 2017). The remaining reactivity was attributed to the unconstrained OVOCs reactivity, which came from the model-generated intermediate species (glyoxal, methylglyoxal, methyl ethyl ketone, methanol, etc.), with a contribution of 11% to the total $k_{OH}$. Large fraction of OVOCs reactivities in $k_{OH}$ was also found in some previous studies (Lou et al., 2010; Lu et al., 2013; Fuchs et al., 2017; Whalley et al., 2021). About 50% of $k_{OH}$ was explained by OVOCs in Backgarden site, and HCHO, ACD and ALD, and oxygenated isoprene products were the most important OH reactants in OVOCs, with a contribution of 30-40%, and other 10-20% came from other oxygenated compounds (ketones, dicarbonyl compounds, alcohols, hydroperoxides, nitrates etc.) (Lou et al., 2010). HCHO, ACD, MVK, MVCR and glyoxal accounted for one-third of the total $k_{OH}$ in Wangdu site (Fuchs et al., 2017). The large unconstrained OVOCs reactivity indicated it is necessary to measure more VOCs species in the future.

11. Page 8, line 188. Were the aerosol surface concentrations measured? Can the authors discuss maximum possible uptake coefficient from the surface concentrations?

**Reply**

The aerosol surface concentrations observations were not measured in this campaign, but they can be estimated roughly by multiplying the mass concentration of PM$_{2.5}$ by 20 (Chen et al., 2019; Wang et al., 2017). Herein, we discussed the possible HO$_2$ uptake coefficient from the surface concentrations.

First, we note that we rechecked our data and revised the description of HO$_2$ observations. In this campaign, NO mixing ratios were switched between 25 ppm (low NO mode) and 50 ppm (high NO mode). We calculated the HO$_2$ conversion rates under the two different NO

concentrations by calibrating the PKU-LIF system. $HO_2$ conversion rates in low NO mode ranged within 80%-95%, while those in high NO mode were over 100%, demonstrating that the $HO_2$ measurement was affected by $RO_2$ radicals. Prior studies have reported the relative detection sensitivities ($\alpha_{RO_2}$) for the major $RO_2$ species, mainly from alkenes, isoprene and aromatics, when the $HO_2$ conversion rate was over 100% (Fuchs et al., 2011; Lu et al., 2012; Lu et al., 2013). Therefore, only the $HO_2$ observations in high NO mode were chosen and they were denoted as $[HO_2^*]$, which was the sum of the true $HO_2$ concentration and a systematic bias from the mixture of $RO_2$ species $i$ which were detected with different relative sensitivities $\alpha_{RO_2}^i$, as shown in Eq. (1) (Lu et al., 2012). The true $HO_2$ concentration was difficult to calculated due to the $RO_2$ concentration measurements and their speciation were not available. Herein, we simulated the $HO_2$ and $HO_2^*$ concentrations by the model. The interference from $RO_2$ was estimated to be the difference between the $HO_2$ and $HO_2^*$ concentrations.

$$[HO_2^*] = [HO_2] + \sum(\alpha_{RO_2}^i \times [RO_2]_i) \tag{1}$$

The figures and descriptions of $HO_2$ concentration were revised in Section 3.2:

[Figure]

Figure 3: (a-b) The diurnal profiles of the observed and modeled OH, $HO_2^*$ and $HO_2$ concentrations. (c) The diurnal profiles of the modeled $k_{OH}$. (d) The composition of the modeled $k_{OH}$. The red areas in (a-b) denote 1-σ uncertainties of the observed OH and $HO_2^*$ concentrations. The blue areas in (a-b) denote 1-σ uncertainties of the modeled OH and $HO_2^*$ concentrations, and the grey area in (b) denotes 1-σ uncertainties of the modeled $HO_2$ concentrations. The grey areas in (a-c) denote nighttime. ACD denotes acetaldehydes. ALD denotes the C3 and higher aldehydes. ACT and KET denote acetone and ketones. MACR and MVK denote methacrolein and methyl vinyl ketone.

To evaluate the contribution of $HO_2$ uptake chemistry to radical concentrations in this study, we coupled $HO_2$ heterogeneous uptake into the base model (RACM2-LIM1) and conducted three sensitivity experiments, as shown in R1 and Eq. (1).

$$HO_2 + aerosol \rightarrow products \tag{R1}$$

$$k_{HO_2+aerosol} = \frac{\gamma * ASA * \nu_{HO_2}}{4} \qquad (1)$$

where ASA [$\mu m^2\ cm^{-3}$], which represents the aerosol surface area concentration, can be estimated by multiplying the mass concentration of PM$_{2.5}$ [$\mu g\ m^{-3}$] by 20 here because there were no direct ASA observations in this campaign (Chen et al., 2019; Wang et al., 2017). $\nu_{HO_2}$, which can be calculated by Eq. (2), refers to the mean molecular velocity of HO$_2$ with a unit of cm s$^{-1}$.

$$\nu_{HO_2} = \sqrt{\frac{8 * R * T}{0.033 * \Pi}} \qquad (2)$$

where T [K] and R [J mol$^{-1}$ K$^{-1}$] denote the ambient temperature and gas constant. $\gamma$, the HO$_2$ effective uptake coefficient, parameterizes the influence of some processes (Tan et al., 2020). $\gamma$ varies in the highly uncertain range of 0-1 (Song et al., 2022), and is the most critical parameters to impact HO$_2$ uptake chemistry. Only several observations of $\gamma$ have been reported (Taketani et al., 2012; Zhou et al., 2021; Zhou et al., 2020). The measured $\gamma$ at the Mt. Tai site and Mt. Mang site were 0.13-0.34 and 0.09-0.40, respectively (Taketani et al., 2012). The averaged value of the measured $\gamma$ was 0.24 in Kyoto, Japan in summer 2018 (Zhou et al., 2020). Zhou et al. (2021) reported the lower-limit values for median and average values of the measured $\gamma$ were 0.19 and 0.23±0.21 in Yokohama, Japan in summer 2019. Additionally, Li et al. (2018) set 0.2 as the value of $\gamma$ in the model, and Tan et al. (2020) calculated the $\gamma$ of 0.08±0.13 by the analysis of the measured radical budget in Wangdu.

Herein, we applied the two $\gamma$ (0.2 and 0.08), which have been used in the model, to evaluate the impact of HO$_2$ uptake on radical concentrations. The modeled HO$_2^*$ cannot match well with the observations when a $\gamma$ of 0.08 or 0.2 was set in the model. As the $\gamma$ increased to approximately 0.3, good agreement between the modeled and observed HO$_2^*$ concentration was achieved, demonstrating that the significant heterogeneous uptake might exist in this campaign.

It is noted that the estimated strong influence is speculative because of the uncertainties of measurements and simulations. Overall, the $\gamma$ evaluated in this study was comparable with those observed at the Mt. Tai and Mt. Mang in China, and Kyoto and Yokohama in Japan.

12. Page 8, line 193. It is worth mentioning where the study (Stevens et al. 1997) took place and add more information.

**Reply:**

The campaign reported by Stevens et al. (1997) was conducted at Idaho Hill, Colorado, during August and September in 1993. The experimental site was located in the remote mountains, with a wide range of chemical conditions to study the kinetics of the photochemistry of radicals. This study reported two major conclusions, one was that the HO$_2$/OH ratios were higher in clean environments than those in polluted environments, and the other was that the measured HO$_2$/OH ratios agreed well with predictions under polluted environments, and they were lower than predicted values under clean environments.

13. Page 8, lines 193-194. I did not understand what the authors meant with the sentence "The comparison of the measured $HO_2/OH$ ratio…".

**Reply:**

Stevens et al. (1997) reported that the agreement between the calculated and measured $HO_2/OH$ ratio was related to several possible explanations, including the $HO_2$ heterogeneous reactions, instrument and/or calibration error in the measurement of radicals, or problems with the chemical oxidation mechanism. Thus, it was inaccurate to justify the environmental conditions (clean or polluted) according to the agreement of the measured and calculated $HO_2/OH$ ratio.

Additionally, the measured $HO_2/OH$ cannot be obtained in this study because the observed $HO_2$ concentrations include the true $HO_2$ concentrations and the interference of $RO_2$ radicals.

Therefore, we revised the description of $HO_2/OH$ ratios in Section 3.2.

**Revision:**

The high modeled $HO_2/OH$ ratio around noontime (11:00-15:00), which was about 138, was found in this campaign, which was higher than those in the Backgarden and Chengdu sites (Hofzumahaus et al., 2009; Yang et al., 2021). High $HO_2/OH$ ratio is normally found only in clean air at low concentrations of Nox (Hofzumahaus et al., 2009; Stevens et al., 1997).

14. Page 10, line 232. The unknown OH source NEEDS TO explain.

**Reply**

To further explain the unknown OH source, we compared the environmental condition under the different NO intervals in Fig. 5 in Section 4.2.1 and Table S4 in the Supplementary Information. Besides, we explored the contribution of isoprene oxidation chemistry on missing OH source in Section 4.2.2.

**Revision**

**(1) Section 4.2.1:**

To further explore the influencing factors of OH underestimation, we presented the speciation VOCs reactivity under the different NO intervals, as shown in Fig. 5 and Table S4 in the Supplementary Information. The isoprene reactivity and total OVOCs reactivity (the sum of HCHO, ACD, ACT, ALD, KET, MACR, MVK and the modeled OVOCs) increased with the decrease of NO concentrations, while the anthropogenic VOCs reactivity (alkanes, alkenes and aromatics) was higher in high NO regime. Additionally, the $O_3$ concentration in low NO regime was significantly higher than those in high NO regime, and the temperature was slightly higher in low NO regime, demonstrating the photochemistry was more active in low NO regime in this campaign. Overall, the photochemistry and composition of VOCs reactivity, especially the isoprene and OVOCs species (mainly ACD, ACT and the modeled OVOCs), might closely impact the missing OH sources.

[Figure]

Figure 5: NO dependence of OH and $HO_2^*$ radicals. The red box-whisker plots give the 10%, 25%, median, 75%, and 90% of the HOx observations. The blue circles show the median values of the HOx simulations by the base model, and the green circles show the HOx simulations by the model with X mechanism. Total VOCs reactivity and their organic speciation are presented by pie charts at the different NO intervals at the top. Only daytime values and NO concentration above the detection limit of the instrument were chosen. ACD and ACT denote acetaldehyde and acetone, respectively. ALD denotes the C3 and higher aldehydes. KET denotes ketones. MACR and MVK, which are both the isoprene oxidation products, denote methacrolein and methyl vinyl ketone, respectively.

**(2) The median values of meteorological and chemical parameters during the daytime at the different NO intervals was added in Table S4 in the Supplementary Information:**

Table S4: The median values of meteorological and chemical parameters during the daytime at the different NO intervals.

| parameters | NO interval (< 0.2 ppb) | NO interval (0.2-0.6 ppb) | NO interval (0.6-2 ppb) | NO interval (> 2 ppb) |
|---|---|---|---|---|
| Temperature / K | 301.4 | 300.8 | 299.1 | 297.9 |
| $j(O^1D)$ / $10^{-6}$ s$^{-1}$ | 4.7 | 8.9 | 8.2 | 7.4 |
| $O_3$ concentration / ppb | 71.7 | 55.1 | 39.6 | 16.9 |
| Alkanes reactivity / s$^{-1}$ | 2.2 | 3.4 | 3.3 | 3.5 |
| Alkenes reactivity / s$^{-1}$ | 1.4 | 1.0 | 1.4 | 2.3 |
| Aromatics reactivity / s$^{-1}$ | 0.9 | 1.0 | 1.5 | 2.4 |

| | | | | |
|---|---|---|---|---|
| Isoprene reactivity / s$^{-1}$ | 1.1 | 1.1 | 0.8 | 0.5 |
| HCHO reactivity / s$^{-1}$ | 1.1 | 0.9 | 0.8 | 0.7 |
| ACD reactivity / s$^{-1}$ | 1.1 | 1.4 | 1.3 | 1.2 |
| ACT reactivity / s$^{-1}$ | 0.2 | 0.2 | 0.2 | 0.2 |
| ALD reactivity / s$^{-1}$ | 1.9 | 1.8 | 1.6 | 1.2 |
| KET reactivity / s$^{-1}$ | 0.0 | 0.0 | 0.0 | 0.0 |
| MACR reactivity / s$^{-1}$ | 0.2 | 0.2 | 0.1 | 0.1 |
| MVK reactivity / s$^{-1}$ | 0.3 | 0.2 | 0.1 | 0.1 |
| Modeled OVOCs reactivity / s$^{-1}$ | 2.5 | 2.2 | 2.2 | 2.4 |
| Alkanes concentration / ppb | 15.0 | 16.8 | 19.0 | 24.6 |
| Alkenes concentration / ppb | 1.6 | 1.6 | 2.0 | 3.4 |
| Aromatics concentration / ppb | 3.3 | 3.3 | 4.8 | 7.9 |
| Isoprene concentration / ppb | 0.4 | 0.4 | 0.3 | 0.2 |
| HCHO concentration / ppb | 5.6 | 4.3 | 3.8 | 3.5 |
| ACD concentration / ppb | 3.0 | 3.7 | 3.5 | 3.3 |
| ACT concentration / ppb | 3.2 | 3.7 | 3.3 | 2.7 |
| ALD concentration / ppb | 3.8 | 3.6 | 3.2 | 2.5 |
| KET concentration / ppb | 0.3 | 0.4 | 0.4 | 0.3 |
| MACR concentration / ppb | 0.2 | 0.2 | 0.1 | 0.1 |
| MVK concentration / ppb | 0.5 | 0.5 | 0.3 | 0.2 |

**(3) Section 4.2.2**

As discussed in Section 4.2.1, isoprene and OVOCs might have potential influence on the missing OH source. RO$_2$ isomerization reactions have also been shown to be of importance for the atmospheric fate of RO$_2$ from isoprene (Peeters et al., 2009; Peeters et al., 2014). The latest isoprene isomerization mechanism, which is called LIM1, has been coupled into our current base model. However, LIM1 mechanism was not included in the OH experimental budget which was conducted with the observations constrained, as shown in Section 4.1. Herein, we evaluated the contribution of LIM1 mechanism to the missing OH sources, as shown in Fig. 4 (b). LIM1 mechanism can explain approximately 7% of the missing OH sources during 10:00-16:00, when the missing OH production rate and the OH production rate derived from LIM1 were 2.47 ppb h$^{-1}$ and 0.17 ppb h$^{-1}$, respectively.

[Figure]

Figure 4: (a) The diurnal profiles of OH production and destruction rates and the proportions of different known sources in the calculated production rate during the daytime. The blue line denotes the OH destruction rate, and the colored areas denote the calculated OH production rates from the known sources. (b) The missing OH production rate which was the discrepancy between the OH destruction and production rates, and the OH production rate which was ten times the production rate derived from LIM1 mechanism. The grey areas denote nighttime.

Additionally, prior studies also reported that OH regeneration might be achieved from the oxidation of MACR and MVK, which are the major first-generated products of isoprene (Fuchs et al., 2018; Fuchs et al., 2014). As a potential explanation for the high OH concentration, the impacts of MACR and MVK oxidation were evaluated here. The modification of MACR oxidation scheme added the H-migration reactions of MACR oxidation products (Fuchs et al., 2014). The modification of MVK oxidation scheme added the reactions of MVK oxidation products with $HO_2$ radicals and the H-migration reactions of MVK oxidation products (Fuchs et al., 2018). As presented in Fig. S3 in the Supplementary Information, no significant of the MACR and MVK oxidation schemes was found in this campaign.

Overall, a large part of missing OH sources was not explained by the isoprene chemistry. In the future, the impact of OVOCs species which was another potential OH source on missing OH sources need to be further evaluated.

[Figure]

Figure S3: The diurnal profiles of the observed and modeled radical concentrations. The red and blue areas denote 1-σ uncertainties of measured and simulated radical concentrations by the base model, respectively. The green lines denote the simulations by the model with the oxidation of MACR, and the dark orange lines denote the simulations by the model with the oxidation of MVK. The grey areas denote nighttime.

15. Page 10, lines 251-252. Species other than NO played a significant role to explain the model's OH underestimation.

**Reply**

In the Reply to Question 14, we compared the composition of VOCs reactivity under the different NO intervals. We found the percentage of isoprene and OVOCs species (especially aldehydes and modeled OVOCs) in the total VOCs reactivity was higher in low NO regime. The contribution of isoprene and its oxidation products (MACR and MVK) to the missing OH source was evaluated. The detailed discussions were presented in the Reply to Question 14.

16. Page 13, Line 301, Table 1. AOC includes all combination of pollutants i and oxidant j (j= OH, $O_3$, $NO_3$)? As the equation (1) does not include j, this in unclear.

**Reply**

Yes, AOC includes all combination of pollutants (eg. VOCs, CO and $CH_4$) and oxidants (OH, $O_3$ and $NO_3$).

**Revision**

**Section 2.3:**

AOC includes all combination of pollutants $Y$ and oxidants $X$.

17. Page 14, Figure 7c. Why different units are used for the AOC and F(O₃)? They can be both in ppb h-1 for example and should have close values.

**Reply**

We revised the unit of AOC to ppb h$^{-1}$, which is the same as the unit of O$_3$ formation. Additionally, we have taken the helpful third comment in your 'General comments' into account, and we added the NO dependence of $P(O_3)$, AOC and the ratio of $P(O_3)$ to AOC in Fig. 7 (b-d).

**Revision**
**(1) Section 4.4:**

As the indictor for secondary pollution, net O$_3$ production rate, $P(O_3)$, can be calculated from the O$_3$ formation rate ($F(O_3)$) and O$_3$ loss rate ($L(O_3)$), as shown in Eq. (3-5) (Tan et al., 2017). The diurnal profiles of the speciation $F(O_3)$ and $L(O_3)$ were shown in Fig. S5 in the Supplementary Information. The diurnal maxima of the modeled $F(O_3)$ and $L(O_3)$ were 18.9 ppb h$^{-1}$ and 2.8 ppb h$^{-1}$, with the maximum $P(O_3)$ of 16.1 ppb h$^{-1}$ at 11:00. The modeled $P(O_3)$ was comparable to that in Wangdu site in summer and much higher than that in Beijing in winter (Tan et al., 2018; Tan et al., 2017).

$$F(O_3) = k_{HO_2+NO}[HO_2][NO] + \sum_i k_{RO_2i+NO}[RO_2]_i[NO] \tag{3}$$

$$L(O_3) = \theta j(O^1D)[O_3] + k_{O_3+OH}[O_3][OH] + k_{O_3+HO_2}[O_3][HO_2] + (\sum(k_{alkenes+O_3}^i[alkenes^i]))[O_3] \tag{4}$$

$$P(O_3) = F(O_3) - L(O_3) \tag{5}$$

where $\theta$ is the fraction of $O^1D$ from ozone photolysis that reacts with water vapor.

Herein, we presented the NO dependence of $P(O_3)$, AOC$_{VOCs}$, and ratio of $P(O_3)$ to AOC$_{VOCs}$ in Fig. 7 (b-d), in which AOC$_{VOCs}$ denotes the atmospheric oxidation capacity only from the VOCs oxidation. An upward trend $P(O_3)$ was presented with the increase of NO concentration when NO concentration was below 1 ppb, while a downward trend was shown with the increase of NO concentration when NO concentration was above 1 ppb. In terms of the NO dependence of AOC$_{VOCs}$, no significant variation was found, indicating VOCs oxidation was weakly impacted by NO concentrations in this campaign. Since AOC$_{VOCs}$ can represent the VOCs oxidant rate, and thus the ratio of $P(O_3)$ to AOC$_{VOCs}$ can reflect the yield of ozone production from VOCs oxidation. Similar to $P(O_3)$, the ratio increased with the increase of NO concentration when NO concentration was below 1 ppb. When NO concentration was above 1 ppb, the ratio decreased with the increase of NO concentration because NO$_2$ became the sink of OH radicals gradually. The maximum of the ratios existed when NO concentration was approximately 1 ppb, with a median of about 2, indicating the yield of ozone production from VOCs oxidation was about 2 in this study.

[Figure]

Figure 7: (a) The diurnal profiles of AOC in this campaign. (b) NO dependence of $P(O_3)$ during the daytime. (c) NO dependence of $AOC_{VOCs}$ during the daytime, and $AOC_{VOCs}$ denotes the atmospheric oxidation capacity only from the VOCs oxidation. (d) NO dependence of the ratio of $P(O_3)$ to $AOC_{VOCs}$ during the daytime. The box-whisker plots in (b-d) give the 10%, 25%, median, 75%, and 90% of $P(O_3)$, AOC and the ratio of $P(O_3)$ to AOC, respectively.

**(2) The diurnal profiles of $P(O_3)$, $F(O_3)$, and $L(O_3)$ were added in Figure S5 in the Supplementary Information:**

[Figure]

Figure S5: The diurnal profiles of $P(O_3)$, $F(O_3)$, and $L(O_3)$ in this campaign. The colored areas denote the speciation of $F(O_3)$ and $L(O_3)$ in the upper panel and lower panel, respectively. The black line denotes the $P(O_3)$, which is the discrepancy between $F(O_3)$ and $L(O_3)$. $MO_2$ denotes the methyl peroxy radicals. ALKAP, ALKEP and ISOP denote the $RO_2$ radicals derived from alkanes, alkenes and isoprene, respectively. $ACO_3$ denotes the acetyl peroxy radicals, and $RCO_3$ denotes the higher saturated acyl peroxy radicals.

18. Page 14, Line 322. Why a fixed value (9x10^-12) is used for the rate constants of the $RO_2$ + NO reactions? They should be variable in RACM depending of R and therefore these values should be used.

**Reply**

Thanks for your helpful suggestions. As shown in the Reply to Question 17, we revised the equations of $O_3$ formation and loss rates, and we re-calculated the $O_3$ formation rate, $O_3$ loss rate and net $O_3$ production rate based on the reactions of specific $RO_2$ speciation and NO.

19. Page 15, lines 351-352. Rewording is necessary.

**Reply**

As shown in the Reply to Question 11, we revised this sentence in the manuscript.

**Revision**

As for $HO_2$ radicals, the overestimation of $HO_2^*$ concentration was found, indicating that $HO_2$ heterogeneous uptake might make a significant role in $HO_2$ sinks.

20. Page 16, line 362. What are the gradients here?

**Reply**

As shown in the Reply to Question 17, we revised the contents of Section 4.4. Thus, the description of AOC in Section 5 was revised as follows.

**Revision**

In this campaign, AOC exhibited well-defined diurnal patterns, with a peak of 11.8 ppb h$^{-1}$. As expected, OH was the dominant oxidant accounting for 95.7% of the total AOC during the daytime. $O_3$ and $NO_3$ contributed 2.9% and 1.4% to total AOC during the daytime, respectively. The ratio of $P(O_3)$ to AOC$_{VOCs}$ trended to increase and then decrease as NO concentration increased, demonstrating the non-linear relationship between $O_3$ production and VOCs oxidation. The maximum of the ratios existed when NO concentration was approximately 1 ppb, with a median of about 2, indicating that the yield of ozone production from VOCs oxidation was about 2 in this campaign.

21. Page 16, line 363. makes the quantification of $F(O_3)$ achieved – rewording is necessary.

**Reply**

As shown in the Reply to Question 17 and Question 20, the description of AOC in Section 5 was revised as follows.

**Revision**

[revised manuscript text omitted]

---

## Author Comment (AC3)

**Response to Reviewer comment #3:**

**General comments:**

This paper presents measurements of OH and $HO_2$ radicals during the STORM campaign in the Pearl River Delta and compare their measurements to model predictions. The authors conclude that the model underestimates the measured OH concentration but can reproduce the measured HO2 concentrations. The authors propose that the "X" mechanism can explain the discrepancy, similar to that proposed in previous studies. The proposed mechanism involves an unmeasured species "X" that converts $RO_2$ to $HO_2$ and $HO_2$ to OH similar to NO. The authors conclude that a mixing ratio of "X" equivalent to 0.1 ppb of NO is needed to bring the measured OH concentrations into agreement with the measurements.

However, it is not clear that their measurements support their conclusion that the model significantly underestimates the measured concentrations, as it appears that the model agrees with the measurements to within the uncertainty of the technique. This is in contrast to the previous measurements highlighted in the paper, where the discrepancy between models and measurements were found to be much greater, such as the factor of 3-5 found by Hofzumahaus et al. (2009). While the addition of the X mechanism does improve the agreement with the measurements, there is no discussion as to why the measurements reported here are in better agreement with the model compared to the previous measurements discussed in the paper. The paper would benefit from an expanded discussion of the measurement-model agreement taking the uncertainties associate with both into account. In addition, the paper would benefit from an expanded discussion of a comparison of their results with the previous measurements mentioned in the manuscript, especially the difference between their measurements and those at the Backgarden and Heshan sites in the PRD (Hofzumahaus et al., 2009; Tan et al., 2019). Such a discussion could provide more information about the source of the model-measurement discrepancies at all these sites.

The measurements of OH and $HO_2$ appear to be high quality and are of interest to the atmospheric chemistry community. In addition to addressing the major comment described above, I believe the paper would be publishable after the authors also address the following in a revised manuscript.

**Reply**

Thanks for your critical comments and suggestions concerning our manuscript. We have studied all the comments which were helpful for revising and improving our manuscript. We have made corrections which we hope meet with approval. Below are our responses to the specific comments, highlighted in blue, with changes to the manuscript highlighted in green.

**Specific comments:**

1. The authors state that the base model agrees with the measurements to within the uncertainties of the measurements and the model (line 177), but then states that the model underestimates the measurements after 10 am when NO decreases. However, based on the information provided in Figure 3a, it appears that the model still agrees with the measurements to within the combined uncertainty of both the model and the measurements. This should be clarified. Addition of uncertainty estimates in Figure 3 would help to illustrate the agreement.

**Reply**

We recheck the data (details in the Reply to Question 4) and added the uncertainty of radical concentrations in fig. 3 as your suggestions. The description of the comparison between the observed and modeled radical concentrations was revised in Section 3.2.

**Revision**

**Section 3.2:**

The observed and modeled OH concentrations agreed within their 1-σ uncertainties of measurement and simulation (11% and 40%). However, when the NO mixing ratio (Fig. 2) dropped from 10:00 gradually, a systematic difference existed, with the observed OH concentration being about $1 \times 10^6$ cm$^{-3}$ higher than the modeled OH concentration.

[Figure]

Figure 3: (a-b) The diurnal profiles of the observed and modeled OH, $HO_2^*$ and $HO_2$ concentrations. (c) The diurnal profiles of the modeled $k_{OH}$. (d) The composition of the modeled $k_{OH}$. The red areas in (a-b) denote 1-σ uncertainties of the observed OH and $HO_2^*$ concentrations. The blue areas in (a-b) denote 1-σ uncertainties of the modeled OH and $HO_2^*$ concentrations, and the grey area in (b) denotes 1-σ uncertainties of the modeled $HO_2$ concentrations. The grey areas in (a-c) denote nighttime. ACD denotes acetaldehydes. ALD denotes the C3 and higher aldehydes. ACT and KET denote acetone and ketones. MACR and MVK denote methacrolein and methyl vinyl ketone.

2. Similarly, the base model predictions at low NO shown in Figure 5, although lower than the median measurements, appear to be within the combined uncertainty of model and measurements. The authors should quantify the discrepancy between the measurements and the model at each NO bin and reassess whether there is significant disagreement at low NO.

**Reply**

Thanks for your suggestions. The box-whisker plots in Fig. 5 denote the 10%, 25%, median, 75%, and 90% of HOx observations during the whole campaign rather than the 1-σ uncertainty of HOx observations. Herein, we compared the daily median of the observed and modeled OH concentrations with 1-σ uncertainty under the low NO intervals during the noontime (< 0.2 ppb, 0.2-0.6 ppb) in the following figure. The medium modeled OH concentrations were lower than the medium observed concentrations on most days. When we considered the combined uncertainty of OH observations (11%) and simulation (40%), the modeled OH concentrations were still lower than the OH observations on several days, especially when NO concentration was below 0.2 ppb.

[Figure]

Figure: The daily median of the observed and modeled OH concentration with 1-σ uncertainty under the low NO intervals (< 0.2 ppb, 0.2-0.6 ppb) during the noontime.

Additionally, we further explored the composition of VOCs reactivity under the different NO intervals, as shown in the revised Fig. 5.

**Revision**

**(1) Section 4.2.1:**

To further explore the influencing factors of OH underestimation, we presented the speciation VOCs reactivity under the different NO intervals, as shown in Fig. 5 and Table S4 in the Supplementary Information. The isoprene reactivity and total OVOCs reactivity (the sum of HCHO, ACD, ACT, ALD, KET, MACR, MVK and the modeled OVOCs) increased with the decrease of NO concentrations, while the anthropogenic VOCs reactivity (alkanes, alkenes and aromatics) was higher in high NO regime. Additionally, the $O_3$ concentration in low NO regime was significantly higher than those in high NO regime, and the temperature was slightly higher in low NO regime, demonstrating the photochemistry was more active in low NO regime in

this campaign. Overall, the photochemistry and composition of VOCs reactivity, especially the isoprene and OVOCs species (mainly ACD, ACT and the modeled OVOCs), might closely impact the missing OH sources.

[Figure]

Figure 5: NO dependence of OH and $HO_2^*$ radicals. The red box-whisker plots give the 10%, 25%, median, 75%, and 90% of the HOx observations. The blue circles show the median values of the HOx simulations by the base model, and the green circles show the HOx simulations by the model with X mechanism. Total VOCs reactivities and their organic speciation are presented by pie charts at the different NO intervals at the top. Only daytime values and NO concentration above the detection limit of the instrument were chosen. ACD and ACT denote acetaldehyde and acetone, respectively. ALD denotes the C3 and higher aldehydes. KET denotes ketones. MACR and MVK, which are both the isoprene oxidation products, denote methacrolein and methyl vinyl ketone, respectively.

**(2) The median values of meteorological and chemical parameters during the daytime at the different NO intervals were added in Table S4 in the Supplementary Information:**

Table S4: The median values of meteorological and chemical parameters during the daytime at the different NO intervals.

| parameters | NO interval (< 0.2 ppb) | NO interval (0.2-0.6 ppb) | NO interval (0.6-2 ppb) | NO interval (> 2 ppb) |
|---|---|---|---|---|
| Temperature / K | 301.4 | 300.8 | 299.1 | 297.9 |
| $j(O^1D)$ / $10^{-6}$ s$^{-1}$ | 4.7 | 8.9 | 8.2 | 7.4 |
| $O_3$ concentration / ppb | 71.7 | 55.1 | 39.6 | 16.9 |

| | | | | |
|---|---|---|---|---|
| Alkanes reactivity / s$^{-1}$ | 2.2 | 3.4 | 3.3 | 3.5 |
| Alkenes reactivity / s$^{-1}$ | 1.4 | 1.0 | 1.4 | 2.3 |
| Aromatics reactivity / s$^{-1}$ | 0.9 | 1.0 | 1.5 | 2.4 |
| Isoprene reactivity / s$^{-1}$ | 1.1 | 1.1 | 0.8 | 0.5 |
| HCHO reactivity / s$^{-1}$ | 1.1 | 0.9 | 0.8 | 0.7 |
| ACD reactivity / s$^{-1}$ | 1.1 | 1.4 | 1.3 | 1.2 |
| ACT reactivity / s$^{-1}$ | 0.2 | 0.2 | 0.2 | 0.2 |
| ALD reactivity / s$^{-1}$ | 1.9 | 1.8 | 1.6 | 1.2 |
| KET reactivity / s$^{-1}$ | 0.0 | 0.0 | 0.0 | 0.0 |
| MACR reactivity / s$^{-1}$ | 0.2 | 0.2 | 0.1 | 0.1 |
| MVK reactivity / s$^{-1}$ | 0.3 | 0.2 | 0.1 | 0.1 |
| Modeled OVOCs reactivity / s$^{-1}$ | 2.5 | 2.2 | 2.2 | 2.4 |
| Alkanes concentration / ppb | 15.0 | 16.8 | 19.0 | 24.6 |
| Alkenes concentration / ppb | 1.6 | 1.6 | 2.0 | 3.4 |
| Aromatics concentration / ppb | 3.3 | 3.3 | 4.8 | 7.9 |
| Isoprene concentration / ppb | 0.4 | 0.4 | 0.3 | 0.2 |
| HCHO concentration / ppb | 5.6 | 4.3 | 3.8 | 3.5 |
| ACD concentration / ppb | 3.0 | 3.7 | 3.5 | 3.3 |
| ACT concentration / ppb | 3.2 | 3.7 | 3.3 | 2.7 |
| ALD concentration / ppb | 3.8 | 3.6 | 3.2 | 2.5 |
| KET concentration / ppb | 0.3 | 0.4 | 0.4 | 0.3 |
| MACR concentration / ppb | 0.2 | 0.2 | 0.1 | 0.1 |
| MVK concentration / ppb | 0.5 | 0.5 | 0.3 | 0.2 |

3. The analysis of the OH measurements assumes that there are no interferences associated with the LIF-FAGE measurements. However, there is no discussion of whether the authors tested for unknown interferences with their measurements through a chemical modulation technique similar to that described in Tan et al. (2019). This should be addressed, as a significant interference would suggest that the model overestimation of OH could be more significant.

**Reply**

Thanks for your suggestions, the pre-injector system really did not be applied in this campaign, and it would introduce uncertainty into the OH measurement. However, it is believed that the interference in OH measurement in this campaign was negligible by analyzing the

PKU-LIF system and the environmental conditions during the campaign.

PKU-LIF system has been used to measure HOx concentrations since 2014. We used the pre-injector system to quantify the possible interferences for several campaigns, including the campaigns conducted in Wangdu site (Tan et al., 2017), Heshan site (Tan et al., 2019), Huairou site (Tan et al., 2018), Taizhou (Ma et al., 2022, in review, ACPD), and Chengdu site (Yang et al., 2021). No significant internal interference was found in the prior studies, demonstrating the accuracy of the PKU-LIF system has been determined several times.

Moreover, the potential interference may exist when the sampled air contained alkenes, ozone, and BVOCs (Mao et al., 2012; Fuchs et al., 2016; Novelli et al., 2014), indicating the environmental conditions, especially $O_3$, alkenes and isoprene, are important to the OH interferences. To further explore the potential interference in this campaign, we take Wangdu campaign as an example to compare the major environmental conditions between the prior campaigns and Shenzhen campaign here. During the Wangdu campaign, the chemical modulation tests were conducted on 29 June, 30 June, 02 July, and 05 July 2014, respectively (Tan et al., 2017). The daily mean $O_3$, alkenes (ethene, butadiene and other anthropogenic dienes, internal alkenes and terminal alkenes) and isoprene concentrations during the daytime on 29 June were 94.1, 3.8, 1.9 ppb, those on 30 June were 92.2, 2.7, and 1.9 ppb, those on 02 July were 52.9, 1.5, and 0.5 ppb, and those on 05 July were 68.5, 2.4, and 0.9 ppb. The $O_3$, alkenes and isoprene concentrations on 29 June were highest among those on 29 June, 30 June, 02 July and 05 July, and thus the potential interference on 29 June can be considered the highest among the four days. The results indicated that the potential interference during the daytime in Wangdu was negligible.

Here, we also showed the major parameters related to OH interference in Shenzhen in Table S2 in the Supplementary Information. The daily mean $O_3$ and isoprene concentrations during the daytime in Shenzhen were within 8.6-91.7 ppb and 0.1-1.0 ppb, which were both lower than those on 29 June in Wangdu. In terms of the alkenes, only 10,16-17 October 2018 in Shenzhen campaign were higher than that observed on 29 June in Wangdu, but the $O_3$ concentrations on the three days in Shenzhen were only 21.9, 13.9, and 8.6 ppb, and the isoprene concentrations on the three days in Shenzhen were only 0.3, 0.2, and 0.1 ppb, respectively. Overall, the environmental condition in Shenzhen was less conducive to generating potential OH interference than that in Wangdu. Therefore, it is not expected that OH measurement in this campaign was affected by the internal interference.

We have added the description of interval interference in Section 2.2 and the Supplementary Information.

Table S2: The daily mean $O_3$, alkenes (ethene, butadiene and other anthropogenic dienes, internal alkenes and terminal alkenes) and isoprene concentrations during the daytime (08:00-17:00) in the STORM campaign in this study.

| Date / Species | 10-05 | 10-06 | 10-07 | 10-08 | 10-09 | 10-10 | 10-11 | 10-12 | 10-13 | 10-14 | 10-15 | 10-16 |
|---|---|---|---|---|---|---|---|---|---|---|---|---|
| $O_3$ (ppb) | 81.4 | 83.8 | 91.7 | 86.7 | 48.1 | 21.9 | 30.2 | 42.6 | 46.8 | 38.7 | 40.2 | 13.9 |
| Alkenes (ppb) | 1.4 | 1.8 | 3.6 | 2.3 | 3.2 | 5.4 | 2.9 | 2.4 | 2.6 | 1.4 | 1.6 | 4.9 |
| Isoprene (ppb) | 0.4 | 0.4 | 0.4 | 0.5 | 0.4 | 0.3 | 0.1 | 0.3 | 0.4 | 0.4 | 0.6 | 0.2 |

| Date / Species | 10-17 | 10-18 | 10-19 | 10-20 | 10-21 | 10-22 | 10-23 | 10-24 | 10-25 | 10-26 | 10-27 | 10-28 |
|---|---|---|---|---|---|---|---|---|---|---|---|---|
| $O_3$ (ppb) | 8.6 | 16.2 | 39.4 | 45.8 | 47.2 | 25.2 | 40.9 | 36.5 | 55.2 | 56.5 | 60.9 | 60.8 |
| Alkenes (ppb) | 4.7 | 3.2 | 2.1 | 1.3 | 1.2 | 2.5 | 2.9 | 2.7 | 1.2 | 3.4 | 1.7 | 1.7 |
| Isoprene (ppb) | 0.1 | 0.1 | 0.5 | 0.3 | 0.8 | 0.8 | 0.4 | 0.3 | 0.5 | 1.0 | 0.6 | 0.8 |

**Revision**

**(1) Section 2.2:**

Additionally, prior studies reported that OH measurement might be affected by the potential interference, when the sampled air contained ozone, alkenes and BVOCs (Mao et al., 2012; Fuchs et al., 2016; Novelli et al., 2014), indicating the environmental conditions are important to the production of interference. The pre-injector is usually used to test the potential OH interference, and has been applied to our PKU-LIF system to quantify the possible interferences for several campaigns, including the campaigns conducted in Wangdu, Heshan, Huairou, Taizhou and Chengdu sites (Tan et al., 2017; Tan et al., 2019; Tan et al., 2018; Yang et al., 2021). No significant internal interference was found in the prior studies, demonstrating the accuracy of the PKU-LIF system has been determined for several times. Moreover, to further explore the potential interference in this campaign, we compared the major environmental conditions, especially $O_3$, alkenes and isoprene, between Shenzhen and Wangdu sites, as shown in the Supplementary Information. The environmental condition in Shenzhen was less conducive to generating interference than that in Wangdu, and the details were presented in the Supplementary Information. Therefore, it is not expected that the OH measurements in this campaign were affected by the internal interference.

**(2) The detailed information on potential OH interference was added in the Supplementary Information:**

We compared the environmental conditions in Shenzhen and Wangdu sites. The chemical modulation tests, which was applied to test the potential OH interference, were conducted on 29 June, 30 June, 02 July and 05 July 2014 in Wangdu (Tan et al., 2017). During the campaign in Wangdu, the daily mean $O_3$, alkenes (ethene, butadiene and other anthropogenic dienes, internal alkenes and terminal alkenes) and isoprene concentrations during the daytime on 29 June were 94.1, 3.8, 1.9 ppb, those on 30 June were 92.2, 2.7, and 1.9 ppb, those on 02 July were 52.9, 1.5, and 0.5 ppb, and those on 05 July were 68.5, 2.4, and 0.9 ppb, respectively. The $O_3$, alkenes and isoprene concentrations on 29 June were the highest among those on 29 June, 30 June, 02 July and 05 July, and thus the potential interference on 29 June can be considered the highest among the four days. The chemical modulation results indicated that the potential interference during the daytime in Wangdu was negligible (Tan et al., 2017).

As shown in Table S2, the $O_3$, alkenes and isoprene concentrations in Shenzhen were within 8.6-91.7 ppb, 1.2-5.4 ppb, and 0.1-1.0 ppb, respectively. The $O_3$ concentrations in Shenzhen (8.6-91.7 ppb) were lower than those on 29 June (94.1 ppb) and 30 June (92.2 ppb) in Wangdu. Similarly, the isoprene concentrations in Shenzhen (0.1-1.0 ppb) were also lower than those on

29 June (1.9 ppb) and 30 June (1.9 ppb) in Wangdu. In terms of the alkenes, only the concentrations on 10, 16-17 October 2018 (4.7-5.4 ppb) in Shenzhen were higher than that observed on 29 June (3.8 ppb) in Wangdu, but the $O_3$ concentrations on the three days in Shenzhen were only 21.9, 13.9, and 8.6 ppb, and the isoprene concentrations on the three days in Shenzhen were only 0.3, 0.2, and 0.1 ppb, respectively.

Overall, the environmental condition in Shenzhen was less conducive to generating potential OH interference than that in Wangdu. Therefore, it is not expected that OH measurement in this campaign was affected by the internal interference.

4. I assume that the higher NO flow that was used in the $HO_2$ measurements was required to increase the signal to allow for adjusting the laser wavelength given the failure of the reference cell. Were these measurements included in the data? While the authors claim that the NO concentrations were still low enough to minimize $RO_2$ conversion to OH, did the authors perform calibrations of some $RO_2$ conversion efficiencies to confirm this? What $HO_2$ to OH conversion efficiencies did these two NO flows correspond to? Providing more details on the potential for $RO_2$ interferences with the $HO_2$ measurements would improve the reader's confidence in the measurements.

**Reply**

Thanks for your helpful suggestions. We have rechecked all the data and made corrections in the revised manuscript.

**Revision:**

**(1) Section 2.2:**

In this campaign, NO mixing ratios were switched between 25 ppm (low NO mode) and 50 ppm (high NO mode). We calculated the $HO_2$ conversion rates under the two different NO concentrations by calibrating the PKU-LIF system. $HO_2$ conversion rates in low NO mode ranged within 80%-95%, while those in high NO mode were over 100%, demonstrating that the $HO_2$ measurement was affected by $RO_2$ radicals. Prior studies have reported the relative detection sensitivities ($\alpha_{RO_2}$) for the major $RO_2$ species, mainly from alkenes, isoprene and aromatics, when the $HO_2$ conversion rate was over 100% (Fuchs et al., 2011; Lu et al., 2012; Lu et al., 2013). Therefore, only the $HO_2$ observations in high NO mode were chosen and they were denoted as $[HO_2^*]$, which was the sum of the true $HO_2$ concentration and a systematic bias from the mixture of $RO_2$ species $i$ which were detected with different relative sensitivities $\alpha_{RO_2}^i$, as shown in Eq. (1) (Lu et al., 2012). The true $HO_2$ concentration was difficult to calculated due to the $RO_2$ concentration measurements and their speciation were not available. Herein, we simulated the $HO_2$ and $HO_2^*$ concentrations by the model. The interference from $RO_2$ was estimated to be the difference between the $HO_2$ and $HO_2^*$ concentrations.

$$[HO_2^*] = [HO_2] + \Sigma(\alpha_{RO_2}^i \times [RO_2]_i) \tag{1}$$

**(2) The figures and descriptions of $HO_2$ concentration were revised in Section 3.2:**

The diurnal maximum of the observed $HO_2^*$, the modeled $HO_2^*$ and the modeled $HO_2$ concentrations were $4.2\times10^8$ cm$^{-3}$, $6.1\times10^8$ cm$^{-3}$, and $4.4\times10^8$ cm$^{-3}$, respectively. The difference between the modeled $HO_2^*$ and $HO_2$ concentrations can be considered a modeled $HO_2$ interference from $RO_2$ (Lu et al., 2012). The $RO_2$ interference was small in the morning, while it became larger in the afternoon. It ranged within 23%-28% during the daytime (08:00-17:00), which was comparable with those in the Backgarden and Yufa sites in China, Borneo rainforest in Malaysia (OP3 campaign, aircraft), and UK (RONOCO campaign, aircraft) (Lu et al., 2012; Lu et al., 2013; Jones et al., 2011; Stone et al., 2014). The observed $HO_2^*$ was overestimated by the model, indicating the $HO_2$ heterogeneous uptake might have a significant impact during this campaign. The diurnal maximum of $HO_2^*$ concentration observed in Shenzhen was much lower than those observed in the Yufa and Backgarden sites (Hofzumahaus et al., 2009; Lu et al., 2012; Lu et al., 2013).

[Figure]

Figure 3: (a-b) The diurnal profiles of the observed and modeled OH, $HO_2^*$ and $HO_2$ concentrations. (c) The diurnal profiles of the modeled $k_{OH}$. (d) The composition of the modeled $k_{OH}$. The red areas in (a-b) denote 1-σ uncertainties of the observed OH and $HO_2^*$ concentrations. The blue areas in (a-b) denote 1-σ uncertainties of the modeled OH and $HO_2^*$ concentrations, and the grey area in (b) denotes 1-σ uncertainties of the modeled $HO_2$ concentrations. The grey areas in (a-c) denote nighttime. ACD denotes acetaldehydes. ALD denotes the C3 and higher aldehydes. ACT and KET denote acetone and ketones. MACR and MVK denote methacrolein and methyl vinyl ketone.

**(3) The timeseries of $HO_2$ concentrations were revised in Figure S1:**

[Figure]

Figure S1: Timeseries of the OH, $HO_2^*$, $HO_2$ concentrations and $k_{OH}$ in this study. The grey areas denote nighttime.

**(4) The observed HO₂ concentrations can influence the OH experimental budget, so the description in Section 4.1 was revised:**

It is noted that the OH production rate was overestimated because we used $HO_2^*$ concentrations instead of $HO_2$ concentrations here. Thus, the missing OH source was the lower limit here, demonstrating more unknown OH sources need to be further explored.

[Figure]

Figure 4: (a) The diurnal profiles of OH production and destruction rates and the proportions of different known sources in the calculated production rate during the daytime. The blue line denotes the OH destruction rate, and the colored areas denote the calculated OH production rates from the known sources. (b) The missing OH source which was the discrepancy between the OH destruction and production rates, and the OH production rate which was ten times the production rate derived from LIM1 mechanism. The grey areas denote nighttime.

**(5) The NO dependence of HOx radicals in Fig. 5 was revised:**

[Figure]

Figure 5: NO dependence of OH and $HO_2^*$ radicals. The red box-whisker plots give the 10%, 25%, median, 75%, and 90% of the HOx observations. The blue circles show the median values of the HOx simulations by the base model, and the green circles show the HOx simulations by the model with X mechanism. Total VOCs reactivity and their organic speciation are presented by pie charts at the different NO intervals at the top. Only daytime values and NO concentration above the detection limit of the instrument were chosen. ACD and ACT denote acetaldehyde and acetone, respectively. ALD denotes the C3 and higher aldehydes. KET denotes ketones. MACR and MVK, which are both the isoprene oxidation products, denote methacrolein and methyl vinyl ketone, respectively.

5.  The authors should clarify that the rate of ozone production shown in equation 2 (line 322) represents the gross instantaneous rate of ozone production rather than the net rate of ozone production, as it does not take into account any $NO_2$ formed that does not lead to $O_3$ production through the formation of $HNO_3$ from the $OH + NO_2$ reaction. In contrast Tan et al. (2017) appear to use the net rate of ozone production in their analysis of the chemistry at the Wangdu site. As a result, the comparison of the rate of ozone production between the sites shown in Figure 7c may not be an appropriate comparison. This should be clarified.

**Reply**

Thanks for your helpful suggestions. We calculated the net rate of ozone production and added the NO dependence of $P(O_3)$, AOC and the ratio of $P(O_3)$ to AOC in Fig. 7 (b-d).

**Revision**
**(1) Section 4.4:**

As the indictor for secondary pollution, net $O_3$ production rate, $P(O_3)$, can be calculated from

the $O_3$ formation rate ($F(O_3)$) and $O_3$ loss rate ($L(O_3)$), as shown in Eq. (3-5) (Tan et al., 2017). The diurnal profiles of the speciation $F(O_3)$ and $L(O_3)$ were shown in Fig. S5 in the Supplementary Information. The diurnal maxima of the modeled $F(O_3)$ and $L(O_3)$ were 18.9 ppb h$^{-1}$ and 2.8 ppb h$^{-1}$, with the maximum $P(O_3)$ of 16.1 ppb h$^{-1}$ at 11:00. The modeled $P(O_3)$ was comparable to that in Wangdu site in summer and much higher than that in Beijing in winter (Tan et al., 2018; Tan et al., 2017).

$$F(O_3) = k_{HO_2+NO}[HO_2][NO] + \sum_i k_{RO_2i+NO}[RO_2]_i[NO] \tag{3}$$

$$L(O_3) = \theta j(O^1D)[O_3] + k_{O_3+OH}[O_3][OH] + k_{O_3+HO_2}[O_3][HO_2] + (\sum(k^i_{alkenes+O_3}[alkenes^i]))[O_3] \tag{4}$$

$$P(O_3) = F(O_3) - L(O_3) \tag{5}$$

where $\theta$ is the fraction of $O^1D$ from ozone photolysis that reacts with water vapor.

Herein, we presented the NO dependence of $P(O_3)$, $AOC_{VOCs}$, and ratio of $P(O_3)$ to $AOC_{VOCs}$ in Fig. 7 (b-d), in which $AOC_{VOCs}$ denotes the atmospheric oxidation capacity only from the VOCs oxidation. An upward trend $P(O_3)$ was presented with the increase of NO concentration when NO concentration was below 1 ppb, while a downward trend was shown with the increase of NO concentration when NO concentration was above 1 ppb. In terms of the NO dependence of $AOC_{VOCs}$, no significant variation was found, indicating VOCs oxidation was weakly impacted by NO concentrations in this campaign. Since $AOC_{VOCs}$ can represent the VOCs oxidant rate, and thus the ratio of $P(O_3)$ to $AOC_{VOCs}$ can reflect the yield of ozone production from VOCs oxidation. Similar to $P(O_3)$, the ratio increased with the increase of NO concentration when NO concentration was below 1 ppb. When NO concentration was above 1 ppb, the ratio decreased with the increase of NO concentration because $NO_2$ became the sink of OH radicals gradually. The maximum of the ratios existed when NO concentration was approximately 1 ppb, with a median of about 2, indicating the yield of ozone production from VOCs oxidation was about 2 in this study.

[Figure]

Figure 7: (a) The diurnal profiles of AOC in this campaign. (b) NO dependence of $P(O_3)$ during the daytime. (c) NO dependence of $AOC_{VOCs}$ during the daytime, and $AOC_{VOCs}$ denotes the atmospheric oxidation capacity only from the VOCs oxidation. (d) NO

dependence of the ratio of $P(O_3)$ to $AOC_{VOCs}$ during the daytime. The box-whisker plots in (b-d) give the 10%, 25%, median, 75%, and 90% of $P(O_3)$, AOC and the ratio of $P(O_3)$ to AOC, respectively.

**(2) The diurnal profiles of $P(O_3)$, $F(O_3)$, and $L(O_3)$ were added in Figure S5 in the Supplementary Information:**

[Figure]

Figure S5: The diurnal profiles of $P(O_3)$, $F(O_3)$, and $L(O_3)$ in this campaign. The colored areas denote the speciation of $F(O_3)$ and $L(O_3)$ in the upper panel and lower panel, respectively. The black line denotes the $P(O_3)$, which is the discrepancy between $F(O_3)$ and $L(O_3)$. $MO_2$ denotes the methyl peroxy radicals. ALKAP, ALKEP and ISOP denote the $RO_2$ radicals derived from alkanes, alkenes and isoprene, respectively. $ACO_3$ denotes the acetyl peroxy radicals, and $RCO_3$ denotes the higher saturated acyl peroxy radicals.

**References**

[revised manuscript text omitted]

---

## Referee Report (RR1)

The authors have done a lot of work to improve the paper. My main concerns were on the quality of the data set, because in my opinion some major data were missing:

- Total OH reactivity: in the revised version, such measurements have appeared for a short period during the campaign. These data show a good agreement with the model, so I am really wondering why these data have not been shown in the original version of the manuscript. Anyway, these data allow now to get a better confidence in the data treatment concerning the OH losses.

- $HO_2$ measurements where done at very high NO concentrations for $HO_2$ conversion in order to use the $HO_2$ signal as reference cell for stabilizing the laser wavelength. In the revised version, the calibration of $HO_2$ with increasing added NO concentration has been discussed, and indeed under the "low" NO conditions the $HO_2$ conversion rate is up to 95%, while under the "high" NO conditions the conversion rate is over 100%, showing that $RO_2$ interference plays a role in both conditions. However, no $RO_2$ measurements are available from this campaign, and therefore the correction of the $HO_2^*$ signal can only be done based on modelled $RO_2$ concentrations and supposed $RO_2$ interference yields. It is not really clear to me, which $RO_2$ interference yields have been used for correction: did the authors measure it themselves or did they use data from Fuchs et al? This increases the uncertainty on the $HO_2$ concentration, and thus on the OH data as well.

- The OH concentration is underestimated by the model at low NO concentrations, and this disagreement has even increased since the $HO_2$ concentration has been corrected for $RO_2$ interference. The mysterious X species has been added to the model, and its concentration has been adjusted to bring into better agreement model and measurement. There is now some discussion on the concentration of this species and comparison with earlier studies. However, I am still regretting that the pre-injector system has not been used in this campaign to fully exclude any interference in the OH measurements. The argument that the PKU-LIF instrument has been proven free from interference in earlier campaigns, thus demonstrating the accuracy of the PKU-LIF system, does not fully convince me neither: looking for example to the Wangdu data, where a pre-injector has been used, some unexplained OH has been detected. There is a table in the paper giving the unexplained OH concentration together with NO and total OH concentrations. If one plots this unexplained OH (divided by total OH in order to normalize to overall photochemical activity) as a function of NO concentration, one gets a clear increase of the unexplained OH with decreasing NO:

[Figure]

In the new version, the authors at least mention that there have been reports on interferences in OH measurements when sampled air contains ozone, alkenes and BVOCs. They do not mention that also an interference has been detected in FAGE instruments due to ROOOH, the product of the reaction of $RO_2$ with OH (Atmos. Chem. Phys. 2019, 19, 349-362). In a very recent Science paper by Berndt et al. (Hydrotrioxide (ROOOH) formation in the atmosphere.

Science 2022, 376, 979-982) it is experimentally proven that ROOOH species have lifetimes of up to several hours, and that in low NO environments up to 1% of isoprene can be transformed to various ROOOH species. Even if $RO_2$ concentrations have not been measured during this campaign, the reaction of RO2 + OH should be added to the mechanism and the sum of modelled ROOOH concentrations should be compared against the modelled underestimation of OH concentration.

Finally, the authors have done a good job in improving the paper as good as possible, however, the data set (even if now somehow completed with a few days of OH lifetime measurements) is still lacking information to allow drawing solid conclusions and obtain new knowledge on atmospheric chemistry. I still have doubts that it is meeting the standard of ACP.

---

## Author Response (AR2)

**Response to Reviewer comment #1:**

**General comments:**

The authors have done a lot of work to improve the paper. My main concerns were on the quality of the data set, because in my opinion some major data were missing:

**Reply**

We gratefully thank you for all the suggestions which are helpful for the important guiding significance to us. Below are our responses to the specific comments, highlighted in blue, with changes to the manuscript highlighted in green.

**Specific comments:**

1. Total OH reactivity: in the revised version, such measurements have appeared for a short period during the campaign. These data show a good agreement with the model, so I am really wondering why these data have not been shown in the original version of the manuscript. Anyway, these data allow now to get a better confidence in the data treatment concerning the OH losses.

**Reply**

Thanks for your affirmation about the OH reactivity in the original comments.

2. $HO_2$ measurements where done at very high NO concentrations for $HO_2$ conversion in order to use the $HO_2$ signal as reference cell for stabilizing the laser wavelength. In the revised version, the calibration of $HO_2$ with increasing added NO concentration has been discussed, and indeed under the "low" NO conditions the $HO_2$ conversion rate is up to 95%, while under the "high" NO conditions the conversion rate is over 100%, showing that $RO_2$ interference plays a role in both conditions. However, no $RO_2$ measurements are available from this campaign, and therefore the correction of the $HO_2$* signal can only be done based on modelled $RO_2$ concentrations and supposed $RO_2$ interference yields. It is not really clear to me, which $RO_2$ interference yields have been used for correction: did the authors measure it themselves or did they use data from Fuchs et al? This increases the uncertainty on the $HO_2$ concentration, and thus on the OH data as well.

**Reply**

The $RO_2$ interference yields which have been used for correction were the reported data in Lu et al. (2012). Herein, the $RO_2$ species include methane, ethane, propane, n-butane, i-pentane, n-hexane, n-heptane, 2-methylhexane, 3-methylhexane, cyclohexane, ethene, propene, 1-butene, 1-pentene, cis-2-butene, trans-2-butene, trans-2-pentene, isoprene, MACR, MVK, benzene, toluene, ethylbenzene, o-xylene, m-xylene, p-xylene, styrene, and 1,2,4-trimethylbenzene (Lu et al., 2012). Lu et al. (2012) reported the experimental values and the modeled values simultaneously, and we used the modeled values by MCMv3.2 here. Herein, we further added the description of the $RO_2$ interference yields in Section 2.2 in the revised manuscript.

**Revision**

The RO$_2$ interference yields which was used for correction were the modeled values reported by Lu et al. (2012).

3.  The OH concentration is underestimated by the model at low NO concentrations, and this disagreement has even increased since the HO$_2$ concentration has been corrected for RO$_2$ interference. The mysterious X species has been added to the model, and its concentration has been adjusted to bring into better agreement model and measurement. There is now some discussion on the concentration of this species and comparison with earlier studies. However, I am still regretting that the pre-injector system has not been used in this campaign to fully exclude any interference in the OH measurements. The argument that the PKU-LIF instrument has been proven free from interference in earlier campaigns, thus demonstrating the accuracy of the PKU-LIF system, does not fully convince me neither: looking for example to the Wangdu data, where a pre-injector has been used, some unexplained OH has been detected. There is a table in the paper giving the unexplained OH concentration together with NO and total OH concentrations. If one plots this unexplained OH (divided by total OH in order to normalize to overall photochemical activity) as a function of NO concentration, one gets a clear increase of the unexplained OH with decreasing NO:

[Figure]

In the new version, the authors at least mention that there have been reports on interferences in OH measurements when sampled air contains ozone, alkenes and BVOCs. They do not mention that also an interference has been detected in FAGE instruments due to ROOOH, the product of the reaction of RO$_2$ with OH (Atmos. Chem. Phys. 2019, 19, 349-362). In a very recent Science paper by Berndt et al. (Hydrotrioxide (ROOOH) formation in the atmosphere. Science 2022, 376, 979-982) it is experimentally proven that ROOOH species have lifetimes of up to several hours, and that in low NO environments up to 1% of isoprene can be transformed to various ROOOH species. Even if RO$_2$ concentrations have not been measured during this campaign, the reaction of RO$_2$ + OH should be added to the mechanism and the sum of modelled ROOOH concentrations should be compared against the modelled underestimation of OH concentration.

**Reply**

Thanks for your helpful suggestions. The data presented in the above figure were from 13:00-15:00 on 29 June, 14:40-16:10 on 30 June, 16:30-17:40 on 05 July, and 18:00-21:00 on 05 July

in Wangdu campaign (Tan et al., 2017). The isoprene and $O_3$ concentrations varied within 1.4-2.8 ppb and 77-126 ppb, respectively. Herein, to compare the environmental conditions between Wangdu campaign and Shenzhen campaign, we presented the mean $O_3$ and isoprene concentrations in Shenzhen campaign during the corresponding period in Wangdu campaign (13:00-15:00, 14:40-16:10, 16:30-17:40 and 18:00-21:00) in the below table. The mean isoprene concentrations were 0.1-1.2 ppb, 0.1-0.7 ppb, 0.1-1.2 ppb, and 0.1-1.1 ppb in the four periods, which were all lower than the corresponding periods of Wangdu campaign. In terms of the $O_3$ concentrations, the mean $O_3$ concentrations were 11-109 ppb, 13-119 ppb, 8-116 ppb and 3-71 ppb in the four periods. The $O_3$ concentrations in Shenzhen during the 13:00-15:00 period and the 18:00-21:00 period were all lower than those during the corresponding periods in Wangdu. As for the $O_3$ concentration during the 14:40-16:10 period and the 16:30-17:40 period, only the $O_3$ concentration on 07-08 October were higher than those during the corresponding periods in Wangdu, but the isoprene concentrations on 07-08 October in Shenzhen were significantly lower than those in Wangdu. Overall, Tan et al. (2017) reported that the unknown OH interference have a minor impact on the daytime OH measurements. Thus, we can believe that the OH interference have a minor impact on the daytime OH observations from the perspective of the environmental conditions in this study by comparing the isoprene and $O_3$ concentrations in Shenzhen and Wangdu.

Table: The mean $O_3$ and isoprene concentrations in Shenzhen campaign during the corresponding period of Wangdu campaign (13:00-15:00, 14:40-16:10, 16:30-17:40 and 18:00-21:00).

| Date / Species | 10-05 | 10-06 | 10-07 | 10-08 | 10-09 | 10-10 | 10-11 | 10-12 | 10-13 | 10-14 | 10-15 | 10-16 |
|---|---|---|---|---|---|---|---|---|---|---|---|---|
| $O_3$ (ppb) 13:00-15:00 | 86 | 98 | 109 | 109 | 63 | 32 | 34 | 54 | 62 | 43 | 43 | 25 |
| $O_3$ (ppb) 14:40-16:10 | 82 | 98 | 98 | 119 | 70 | 32 | 36 | 48 | 61 | 41 | 43 | 18 |
| $O_3$ (ppb) 16:30-17:40 | 83 | 94 | 116 | 81 | 49 | 22 | 38 | 37 | 46 | 40 | 36 | 8 |
| $O_3$ (ppb) 18:00-21:00 | 42 | 57 | 71 | 61 | 27 | 22 | 27 | 15 | 38 | 37 | 30 | 11 |
| Isoprene (ppb) 13:00-15:00 | 0.4 | 0.4 | 0.6 | 0.4 | 0.5 | 0.4 | 0.1 | 0.4 | 0.5 | 0.5 | 0.4 | 0.3 |
| Isoprene (ppb) 14:40-16:10 | 0.6 | 0.4 | 0.5 | 0.4 | 0.4 | 0.3 | 0.2 | 0.2 | 0.3 | 0.3 | 0.4 | 0.2 |
| Isoprene (ppb) 16:30-17:40 | 1.2 | 0.2 | 0.2 | 0.2 | 0.3 | 0.2 | 0.1 | 0.1 | 0.3 | 0.2 | 0.2 | 0.1 |
| Isoprene (ppb) 18:00-21:00 | 1.1 | 0.1 | 0.1 | 0.1 | 0.2 | 0.1 | 0.1 | 0.1 | 0.1 | 0.1 | 0.1 | 0.1 |

| Date / Species | 10-17 | 10-18 | 10-19 | 10-20 | 10-21 | 10-22 | 10-23 | 10-24 | 10-25 | 10-26 | 10-27 | 10-28 |
|---|---|---|---|---|---|---|---|---|---|---|---|---|
| $O_3$ (ppb) 13:00-15:00 | 11 | 25 | 50 | 54 | 54 | 15 | 62 | 48 | 60 | 95 | 77 | 78 |
| $O_3$ (ppb) 14:40-16:10 | 13 | 27 | 42 | 54 | 51 | 19 | 65 | 49 | 62 | 74 | 77 | 72 |

| | | | | | | | | | | | |
|---|---|---|---|---|---|---|---|---|---|---|---|
| $O_3$ (ppb) 16:30-17:40 | 13 | 26 | 47 | 46 | 38 | 14 | 47 | 45 | 51 | 56 | 70 | 63 |
| $O_3$ (ppb) 18:00-21:00 | 3 | 14 | 37 | 33 | 41 | 2 | 29 | 33 | 28 | 20 | 44 | 24 |
| Isoprene (ppb) 13:00-15:00 | 0.1 | 0.2 | 0.6 | 0.3 | 1.4 | 0.8 | 0.5 | 0.4 | 0.5 | 1.1 | 0.8 | 1.2 |
| Isoprene (ppb) 14:40-16:10 | 0.1 | 0.1 | 0.3 | 0.2 | 0.8 | 0.6 | 0.7 | 0.2 | 0.3 | 0.7 | 0.6 | 0.7 |
| Isoprene (ppb) 16:30-17:40 | 0.1 | 0.1 | 0.1 | 0.1 | 0.5 | 0.3 | 0.3 | 0.2 | 0.2 | 0.5 | 0.5 | 0.6 |
| Isoprene (ppb) 18:00-21:00 | 0.1 | 0.1 | 0.1 | 0.1 | 0.1 | 0.3 | 0.1 | 0.1 | 0.1 | 0.1 | 0.2 | 0.2 |

Besides the environmental conditions, the prior studies reported that the product of the reactions of $RO_2$ with OH, trioxides (ROOOH), might lead to an OH interference signal. The reactions of $RO_2$ radicals with OH radicals might be competitive with other sinks for $RO_2$ radicals (Fittschen, 2019;Fittschen et al., 2019;Berndt et al., 2022). However, Fittschen et al. (2019) reported that the ROOOH interference is highly dependent on the design and measurement conditions of different FAGE instruments. Therefore, we integrated the reactions of the ROOOH production and destruction into the base model herein, as shown in Eq. (1-2).

$$RO_2 + OH \rightarrow ROOOH \tag{1}$$

$$ROOOH \rightarrow RO + HO_2 \tag{2}$$

where the $RO_2$ is across all $RO_2$ radicals in the model excluding methyl peroxy radicals, for which it has been shown that the production of a trioxide species is only a minor product channel while the trioxide yield is expected to be close to 1 for larger peroxy radicals. The rate constant of Eq. (1) is $1.5 \times 10^{-10}$ cm$^3$ s$^{-1}$ (Fittschen et al., 2019). In Eq. (2), the rate constant is $10^{-4}$ s$^{-1}$, leading to ROOOH lifetimes of around 3 h, of the same order as the lifetime of ROOH species (Fittschen et al., 2019).

Figure. S1 (a) presents the modeled ROOOH concentrations during this campaign, and the maximum ROOOH concentration was $4.4 \times 10^9$ cm$^{-3}$. The correlation of the modeled ROOOH concentrations and the ratios of OH observations to OH simulations, and the correlation of the modeled ROOOH concentrations and the difference between OH observations and simulations both demonstrated that no significant relevance between ROOOH and the underestimation of OH radicals, as shown in Fig. S1 (b-c). Additionally, the ROOOH values modeled in our another campaign (Taizhou, 2018) were comparable to or even slightly higher than the simulations in this study, and the chemical modulation tests in Taizhou confirmed the ROOOH is not a significant OH interference in our PKU-LIF system (Ma et al., 2022).

We have added the ROOOH interference in Section 2.2 in the revised manuscript and the revised Supplementary Information.

[Figure]

Figure S1: (a) The timeseries of ROOOH concentration from the reactions of $RO_2$ radicals excluding methyl peroxy radicals with OH radicals. (b) The correlation of the modeled ROOOH concentrations and the ratios of OH observations to OH simulations. (c) The correlation of the modeled ROOOH concentrations and the difference between OH observations and simulations. Only daytime values were chosen in (b-c).

**Revision**
**(1) Section 2.2**

Besides the environmental conditions, the prior studies reported that the product of the reaction of $RO_2$ with OH, trioxides (ROOOH), might lead to an OH interference signal. The reactions of $RO_2$ radicals with OH radicals might be competitive with other sinks for $RO_2$ radicals (Fittschen, 2019;Fittschen et al., 2019;Berndt et al., 2022). However, Fittschen et al. (2019) reported that the ROOOH interference is highly dependent on the design and measurement conditions of different FAGE instruments. Therefore, we integrated the reactions of the ROOOH production and destruction into the base model herein, with the ROOOH production rate constant of $1.5 \times 10^{-10}$ $cm^3$ $s^{-1}$ and the destruction rate constant of $10^{-4}$ $s^{-1}$ (the details are presented in the Supplementary Information) (Fittschen et al., 2019). Figure. S1 (a) presents the modeled ROOOH concentrations during this campaign, with a maximum of about $4.4 \times 10^9$ $cm^{-3}$. The correlation of the modeled ROOOH concentrations and the ratios of OH observations to OH simulations, and the correlation of the modeled ROOOH concentrations and the difference between OH observations and simulations both demonstrated that no significant relevance between ROOOH and the underestimation of OH radicals, as shown in Fig. S1 (b-c). Additionally, the ROOOH values modeled in our another campaign (Taizhou, 2018) were comparable to or even slightly higher than the simulations in this study, and the chemical modulation tests in Taizhou confirmed the ROOOH is not a significant OH

interference in our PKU-LIF system (Ma et al., 2022).

Overall, the OH interference during this campaign was negligible according to the analysis of the behavior of PKU-LIF system in previous campaigns, the comparison of environmental conditions between this campaign and Wangdu campaign, and the exploration of the impact of ROOOH on the discrepancy of OH observations and simulations. However, we should acknowledge that the unmeasured interference might have an effect on radical measurement. More precise chemical modulation tests are needed in the future.

**(2) Figure S1 in the Supplementary Information**

To evaluate the impact of the interference from ROOOH on radical concentrations, we integrated the reactions of the ROOOH production and destruction into the base model herein, as shown in Eq. (S1-S2).

$$RO_2 + OH \rightarrow ROOOH \tag{S1}$$

$$ROOOH \rightarrow RO + HO_2 \tag{S2}$$

where the $RO_2$ is across all $RO_2$ radicals in the model excluding methyl peroxy radicals, for which it has been shown that the production of a trioxide species is only a minor product channel while the trioxide yield is expected to be close to 1 for larger peroxy radicals. The rate constant of Eq. (S1) is $1.5 \times 10^{-10}$ cm$^3$ s$^{-1}$ (Fittschen et al., 2019). In Eq. (S2), the rate constant is $10^{-4}$ s$^{-1}$, leading to ROOOH lifetimes of around 3 h, of the same order as the lifetime of ROOH species (Fittschen et al., 2019).

[Figure]

Figure S1: (a) The timeseries of ROOOH concentration from the reactions of $RO_2$ radicals excluding methyl peroxy radicals with OH radicals. (b) The correlation of the modeled ROOOH concentrations and the ratios of OH observations to OH simulations. (c) The correlation of the

modeled ROOOH concentrations and the difference between OH observations and simulations. Only daytime values were chosen in (b-c).


Here, we applied the two γ (0.2 and 0.08), which have been used in the model, to evaluate the impact of $HO_2$ uptake on radical concentrations, as shown in Fig. 6. The modeled $HO_2^*$ cannot match well with the observations when γ of 0.08 and 0.2 was set in the model. As the γ increased to approximately 0.3, good agreement between the modeled and observed $HO_2^*$ concentration was achieved, demonstrating that the significant heterogeneous uptake might exist in this campaign.

It is noted that the estimated strong influence is speculative because of the uncertainties of measurements and simulations. Overall, the γ evaluated in this study was comparable with those observed at the Mt. Tai and Mt. Mang in China, and Kyoto and Yokohama in Japan.

[Figure]

Figure 6: The diurnal profiles of the observed and modeled radical concentrations. The red and blue areas denote 1-σ uncertainties of measured and simulated radical concentrations by the base model, respectively. The orange, purple and black lines denote the simulations by the model which added the $HO_2$ heterogeneous uptake with different uptake coefficient. The grey areas denote nighttime.

**(2) Conclusion**

As for $HO_2$ radicals, the overestimation of $HO_2^*$ concentration was found, indicating that $HO_2$ heterogeneous uptake with the effective uptake coefficient of 0.3 might make a significant role in $HO_2$ sinks.

**(3) Abstract**

A significant $HO_2$ heterogeneous uptake was found in this study, with an effective uptake coefficient of 0.3.

3. Previous specific point #11:

The authors stated "over 100% conversion rate" of $HO_2$ in the reply and in L116 of the ATC version. I believe this measurement of the conversion efficiency is done in the environment where $HO_2$ is only present (i.e., $RO_2$ is not) and wonder how this over 100% conversion is measured.

**Reply**

The measurement of the conversion efficiency is done in the environment where $RO_2$ is not present. We calculated the $HO_2$ conversion by the calibration source. In the calibration experiments of PKU-LIF, we set two different modes, which are HOx mode and $HO_2$ mode. Photolysis of water molecules at 185 nm leads to the production of OH radical and H atom and it is generally assumed that the H-atoms are completely converted to $HO_2$ radical. In this mode, equal amounts of OH and $HO_2$ radicals are present simultaneously, and it is called HOx mode. The OH signal ($C^*$) includes the OH from the water photolysis ($C_1$) and the OH which is converted by $HO_2$ and NO in the detection cell ($C_2$).

In another mode, CO is added as an OH scavenger, in order to convert all OH to $HO_2$ between photolytical generation and intake into the detection cell, which is called $HO_2$ mode. In this mode, OH is converted to $HO_2$ by CO in the calibration cell. The amounts of $HO_2$ from the water photolysis and the $HO_2$ converted by OH via CO are equal, which are equal to the OH from the water photolysis separately. The total OH signal ($C^\#$) in this mode denotes the sum OH which is converted by the HO2 radicals from the water photolysis and by the HO2 radicals from the reaction of OH and CO, and thus the OH signal which is converted by $HO_2$ from the water photolysis is $C^\#/2$.

The $HO_2$-to-OH conversion efficiency ($V$) can be denoted by the ratio of the OH signal which is converted by $HO_2$ and NO in the detection cell ($C_2$) to the $HO_2$ from the water photolysis which is equal to the OH from the water photolysis ($C_1$). Besides, $C_2$ in HOx mode is equal to $C^\#/2$ in $HO_2$ mode, and $C_1$ in HOx mode is equal to the difference between $C^*$ and $C_2$. Therefore, the conversion efficiency ($V$) is calculated according to the Eq. (1).

$$V = \frac{C_2}{C_1} = \frac{C^\#/2}{C^* - C_2} = \frac{C^\#/2}{C^* - C^\#/2} \tag{1}$$

In this campaign, NO mixing ratios were switched between 25 ppm (low NO mode) and 50 ppm (high NO mode). We calculated the $HO_2$-to-OH conversion efficiency in low and high NO modes according to Eq. (1). In high NO mode, the value of $C_2$ was larger than the value of $C_1$, indicating the reactions of $RO_2$ and NO lead to the additional OH signal.

In the manuscript, we stated 'over 100% conversion rate' was not accurate, and we replaced the expression with 'while those in high NO mode reached 100%'.

**Revision**

$HO_2$-to-OH conversion efficiencies in low NO mode ranged within 80%-95%, while those in high NO mode reached 100%, demonstrating that the high NO concentration is sufficiently to achieve complete $HO_2$ to OH conversion and thus the $HO_2$ measurement was affected by $RO_2$ radicals.

4. Previous specific point #17:

Though defined in L508, it is not clear if OVOCs (and CO) are included in the AOC_VOCs or not. This will affect the interpretation of the ozone yield per VOCs oxidation as about 2.

**Reply**

In the previous response, the $AOC_{VOCs}$ include the AOC derived from the oxidation of CO and OVOCs besides VOCs. Herein, we redefine the $AOC_{VOCs}$, which denote the channels of primary VOCs (excluding OVOCs, and mainly alkanes, alkenes, aromatics and isoprene) with OH radicals. The figure 7 (c-d) and the related descriptions were revised in the revised manuscript.

**Revision**

Herein, we presented the NO dependence on $P(O_3)_{net}$, $AOC_{VOCs}$, and the ratio of $P(O_3)_{net}$ to $AOC_{VOCs}$ in Fig. 8 (b-d), in which $AOC_{VOCs}$ denotes the atmospheric oxidation capacity only from the VOCs oxidation, which includes the channels of primary VOCs (excluding OVOCs, and mainly alkanes, alkenes, aromatics and isoprene) with OH radicals.

An upward trend of $P(O_3)_{net}$ was presented with the increase of NO concentration when NO concentration was below 1 ppb, while $P(O_3)_{net}$ decreased with the increase of NO concentration because $NO_2$ became the sink of OH radicals gradually when NO concentration was above 1 ppb. In terms of the NO dependence on $AOC_{VOCs}$, no significant variation was found, indicating VOCs oxidation was weakly impacted by NO concentrations in this campaign. Since $AOC_{VOCs}$ can represent the VOCs oxidant rate, and thus the ratio of $P(O_3)_{net}$ to $AOC_{VOCs}$ can reflect the yield of net ozone production from VOCs oxidation. Similar to $P(O_3)_{net}$, the ratios increased with the increase of NO concentration when NO concentration was below 1 ppb, while the ratios decreased with the increase of NO concentration when NO concentration was above 1 ppb, indicating the yield of net $O_3$ production from VOCs oxidation would be lower within the low NO regime (< 1 ppb) and high NO regime (> 1 ppb). The median ratios ranged from 1.0 to 4.5, and the maximum of the median ratios existed when NO concentration was approximately 1 ppb, with a value of approximately 4.5. The nonlinear response of the yield of net ozone production to NO indicated that it is necessary to optimize the NOx and VOC control strategies for the reduction of $O_3$ pollution effectively.

[Figure]

Figure 8: (a) The diurnal profiles of AOC in this campaign. (b) NO dependence on $P(O_3)_{net}$ during the daytime. (c) NO dependence on $AOC_{VOCs}$ during the daytime, and $AOC_{VOCs}$ denotes the atmospheric oxidation capacity only from the VOCs oxidation. (d) NO dependence on the ratio of $P(O_3)_{net}$ to $AOC_{VOCs}$ during the daytime. The box-whisker plots in (b-d) give the 10%, 25%, median, 75%, and 90% $P(O_3)_{net}$, $AOC_{VOCs}$ and the ratio of $P(O_3)_{net}$ to $AOC_{VOCs}$, respectively.

5.   Previous specific point #17:

The new statement from L514 (ATC version), "When NO concentration was above 1 ppb, the ratio decreased with the increase of NO concentration because $NO_2$ became the sink of OH radicals gradually." is wrong. The statement would be valid with AOC, but is not with AOC_VOC.

**Reply**

Thanks for your helpful suggestions. We revised the description of figure 7 (b-d) as follows.

**Revision**

An upward trend of $P(O_3)_{net}$ was presented with the increase of NO concentration when NO concentration was below 1 ppb, while $P(O_3)_{net}$ decreased with the increase of NO concentration because $NO_2$ became the sink of OH radicals gradually when NO concentration was above 1 ppb. In terms of the NO dependence on $AOC_{VOCs}$, no significant variation was found, indicating VOCs oxidation was weakly impacted by NO concentrations in this campaign. Since $AOC_{VOCs}$ can represent the VOCs oxidant rate, and thus the ratio of $P(O_3)_{net}$ to $AOC_{VOCs}$ can reflect the yield of net ozone production from VOCs oxidation. Similar to $P(O_3)_{net}$, the ratios increased with the increase of NO concentration when NO concentration was below 1 ppb, while the ratios decreased with the increase of NO concentration when NO concentration was above 1 ppb, indicating the yield of net $O_3$ production from VOCs oxidation would be lower within the low NO regime (< 1 ppb) and high NO regime (> 1 ppb). The median ratios ranged from 1.0 to 4.5, and the maximum of the

median ratios existed when NO concentration was approximately 1 ppb, with a value of approximately 4.5. The nonlinear response of the yield of net ozone production to NO indicated that it is necessary to optimize the NOx and VOC control strategies for the reduction of O$_3$ pollution effectively.

**Reply**

As your suggestions, we added the NO dependence on the HOx observed-to-modeled ratio to provide a better illustration of the comparison between HOx observations and simulations in Section 4.2.1.

As for the comparison of missing OH sources between this study and other campaigns, we added some discussion in Section 4.2.2.

**Revision**

**(1) Section 4.2.1**

The NO dependence on observed and modeled HOx concentrations and the NO dependence on HOx observed-to-modeled ratios were illustrated in Fig. 5 and Fig. S4.

[Figure]

Figure S4: NO dependence on the ratios of HOx observations to simulations for daytime conditions. The vertical lines denote the combined uncertainty from radical measurements and model calculations via error propagation.

**(2) Section 4.2.2**

Compared to the Shenzhen site, the X concentration in the Backgarden and Heshan sites in PRD was higher, which might be affected by the different air masses in the three sites. The $k_{OH}$ in the Shenzhen site was much lower than those in Backgarden and Heshan sites, and a weaker variation of $k_{OH}$ in Shenzhen was observed. Under the influence of the East Asian monsoon, the prevailing wind for the PRD area is mostly southerly during the summer months and mostly northerly during the winter months (Fan et al., 2005;Zhang et al., 2008). The Backgarden site is located in Guangzhou, and the Heshan site is located in Jiangmen. The two cities are along the north-south axis, and thus the air masses of the Backgarden and Heshan sites are intimately linked with each other, while the air mass in Shenzhen is more similar to Hongkong (Zhang et al., 2008).

Compared to the VOCs reactivity in the air mass at Backgarden and Yufa sites reported by Lu et al. (2013), lower isoprene reactivity and OVOCs reactivity were observed in Shenzhen site. As discussed in Section 4.2.1, the OH underestimation might be closely related to the composition of VOCs reactivity. Therefore, further exploration of this unclassical OH recycling is needed to improve our understanding of radical chemistry, especially the mechanisms related to isoprene and OVOCs.

2. Related to this, it is also not clear whether the difference between the modeled loss of OH and the measured rate of production illustrated in Figure 4 is significant. The authors should add the estimated and propagated uncertainty associated with the modeled loss and calculated production in order to demonstrate that their proposed missing OH source is significant.

**Reply**

Thanks for your helpful suggestions. We added the uncertainty in Fig. 4, and revised the

description of Fig. 4.

**Revision**

As shown in Fig. 4 (b), the discrepancy between the OH production and destruction rates at around 11:00-15:00, which was approximately of (3.1~4.6) ppb h$^{-1}$, cannot be explained by the combined experimental uncertainties. The discrepancy was attributed to the missing OH sources because $k_{OH}$ was constrained in this study.

[Figure]

Figure 4: (a) The diurnal profiles of OH production and destruction rates and the proportions of different known sources in the calculated production rate during the daytime. The blue line denotes the OH destruction rate, and the colored areas denote the calculated OH production rates from the known sources. (b) The missing OH source which was the discrepancy between the OH destruction and production rates, and the OH production rate which was ten times the production rate derived from LIM1 mechanism. The red shaded areas denote the combined uncertainty from the experimental errors of the measured quantities (Table S1) and the reaction rate coefficients. The grey areas denote nighttime.

3. The potential for interferences in the OH measurements still needs to be considered. Unfortunately, I misspoke in my previous review regarding the impact of potential interferences on the agreement between the modeled and measured OH. Since the authors did not test for interferences, the measured OH concentrations should be considered an upper limit to the actual OH concentrations, and the presence of an interference could explain the discrepancy between the measurements and the model. While the authors provide some evidence that an interference from the ozonolysis of alkenes may not have impacted their measurements, other interferences may have impacted their measurements such as the decomposition of trioxides inside their detection cell, especially under low NO

conditions (see Fittschen et al., Atmos. Chem. Phys., 19, 349-362, 2019). While it may be true that the measurements are free from interferences, the authors should at least acknowledge that unmeasured interferences could contribute to the discrepancy with the model results.

**Reply**

Thanks for your suggestions, we conducted sensitivity experiments to explore the impact of trioxides (ROOOH) on radical concentrations. Herein, we integrated the reactions of the ROOOH production and destruction into the base model, as shown in Eq. (1-2).

$$RO_2 + OH \rightarrow ROOOH \tag{1}$$

$$ROOOH \rightarrow RO + HO_2 \tag{2}$$

where the $RO_2$ is across all $RO_2$ radicals in the model excluding methyl peroxy radicals, for which it has been shown that the production of a trioxide species is only a minor product channel while the trioxide yield is expected to be close to 1 for larger peroxy radicals. The rate constant of Eq. (1) is $1.5 \times 10^{-10}$ cm$^3$ s$^{-1}$ (Fittschen et al., 2019). In Eq. (2), the rate constant is $10^{-4}$ s$^{-1}$, leading to ROOOH lifetimes of around 3 h, of the same order as the lifetime of ROOH species (Fittschen et al., 2019).

Figure. S1 (a) presents the modeled ROOOH concentrations during this campaign, and the maximum ROOOH concentration was $4.4 \times 10^9$ cm$^{-3}$. The correlation of the modeled ROOOH concentrations and the ratios of OH observations to OH simulations, and the correlation of the modeled ROOOH concentrations and the difference between OH observations and simulations both demonstrated that no significant relevance between ROOOH and the underestimation of OH radicals, as shown in Fig. S1 (b-c). Additionally, the ROOOH values modeled in our another campaign (Taizhou, 2018) were comparable to or even slightly higher than the simulations in this study, and the chemical modulation tests in Taizhou confirmed the ROOOH is not a significant OH interference in our PKU-LIF system (Ma et al., 2022).

We have added the ROOOH interference in Section 2.2 in the revised manuscript and the Supplementary Information. Besides, we should note that the unmeasured interferences could contribute to the discrepancy between the radical observations and simulations.

[Figure]

Figure S1: (a) The timeseries of ROOOH concentration from the reactions of $RO_2$ radicals excluding methyl peroxy radicals with OH radicals. (b) The correlation of the modeled ROOOH concentrations and the ratios of OH observations to OH simulations. (c) The correlation of the modeled ROOOH concentrations and the difference between OH observations and simulations. Only daytime values were chosen in (b-c).

**Revision**

**(1) Section 2.2**

Besides the environmental conditions, the prior studies reported that the product of the reaction of $RO_2$ with OH, trioxides (ROOOH), might lead to an OH interference signal. The reactions of $RO_2$ radicals with OH radicals might be competitive with other sinks for $RO_2$ radicals (Fittschen, 2019;Fittschen et al., 2019;Berndt et al., 2022). However, Fittschen et al. (2019) reported that the ROOOH interference is highly dependent on the design and measurement conditions of different FAGE instruments. Therefore, we integrated the reactions of the ROOOH production and destruction into the base model herein, with the ROOOH production rate constant of $1.5\times10^{-10}$ $cm^3$ $s^{-1}$ and the destruction rate constant of $10^{-4}$ $s^{-1}$ (the details are presented in the Supplementary Information) (Fittschen et al., 2019). Figure. S1 (a) presents the modeled ROOOH concentrations during this campaign, with a maximum of about $4.4\times10^9$ $cm^{-3}$. The correlation of the modeled ROOOH concentrations and the ratios of OH observations to OH simulations, and the correlation of the modeled ROOOH concentrations and the difference between OH observations and simulations both demonstrated that no significant relevance between ROOOH and the underestimation of OH radicals, as shown in Fig. S1 (b-c). Additionally, the ROOOH values modeled in our another campaign (Taizhou, 2018) were comparable to or even slightly higher than the simulations in this study, and the chemical modulation tests in Taizhou confirmed the ROOOH is not a significant OH interference in our PKU-LIF system (Ma et al., 2022).

Overall, the OH interference during this campaign was negligible according to the analysis of the behavior of PKU-LIF system in previous campaigns, the comparison of environmental conditions between this campaign and Wangdu campaign, and the exploration of the impact of ROOOH on the discrepancy of OH observations and simulations. However, we should acknowledge that the unmeasured interference might have an effect on radical measurement. More precise chemical modulation tests are needed in the future.

**(2) Figure S1 in the Supplementary Information**

To evaluate the impact of the interference from ROOOH on radical concentrations, we integrated the reactions of the ROOOH production and destruction into the base model herein, as shown in Eq. (S1-S2).

$$RO_2 + OH \rightarrow ROOOH \qquad\qquad\qquad\qquad\qquad S1$$

$$ROOOH \rightarrow RO + HO_2 \qquad\qquad\qquad\qquad\qquad S2$$

where the $RO_2$ is across all $RO_2$ radicals in the model excluding methyl peroxy radicals, for which it has been shown that the production of a trioxide species is only a minor product channel while the trioxide yield is expected to be close to 1 for larger peroxy radicals. The rate constant of Eq. (S1) is $1.5 \times 10^{-10}$ cm$^3$ s$^{-1}$ (Fittschen et al., 2019). In Eq. (S2), the rate constant is $10^{-4}$ s$^{-1}$, leading to ROOOH lifetimes of around 3 h, of the same order as the lifetime of ROOH species (Fittschen et al., 2019).

[Figure]

Figure S1: (a) The timeseries of ROOOH concentration from the reactions of $RO_2$ radicals excluding methyl peroxy radicals with OH radicals. (b) The correlation of the modeled ROOOH concentrations and the ratios of OH observations to OH simulations. (c) The correlation of the modeled ROOOH concentrations and the difference between OH observations and simulations. Only daytime values were chosen in (b-c).

4.  The authors have provided additional information regarding interferences in their $HO_2$ measurements and have highlighted in their discussion that their measurements actually reflect $HO_2^*$ rather than $HO_2$. This results in the model overestimating the measured concentrations, in contrast to the apparent agreement between the measurements and the model when the measurements were assumed to reflect only $HO_2$. Unfortunately, there is little discussion about this result except to say that the model may be missing heterogeneous uptake onto aerosols. Can the authors estimate the loss of peroxy radicals on aerosols and whether it could explain the discrepancy? Is this result consistent with the other studies highlighted in the paper? Given the change in the model- measurement agreement, the paper would benefit from an additional discussion related to the $HO_2^*$ measurement and modeled discrepancy, which appears to be more significant than the discrepancy between the measured and modeled OH concentrations.

    Also, the authors state that the fact that the $HO_2$-to-OH conversion efficiency was greater than 100% during the high NO mode. It is not clear how this was measured. Was this measured as part of calibrations of the $RO_2$ conversion efficiencies? This should be clarified in the revised manuscript to give confidence in the measured and modeled $HO_2^*$ concentrations.

**Reply**

Thanks for your suggestions. As for the $HO_2$ heterogeneous uptake, we added a new section (Section 4.3) to present the quantitative analysis in the revised manuscript.

As for the $HO_2$-to-OH conversion efficiency, our statement of 'the conversion efficiency was greater than 100% during the high NO mode' was not accurate, we replaced the expression with 'the conversion efficiency reached 100% during the high NO mode'. We determined the $HO_2$-to-OH conversion efficiency by calibrating the PKU-LIF system. In the calibration experiments of PKU-LIF, we set two different modes, which are HOx mode and $HO_2$ mode. Photolysis of water molecules at 185 nm leads to the production of OH radical and H atom and it is generally assumed that the H-atoms are completely converted to $HO_2$ radical. In this mode, equal amounts of OH and $HO_2$ radicals are present simultaneously, and it is called HOx mode. The OH signal ($C^*$) includes the OH from the water photolysis ($C_1$) and the OH which is converted by $HO_2$ and NO in the detection cell ($C_2$).

In another mode, CO is added as an OH scavenger, in order to convert all OH to $HO_2$ between photolytical generation and intake into the detection cell, which is called $HO_2$ mode. In this mode, OH is converted to $HO_2$ by CO in the calibration cell. The amounts of $HO_2$ from the water photolysis and the $HO_2$ converted by OH via CO are equal, which are equal to the OH from the water photolysis separately. The total OH signal ($C^\#$) in this mode denotes the sum OH which is converted by the HO2 radicals from the water photolysis and by the HO2 radicals from the reaction of OH and CO, and thus the OH signal which is converted by $HO_2$ from the water photolysis is $C^\#/2$.

The $HO_2$-to-OH conversion efficiency ($V$) can be denoted by the ratio of the OH signal which is converted by $HO_2$ and NO in the detection cell ($C_2$) to the $HO_2$ from the water photolysis which is equal to the OH from the water photolysis ($C_1$). Besides, $C_2$ in HOx mode is equal to $C^\#/2$ in $HO_2$ mode, and $C_1$ in HOx mode is equal to the difference between $C^*$ and $C_2$. Therefore,

the conversion efficiency ($V$) is calculated according to the Eq. (1).

$$V = \frac{C_2}{C_1} = \frac{c^\#/2}{c^* - c_2} = \frac{c^\#/2}{c^* - c^\#/2} \tag{1}$$

In this campaign, NO mixing ratios were switched between 25 ppm (low NO mode) and 50 ppm (high NO mode). We calculated the $HO_2$-to-OH conversion efficiency in low and high NO modes according to Eq. (1). In high NO mode, the value of $C_2$ was larger than the value of $C_1$, indicating the reactions of $RO_2$ and NO lead to the additional OH signal.

**Revision**

**(1) $HO_2$ heterogeneous uptake in Section 4.3**

The $HO_2$ heterogeneous uptake has been proposed to be a potential sink of $HO_2$ radicals, and thus could influence the radical chemistry and the formation of secondary pollution, especially in high-aerosol environments (Song et al., 2021;Song et al., 2022;Tan et al., 2020;Kanaya et al., 2000;Kanaya et al., 2007;Li et al., 2019). The impact of $HO_2$ uptake chemistry on radical concentration is different under different environmental conditions (Whalley et al., 2015;Mao et al., 2010;Li et al., 2019). To evaluate the contribution of $HO_2$ uptake chemistry to radical concentrations in this study, we coupled $HO_2$ heterogeneous uptake into the base model (RACM2-LIM1) and conducted three sensitivity experiments, as shown in R1 and Eq. (3).

$$HO_2 + aerosol \rightarrow products \tag{R1}$$

$$k_{HO_2+aerosol} = \frac{\gamma * ASA * v_{HO_2}}{4} \tag{3}$$

where ASA [$\mu m^2\ cm^{-3}$], which represents the aerosol surface area concentration, can be estimated by multiplying the mass concentration of $PM_{2.5}$ [$\mu g\ m^{-3}$] by 20 here because there were no direct ASA observations in this campaign (Chen et al., 2019;Wang et al., 2017). $v_{HO_2}$, which can be calculated by Eq. (4), refers to the mean molecular velocity of $HO_2$ with a unit of $cm\ s^{-1}$.

$$v_{HO_2} = \sqrt{\frac{8 * R * T}{0.033 * \Pi}} \tag{4}$$

where T [K] and R [$J\ mol^{-1}\ K^{-1}$] denote the ambient temperature and gas constant. $\gamma$, the $HO_2$ effective uptake coefficient, parameterizes the influence of some processes (Tan et al., 2020). $\gamma$ varies in the highly uncertain range of 0-1 (Song et al., 2022), and is the most critical parameter to impact $HO_2$ uptake chemistry. Only several observations of $\gamma$ have been reported (Taketani et al., 2012;Zhou et al., 2021;Zhou et al., 2020). The measured $\gamma$ at the Mt. Tai site and Mt. Mang site were 0.13-0.34 and 0.09-0.40, respectively (Taketani et al., 2012). The average value of the measured $\gamma$ was 0.24 in Kyoto, Japan in the summer of 2018 (Zhou et al., 2020). Zhou et al. (2021) reported the lower-limit values for median and average values of the measured $\gamma$ were 0.19 and 0.23±0.21 in Yokohama, Japan in the summer of 2019. Additionally, Li et al. (2018)

set 0.2 as the value of γ in the model, and Tan et al. (2020) calculated the γ of 0.08±0.13 by the analysis of the measured radical budget in Wangdu.

Here, we applied the two γ (0.2 and 0.08), which have been used in the model, to evaluate the impact of $HO_2$ uptake on radical concentrations, as shown in Fig. 6. The modeled $HO_2^*$ cannot match well with the observations when γ of 0.08 and 0.2 was set in the model. As the γ increased to approximately 0.3, good agreement between the modeled and observed $HO_2^*$ concentration was achieved, demonstrating that the significant heterogeneous uptake might exist in this campaign.

It is noted that the estimated strong influence is speculative because of the uncertainties of measurements and simulations. Overall, the γ evaluated in this study was comparable with those observed at the Mt. Tai and Mt. Mang in China, and Kyoto and Yokohama in Japan.

[Figure]

Figure 6: The diurnal profiles of the observed and modeled radical concentrations. The red and blue areas denote 1-σ uncertainties of measured and simulated radical concentrations by the base model, respectively. The orange, purple and black lines denote the simulations by the model which added the $HO_2$ heterogeneous uptake with different uptake coefficient. The grey areas denote nighttime.

**(2) $HO_2$ conversion efficiency in Section 2.2**

$HO_2$-to-OH conversion efficiencies in low NO mode ranged within 80%-95%, while those in high NO mode reached 100%, demonstrating that the high NO concentration is sufficiently to achieve complete $HO_2$ to OH conversion and thus the $HO_2$ measurement was affected by $RO_2$ radicals.

5. As mentioned in my previous review, the authors should clarify that their calculation of the rate of ozone production reflects the gross rate of production not the net rate, as it does not take into account the $NO_2$ formed that does not lead to $O_3$ production through the formation

of $HNO_3$ from the $OH + NO_2$ reaction. As I mentioned previously, Tan et al. (2017) used the net rate of ozone production in their analysis of the chemistry at the Wangdu site. Note the difference between equation 5 in Tan et al. (2017) and equation 3 in this paper. In contrast, Tan et al. (2018) calculated the gross rate of ozone production in equation 6 in their paper. These differences should be clarified as the authors still compare their estimated rates of ozone production to the results from both the Tan et al. 2017 and Tan et al., 2018 (lines 438-439 of the revised manuscript).

**Reply**

Thanks for your helpful suggestions. The reaction of $NO_2$ and OH was missed in equation 5 (equation 7 in the revised version) in the previous response and we have revised the equation 5 (equation 7 in the revised version) in the revised manuscript. However, in the calculation of $P(O_3)$ in the previous response, we included the reaction of OH and $NO_2$, as shown in the figure S7 in the revised Supplementary Information. Thus, the values of $P(O_3)$ in the previous response has denoted the net $O_3$ production rate, and the Fig. 8 (b) was not be changed.

Overall, in Section 4.5 'AOC evaluation', the $P(O_3)$ in equation 5 (equation 7 in the revised version) was revised. Besides, the comparison between the $O_3$ production rate in this study and that in other studies was revised. Additionally, the Fig. 8 (c-d) was revised, because we redefine the $AOC_{VOCs}$. We included the AOC derived from CO and OVOCs in $AOC_{VOCs}$ in the previous response, and herein, we only included the AOC derived from primary VOCs (excluding OVOCs, and mainly alkanes, alkenes, aromatics and isoprene) in $AOC_{VOCs}$.

**Revision**

**(1) In the revised version, we revised the equation 7, and the equation 5-6 were not be changed.**

$$F(O_3) = k_{HO_2+NO}[HO_2][NO] + \sum_i k_{RO_2i+NO}[RO_2]_i[NO] \tag{5}$$

$$L(O_3) = \theta j(O^1D)[O_3] + k_{O_3+OH}[O_3][OH] + k_{O_3+HO_2}[O_3][HO_2] +$$

$$(\sum(k^i_{alkenes+O_3}[alkenes^i]))[O_3] \tag{6}$$

$$P(O_3)_{net} = F(O_3) - L(O_3) - k_{NO_2+OH}[NO_2][OH] \tag{7}$$

**(2) The comparison between the $O_3$ production rate in this study and that in other studies.**

[revised manuscript text omitted]

---

## Author Response (AR3)

**Response to Editor:**

**Comments:**

1. In your reply to reviewer #1 regarding the $RO_2$ interference yields applied, you state 'The $RO_2$ interference yields which have been used for correction were the reported data in Lu et al. (2012).' It is not clear from your response nor from the changes made in the manuscript whether this correction considers possible differences in the $RO_2$ interference yields owing to differences in the amount of NO added to the detection cell in the two studies? I suggest that the concentration of NO added in both studies is stated in this manuscript and, if they are different, an explanation of how the $RO_2$ interference yield was scaled for the 2018 campaign should be provided.

**Reply**

Thanks for your helpful suggestions. In the PRIDE-PRD2006 campaign, Lu et al. (2012) injected pure NO into the $HO_2$ detection cell, and the high NO concentration is sufficient to achieve the complete $HO_2$-to-OH conversion. Fuchs et al. (2011) reported that the relative $RO_2$ detection sensitivities are approximately constant when the NO concentration is so high that $HO_2$ conversion in the detection is nearly complete. Thus, the relative $RO_2$ detection sensitivities reported by Lu et al. (2012) can be used for the correction of $HO_2$ concentrations as long as the $HO_2$-to-OH conversion efficiencies reach 100%. In this study, in the high NO mode (the NO concentration was 50 ppm, and the $HO_2$-to-OH conversion efficiencies reached 100%) in this study, we used the relative $RO_2$ detection sensitivities reported by Lu et al. (2012). We revised the description in Section 2.2 in the revised version.

**Revision**

**Section 2.2**

(1) Fuchs et al. (2011) reported that the relative $RO_2$ detection sensitivities are approximately constant when the NO concentration is so high that $HO_2$ conversion in the detection is nearly complete. Thus, when the $HO_2$-to-OH conversion efficiencies reach 100%, the relative $RO_2$ detection sensitivities reported by Fuchs et al. (2011) and Lu et al. (2012) can be used for the correction of $HO_2$ concentrations.

(2) Herein, we simulated the $HO_2$ and $HO_2^*$ concentrations by the model, and the $RO_2$ interference yields which were used for correction were the modeled values reported by Lu et al. (2012) in the PRIDE-PRD2006 campaign in which the $HO_2$-to-OH conversion efficiencies also reached 100% due to the injection of pure NO in the $HO_2$ detection cell.

2. In your response to reviewers #2 and #3 regarding the possible impact of heterogeneous losses of $HO_2$ in the study, you have provided a discussion on how heterogeneous losses can impact the agreement between the modelled and measured $HO_2$* and put this into the context of previous ambient measurements of γ. Further to this, however, I think the manuscript would benefit from a discussion on how inclusion of $HO_2$ uptake in the model impacts the modelled OH concentrations also - as figure 6 highlights that the modelled OH

concentration decreases as the $HO_2$ uptake coefficient is increased (leading to an increase in the modelled to measured OH discrepancy); this should be acknowledged and a discussion provided on how the impact of including heterogeneous $HO_2$ loss impacts on the required concentration of species X needed to close the OH budget (and how this then compares to the concentration of species X in earlier studies).

**Reply**

Thanks for your suggestions and we revised Section 4.3 and the Supplementary Information.

**Revision**

**Section 4.3:**

It should be noted that the $HO_2$ heterogeneous uptake ($\gamma = 0.3$) reduced the modeled OH concentrations by around 20% compared to the OH simulations in the base model during the daytime (08:00-18:00). Sensitivity tests illustrated that good agreements of OH observations-simulations and $HO_2^*$ observations-simulations were both achieved when the amount of X changed from 0.1 ppb to 0.25 ppb and the $HO_2$ effective uptake coefficient was 0.3, as shown in Fig. S6 in the Supplementary Information. Compared to the Backgarden and Heshan sites, the amount of X in Shenzhen was lower despite a significant $HO_2$ heterogeneous uptake, which might be closely related to the environmental conditions as discussed in Sect. 4.2.

**Supplementary Information:**

[Figure]

Figure S6: NO dependence on OH and $HO_2^*$ radicals. The red box-whisker plots give the 10%, 25%, median, 75%, and 90% of the HOx observations. The blue circles show the median values of the HOx simulations by the base model, and the purple circles show the HOx simulations by the model with X mechanism (X = 0.25 ppb NO) and $HO_2$ heterogeneous uptake ($\gamma = 0.3$). Only daytime values and NO concentration above the detection limit of the instrument were chosen.

3. With regards to ROOOH species acting as potential OH interferences within the FAGE detection cell, I believe the manuscript (and readers) would benefit from some further details at the beginning of this section on how OH may be formed from the decomposition of ROOOH species within FAGE detection cells. In the Fittschen et al. 2019 paper, they suggest heterogeneous decomposition of ROOOH on the walls of the FAGE cell or the entrance nozzle could yield OH.

**Reply**

Thanks for your suggestions, and we added the description of how OH may be formed from the ROOOH decomposition in the revised manuscript.

**Revision**

**Section 2.2**

Fittschen et al. (2019) reported that the OH interference signals might come from the ROOOH heterogeneous decomposition on the walls of the FAGE cell or the entrance nozzle, but they also noted that the ROOOH interference is highly dependent on the design and measurement conditions of different FAGE instruments.

4. In response to reviewer #3 regarding gross vs net ozone production rates I suggest changing '…while it was much higher than the $O_3$ production rate in Beijing in winter despite being the gross production rate (Tan et al., 2018).' to '…while the net ozone production rate in Shenzhen was much higher than the gross $O_3$ production rate in Beijing in winter (Tan et al., 2018).'

**Reply**

We have revised the expression in the revised version.

**Revision**

The modeled $P(O_3)_{net}$ in this study was comparable to the net $O_3$ production rate in Wangdu in summer (Tan et al., 2017), while the net ozone production rate in Shenzhen was much higher than the gross $O_3$ production rate in Beijing in winter (Tan et al., 2018)

**References**

Tan, Z., Fuchs, H., Lu, K., Hofzumahaus, A., Bohn, B., Broch, S., Dong, H., Gomm, S., Haeseler, R., He, L., Holland, F., Li, X., Liu, Y., Lu, S., Rohrer, F., Shao, M., Wang, B., Wang, M., Wu, Y., Zeng, L., Zhang, Y., Wahner, A., and Zhang, Y.: Radical chemistry at a rural site (Wangdu) in the North China Plain: observation and model calculations of OH, HO2 and RO2 radicals, Atmospheric Chemistry and Physics, 17, 663-690, 10.5194/acp-17-663-2017, 2017.

Tan, Z., Rohrer, F., Lu, K., Ma, X., Bohn, B., Broch, S., Dong, H., Fuchs, H., Gkatzelis, G. I., Hofzumahaus, A., Holland, F., Li, X., Liu, Y., Liu, Y., Novelli, A., Shao, M., Wang, H., Wu, Y., Zeng, L., Hu, M., Kiendler-Scharr, A., Wahner, A., and Zhang, Y.: Wintertime photochemistry in Beijing: observations of ROx radical concentrations in the North China Plain during the BEST-ONE campaign, Atmospheric Chemistry and Physics, 18, 12391-12411, 10.5194/acp-18-12391-2018, 2018.